# Benchmarking porcine pancreatic ductal organoids for drug screening applications

Christos Karampelias [ID] [1,2], Kaiyuan Yang[1,2], Falk J Farkas [ID] [1,2,3], Michael Sterr[1,2], Mireia Molina van Den Bosch[1,2,3], Simone Renner[2,4,5], Janina Fuß[6], Christine von Toerne[7], Sören Franzenburg[6], Tatsuya Kin [ID] [8], Eckhard Wolf [ID] [2,4,5], Elisabeth Kemter [ID] [2,4,5] & Heiko Lickert [ID] [1,2,3 ✉]

## Abstract

**Primary human pancreatic ductal organoids (HPDO) have emerged as a model to study pancreas biology and model disease like pancreatitis and pancreatic cancer. Yet, donor material availability, genetic variability and a lack of extensive benchmarking to healthy and disease pancreas limits the range of applications. To address this gap, we established porcine pancreatic ductal organoids (PPDO) as a system from a reliable, genetically defined and easily obtainable source to model pancreatic ductal/progenitor biology. We benchmarked PPDO to HPDO and primary porcine pancreas using single-cell RNA sequencing (scRNA-Seq). We observed no overt phenotypic differences in PPDO derived from distinct developmental stages using extensive proteomics profiling, with a WNT/basal cell signaling enriched population characterizing PPDO. PPDO exhibited differentiation potential towards mature ductal cells and limited potential towards endocrine lineages. We used PPDO as a chemical screening platform to assess the safety of FDA-approved drugs and showed conserved toxicity of statins and α-adrenergic receptor inhibitors between PPDO and HPDO cultures. Overall, our results highlight the PPDO as a model for mammalian duct/progenitor applications.**

**Keywords** Pancreatic Ductal Organoids; Porcine Pancreas; Chemical Screen; Cross-species Comparison; scRNA-Seq Profile; Proteome Profile; Omics Profile
**Subject Categories** Digestive System; Metabolism; Methods & Resources

## Introduction

The homeostatic turnover of the mammalian adult pancreas is a relatively slow process and has been postulated to rely mainly on cell proliferation (Sznurkowska et al, 2018; Desai et al, 2007; Zhao et al, 2021; Lodestijn et al, 2021). Whether the adult human pancreas contains self-renewing and/or adult stem cells that contribute to exocrine or endocrine tissue (re)generation is still an open question. It is known that the embryonic multipotent ductal epithelium gives rise to all endocrine, exocrine and ductal cells in mammals, but in vivo lineage-tracing studies have shown that ductal progenitors are lineage restricted and do not contribute to other lineages in the adult pancreas (Kopp et al, 2011; Solar et al, 2009; Kopinke et al, 2011; Bastidas-Ponce et al, 2017; Zhao et al, 2021). Exocrine tissue maintenance and regeneration is less studied. Recent work suggested that acinar cell maintenance occurs mainly through self-replication, while some studies indicated the presence of acinar progenitor populations that contribute to tissue maintenance and might be involved in carcinogenesis (Jiang et al, 2023; Lodestijn et al, 2021). Studies on identifying adult human pancreatic progenitors are limited, yet there are indications that postnatal ductal cells could behave as such (Qadir et al, 2020, 2018; Doke et al, 2023; Karampelias et al, 2022; Rovira et al, 2021). Given the limitations to study human adult pancreatic ductal/progenitor biology in large scale, new and thoroughly characterized organoid models are needed to advance basic biology and regenerative medicine.

Organoids are 3D culture models initially established from tissues known to harbor adult stem cells in their epithelium (e.g., the intestine) and their properties are being standardized according to recent community guidelines (Sato et al, 2009; Marsee et al, 2021). In this regard, duct-derived pancreatic organoids from primary tissue can be used as a proxy to address questions on adult pancreatic progenitor and/or ductal biology and disease. Primary adult tissue pancreatic organoids were first generated from mouse ductal cells (Huch et al, 2013; Rovira et al, 2010; Boj et al, 2015; Broutier et al, 2016). Several groups have generated human ductal organoids both from exocrine pancreatic tissue and from pluripotent stem cells derived pancreatic progenitors (Loomans et al, 2018; Rezanejad et al, 2018; Karampelias et al, 2022; Gonçalves et al, 2021; Wiedenmann et al, 2021; Breunig et al, 2021; Huang et al, 2015, 2021; Boj et al, 2015; Zook et al, 2024; Bakhti et al, 2019). Stem cell-derived organoids resemble a more embryonic progenitor phenotype with greater differentiation potential towards endocrine lineages compared to primary HPDO.

[1]Institute of Diabetes and Regeneration Research, Helmholtz Munich, Neuherberg, Germany. [2]German Center for Diabetes Research (DZD), Neuherberg, Germany. [3]School of Medicine and Health, Technische Universität München, Munich, Germany. [4]Gene Center and Center for Innovative Medical Models (CiMM), LMU Munich, Munich, Germany. [5]Interfaculty Center for Endocrine and Cardiovascular Disease Network Modelling and Clinical Transfer (ICONLMU), LMU Munich, Munich, Germany. [6]Institute of Clinical Molecular Biology, Christian-Albrechts-Universität Kiel, Kiel, Germany. [7]Metabolomics and Proteomics Core, Helmholtz Center Munich, German Research Center for Environmental Health, D-80939 Munich, Germany. [8]Clinical Islet Laboratory, University of Alberta Hospital, Edmonton, AB, Canada. ✉E-mail: heiko.lickert@helmholtz-munich.de

So far, pancreatic ductal organoids have mainly been used to model pancreatic cancer and to study endocrine cell differentiation to identify disease mechanisms and advance regenerative therapies. However, human pancreatic tissue for organoid generation is limited by the availability of donor material and suffers from fast degradation due to acinar enzyme mediated digestion. This highlights the need for new, relevant and thoroughly phenotyped and characterized pancreatic organoid models to advance our knowledge of pancreatic adult progenitor populations and pancreatic disease.

To address this technological gap, we generated and characterized in-depth porcine pancreatic ductal organoids (PPDO) across distinct developmental stages. We used PPDO to model ductal/progenitor biology dynamics in vitro and to study drug-induced exocrine-related pathologies (pancreatitis-pancreatic cancer). Our reasoning was that PPDO can serve as a model to study drug toxicology and uncover biological mechanisms of pancreas development and homeostasis. The pig serves as a large animal model to study pancreas biology with its advantages being resemblance in terms of size and metabolism to humans coupled to efficient genetic modification for disease modeling (Renner et al, 2020). Moreover, the availability of porcine pancreata from developmental stages (embryonic, early post-natal) that are difficult to obtain from human donors, makes them an ideal surrogate to study pancreas biology and disease using organoids. In this work, we observed that PPDO were transcriptome- and proteome-wise similar across ages, yet their molecular signature changed after prolonged maintenance in culture. To evaluate their suitability for translational applications, we performed a chemical screen using PPDO to assess the safety of FDA-approved drugs for pancreas diseases, translating these findings to HPDO. Overall, we provide in-depth benchmarking of PPDO as a model for adult pancreatic ductal/progenitor biology.

# Results

## Generation and characterization of PPDO across developmental stages

We generated PPDO from primary porcine pancreatic ductal cells across different ages. This is a unique advantage using the pig model to study human-relevant biology across distinct developmental stages. For the rest of this work, we will use the following nomenclature to clarify the age of the pig that the PPDO were derived from as embryonic (Em), early postnatal (EPN), late postnatal (LPN) and adult (Ad) (see Methods for detailed explanation). To generate PPDO, we manually picked ductal structures following pancreas digestion and cultured them with extracellular matrix (Cultrex) in human pancreatic ductal organoid (HPDO) medium (Broutier et al, 2016) (Fig. 1A). Phenotypically, PPDO initially formed large cystic structures, like the HPDO derived from human pancreatic exocrine fractions. We did not observe any differences in organoid formation capacity or overall morphology between PPDO generated from embryonic compared to postnatal stages (Fig. 1B–D). PPDO cultures could be freeze-thawed and were maintained up to passage 12, at which time point the cultures were cryopreserved due to experimental protocol ending. However, we noticed a phenotypic switch of the PPDO

upon prolonged culturing, with the cystic morphology giving rise to smaller, dense cellular spheres without a discernible lumen. PPDO maintained the typical ductal organoid morphology until passage 3 (early passage) with cultures transitioning to the dense sphere morphology from passage 4 (transitioning passages), and this population dominating the cultures from passage 6 onwards (late passage) (Fig. 1B'–D'). No such morphological transformation occurred for HPDO in our hands, although the HPDO size became smaller over increasing passage number (Fig. 1E–E').

We profiled key pancreatic protein markers in PPDO, including, acinar/pancreatic progenitor and endocrine specific proteins. Immunostaining confirmed that the PPDO express ductal (CF transmembrane conductance regulator (CFTR), SRY-Box transcription factor 9 (SOX9), keratin 7 (KRT7)) and epithelial markers (Pancreatic and duodenal homeobox 1 (PDX1), cadherin 1 (CDH1)), markers but were devoid of endocrine (NK6 Homeobox 1 (NKX6-1), glucagon (GCG) and acinar cell markers (glycoprotein 2 (GP2). F-actin (phalloidin) immunostaining showed higher intensity in the luminal area of the PPDO suggesting apical-basal polarization of the epithelium (Fig. 1F±H). This protein expression pattern was consistent across PPDO derived from different developmental stages, similar to HPDO and porcine pancreas and all cells uniformly expressing the ductal protein markers (Fig. EV1A–N). Moreover, we assayed ductal functionality of early passage PPDO with the CFTR functionality assay, utilizing both forskolin (non-physiological) and secretin (physiological) to activate ductal cell bicarbonate excretion. If the cultures include mature, CFTR-expressing ductal cells, stimulation will lead to fluid uptake and lumen expansion. Lumen area of PPDO significantly expanded following treatment with both agonists similarly to HPDO (Fig. 1I–M) and across PPDO derived from different ages, suggesting functional ductal cells in terms of bicarbonate secretion. On the contrary, late passage PPDO did not expand in response to the stimulants, hinting at an immature ductal fate (Fig. EV1O). Overall, we established and showed the ductal fate of PPDO from distinct porcine developmental stages, a fate that appeared to change upon prolonged culturing.

## PPDO share similar proteome profiles regardless of developmental stage

We utilized proteomics to compare PPDO derived from different developmental stages and compare them to HPDO. To this end, we performed proteomics characterization of PPDO derived from embryonic, early postnatal and adult pig pancreas samples. In our analysis, we also included early and late passage (PS) PPDO from the same early postnatal samples to identify the proteomic signature of the observed phenotypic shift with passaging. In parallel, we performed proteomics on HPDO and the exocrine extract they were derived from to assess the human proteome for cross-species comparison. Principal component analysis (PCA) analysis revealed that the major determinant of variance in the PPDO dataset is the passaging while the developmental stage of the animal that PPDO were derived from had minimal effect on the proteomic signature (Fig. 2A). Similarly, HPDO and human exocrine tissue clustered separately on the PCA plot validating our approach (Fig. 2B). Moreover, not a single protein was

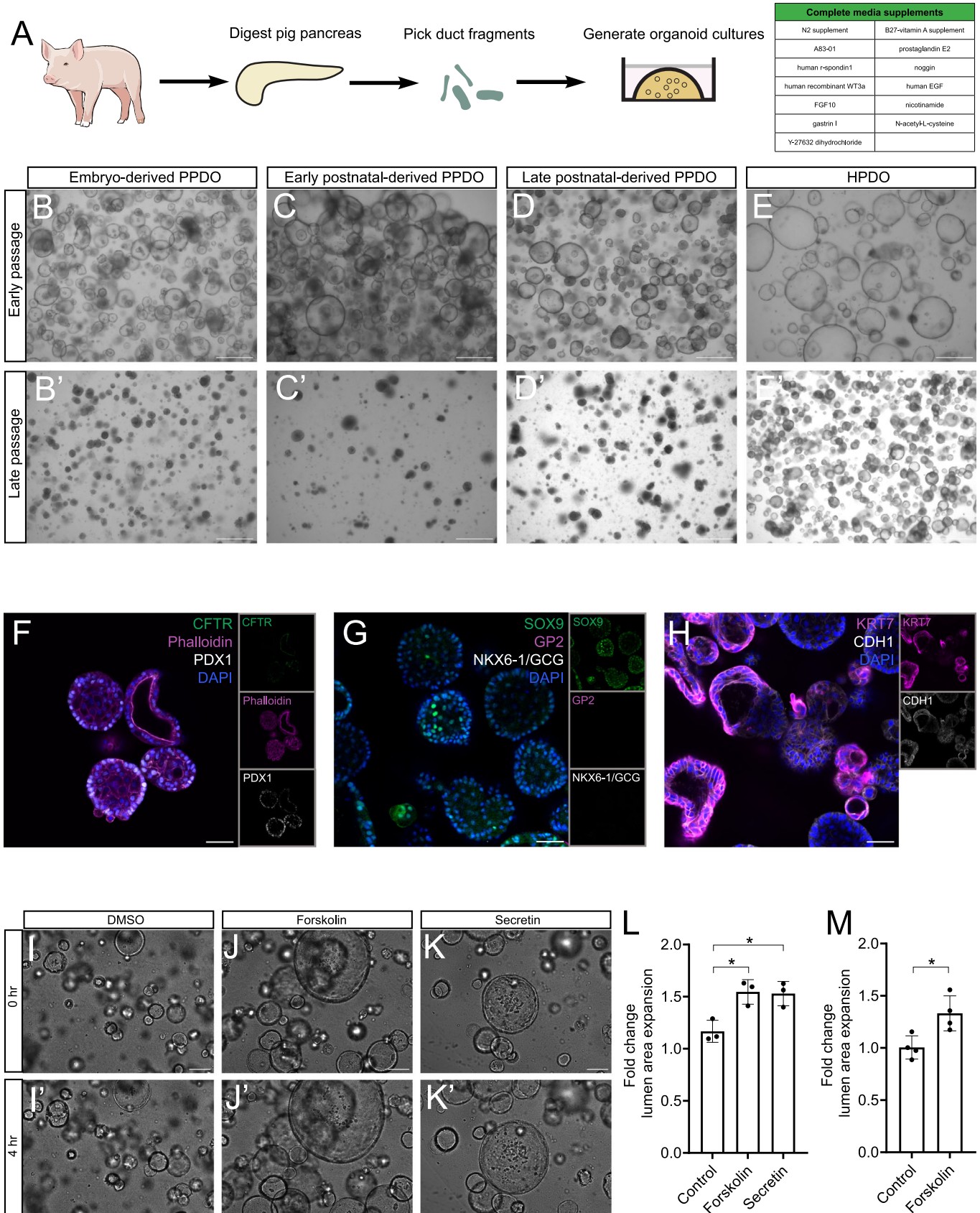

| Complete media supplements | |
|---|---|
| N2 supplement | B27-vitamin A supplement |
| A83-01 | prostaglandin E2 |
| human r-spondin1 | noggin |
| human recombinant WT3a | human EGF |
| FGF10 | nicotinamide |
| gastrin I | N-acetyl-L-cysteine |
| Y-27632 dihydrochloride | |

**Figure 1. Generation and characterization of PPDO across developmental stages.**

(A) Schema showing the process for generating PPDO. (B–E) Brightfield microscopy images of early passage PPDO and HPDO from Em- (B), EPN (C), LPN (D) and from a human donor (E). Scale bar: 500 µm. (B′–E′) Brightfield microscopy images of late passage PPDO and HPDO from Em (B′), EPN (C′), LPN (D′) and from a human donor (E). Scale bar: 500 µm. (F–H) Single-plane confocal images of PPDO derived from EPN pig pancreas immunostained against CFTR-Phalloidin-PDX1 (F), SOX9-GP2-NKX6-1/GCG (G), KRT7-CDH1 (H) and counterstained with DAPI. Insets show the individual channels of the merge image. PPDO were stained at passage 3. Scale bar: 50 µm. (I–K) Brightfield microscopy images of early passage Em PPDO at the beginning of the live imaging, treated with DMSO (I), forskolin (J), and secretin (K). Scale bar 500 µm. (I′–K′) Brightfield microscopy images of early passage Em PPDO at the end of the live imaging, treated with DMSO (I′), forskolin (J′) and secretin (K′). Scale bar 500 µm. (L) Quantification of the lumen area expansion of PPDO following 4 h of live imaging and treatments with forskolin and secretin. $n = 3$ independent experiments with different PPDO lines (1 Em, 1 EPN and 1 Ad). Data are shown as mean ± SD. One-way ANOVA followed by Dunnett's multiple comparisons test was used to assess significance. Pval are: *Padj = 0.015 for DMSO vs forskolin and *Padj = 0.0140 for DMSO vs secretin. (M) Quantification of the lumen area expansion of HPDO following 4 h of live imaging and treatment with forskolin. $n = 4$ independent experiments with 4 HPDO lines. Data are shown as mean ± SD. Mann–Whitney test was used to assess significance. *$P = 0.0286$. Source data are available online for this figure.

significantly changed when comparing EPN with Ad PPDO showcasing that these preparations are very similar (Fig. 2C). The phenotypic switch observed with passaging had a more profound effect on PPDO identity with mature duct and proliferation-relating proteins enriched in the early passage PPDO (Fig. 2D). Late passage PPDO were enriched for basal epithelial cell signature and key proteins in developmentally important pathways (Fig. 2D,E). In the HPDO dataset, there was a clear enrichment for acinar cell markers in the exocrine fractions and important ductal cell proteins in the HPDO proteomics samples showcasing the ductal identity of HPDO (Fig. 2F,G). Almost half of the top 100 proteins were similar between the early passage PPDO and HPDO datasets (Dataset EV1-tab Top 100). All significantly altered proteins along with the absolute, normalized values for all proteins detected can be found in Dataset EV1.

## Deep phenotyping of PPDO and benchmarking to HPDO by scRNA-Seq

To understand the molecular signature of PPDO at the single cell level, we used scRNA-Seq to transcriptionally profile PPDO and benchmark them against the state-of-the-art HPDO cultures. We sampled two PPDO lines (LPN/Ad), which were cultured for four and seven passages, respectively, i.e., most cells have transitioned to the less mature ductal phenotype. Similarly, we performed scRNA-Seq profiling of two HPDO lines originating from adult human pancreatic ductal cells at passage four and seven mixed in one sample. For the integrated PPDO dataset, we identified six cell populations/states corresponding to pancreatic ductal cells, a proliferating population, a potential progenitor population characterized by high WNT signaling, a population/state characterized by expression of basal polarity genes and a population/state that was enriched for glycolysis relating genes and processes (Fig. 3A–C). Moreover, the largest percentage of cell types/states corresponded to the WNT enriched and glycolysis relating populations with the mature pancreatic ductal cell state being the minority of the populations that was further reduced with passaging (Appendix Fig. S1A,B). scRNA-Seq of HPDO identified cell populations/states corresponding to pancreatic ductal cells, proliferating cells, glycolysis relating genes and processes state, a pancreatic progenitor population and a population enriched in ribosome-encoding genes (Fig. 3D,E). Compared to PPDO, HPDO had a higher percentage of mature pancreatic ductal cell signature genes but both cultures shared the proliferating and the glycolysis signature cell states (Fig. 3E). One notable difference was that the progenitor-like state of the HPDO was characterized by elevated

GP2 expression, an established marker of embryonic pancreatic progenitors (Ameri et al, 2017; Cogger et al, 2017), and did not have a clear WNT signature (Fig. 3D). These results correlate with the observed morphological changes, as HPDO maintain their cyst-forming phenotype, mainly characterized by mature ductal epithelial cells, while the mature ductal cell state is a minority in the PPDO that form the dense cultures upon prolonged passaging. Still, the PPDO-HPDO datasets integrated in two-dimensional space highlighting the general pancreatic ductal fate of both cultures (Fig. 3F). Of note, we validated the pancreatic ductal fate of HPDO by mapping the scRNA-Seq dataset to the human tabula sapiens atlas (Jones et al, 2022) (Appendix Fig. S1C,D). Our scRNA-Seq analysis demonstrated the transcriptional similarity of different PPDO lines and highlighted a potential transcriptional progenitor signature difference between PPDO and HPDO.

Using the scRNA-Seq datasets, we compared PPDO and HPDO in terms of major signaling pathways known to be active in pancreatic ductal/progenitor cells. As mentioned above, PPDO had higher expression of WNT-related genes with the prominent example of the WNT co-receptor, leucine-rich repeat containing G protein-coupled receptor 6 (LGR6), which was enriched in the progenitor cluster compared to HPDO (Fig. 3G,H). However, LGR4 expression was uniform among cell clusters/states suggestive of distinct function for the paralogues in WNT signaling in PPDO, as all LGR protein family members bind to RSPO1 (Xu et al, 2015), which is added to the medium. Higher Lymphoid enhancer-binding factor 1 (LEF1) expression in the LGR6+ cell cluster further strengthens the differential WNT signaling activity. Both cultures expressed key genes of the NOTCH signaling pathway, with PPDO showing an elevated expression of NOTCH ligands compared to HPDO cultures (Fig. 3I,J). PPDO and HPDO had similar expression levels of genes involved in the Hippo pathway (Fig. 3K,L). Lastly, we performed a ligand-receptor analysis to uncover intercellular signaling pathways that could be important for cellular identity and progenitor biology. For PPDO, there was a strong enrichment for epidermal growth factor (EGF) signaling ligand-receptor interactions, with additional ligand receptor pairs pointing to WNT and integrin signaling as enriched in these cultures (Appendix Fig. S2A,B). The most common ligand-receptor interactions in HPDO related to integrin signaling, followed by ligand-receptor interactions in the EGF, transforming growth factor (TGF) and mucin-related pathways (Appendix Fig. S2C,D). These differences are present independent of the medium composition as the same medium formulation is used for both HPDO and PPDO.

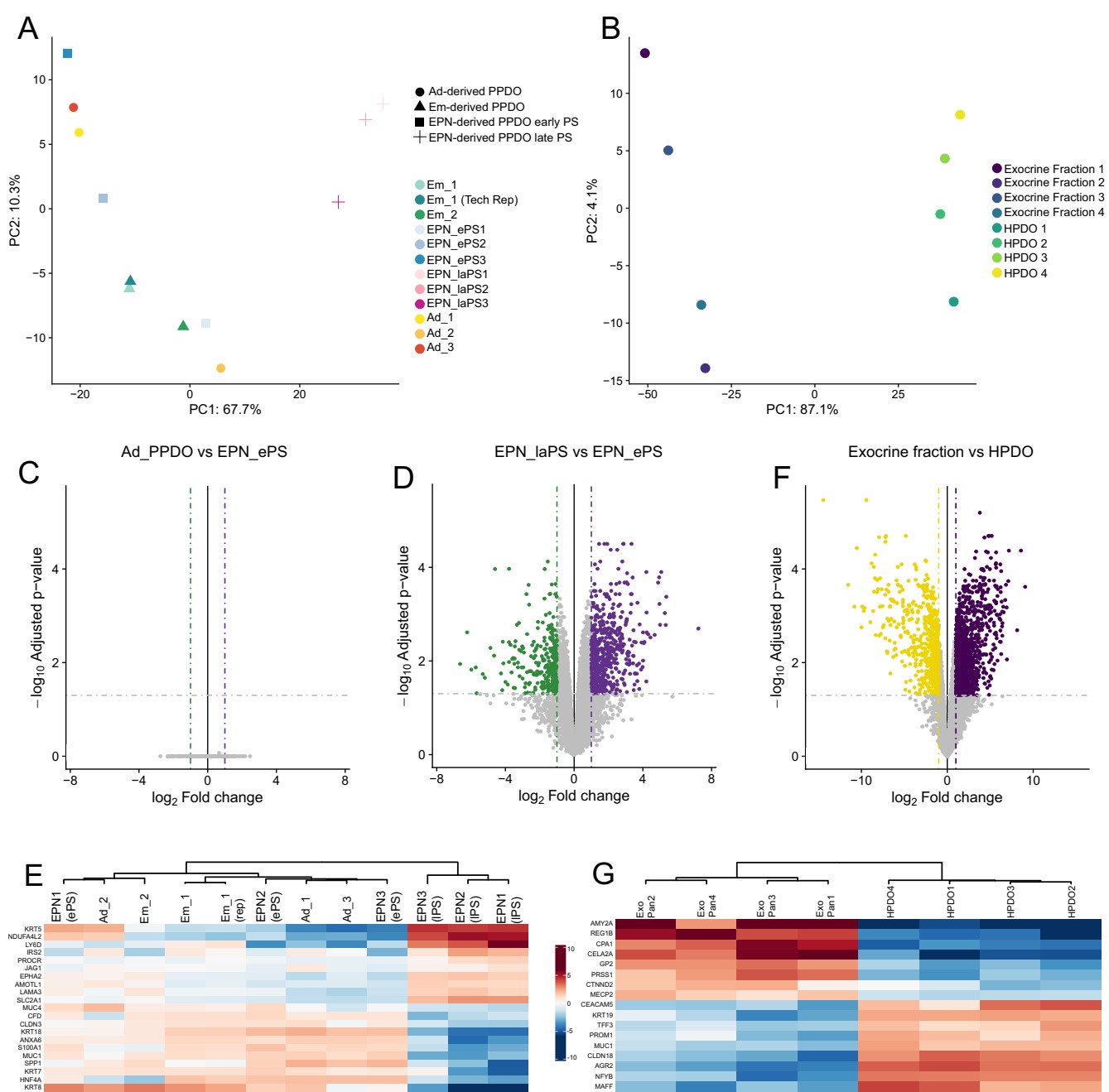

**Figure 2. PPDO share similar proteome profiles regardless of developmental stage.**

(A, B) PCA plots from the PPDO (A) and HPDO (B) proteomics datasets showing the first two principal components. (C, D) Volcano plots for the EPN versus Ad (C) and EPN early passage versus EPN late passage (D) comparisons on the proteomics datasets. pAdj cutoff-0.05 and fold change cutoff = 1. $n = 3$ PPDO lines per group. Statistical test was applied in the limma R package using a moderate t-test (based on a generalized linear modeling framework) with multiple testing correction (Benjamin–Hochberg method). (E) Heatmap showing the log2 centered intensity of the expression of selected proteins for each replicate of the PPDO dataset. (F) Volcano plot for the exocrine fraction versus HPDO differentially altered proteins. pAdj cutoff-0.05 and fold change cutoff = 1. $n = 4$ human donor samples for both exocrine fractions and HPDO. Statistical test was applied in the limma R package using a moderate t-test (based on a generalized linear modeling framework) with multiple testing correction (Benjamin–Hochberg method). (G) Heatmap showing the log2 centered intensity of the expression of selected proteins for each replicate of the HPDO proteomics dataset. Em = Embryo PPDO (2 biological and 1 technical replicate), EPN_ePS = Early postnatal_early passage (3 biological replicates), EPN_laPS = Early postnatal_late passage (3 biological replicates same PPDO as EPN_ePS), Ad = Adult PPDO (3 biological replicates).

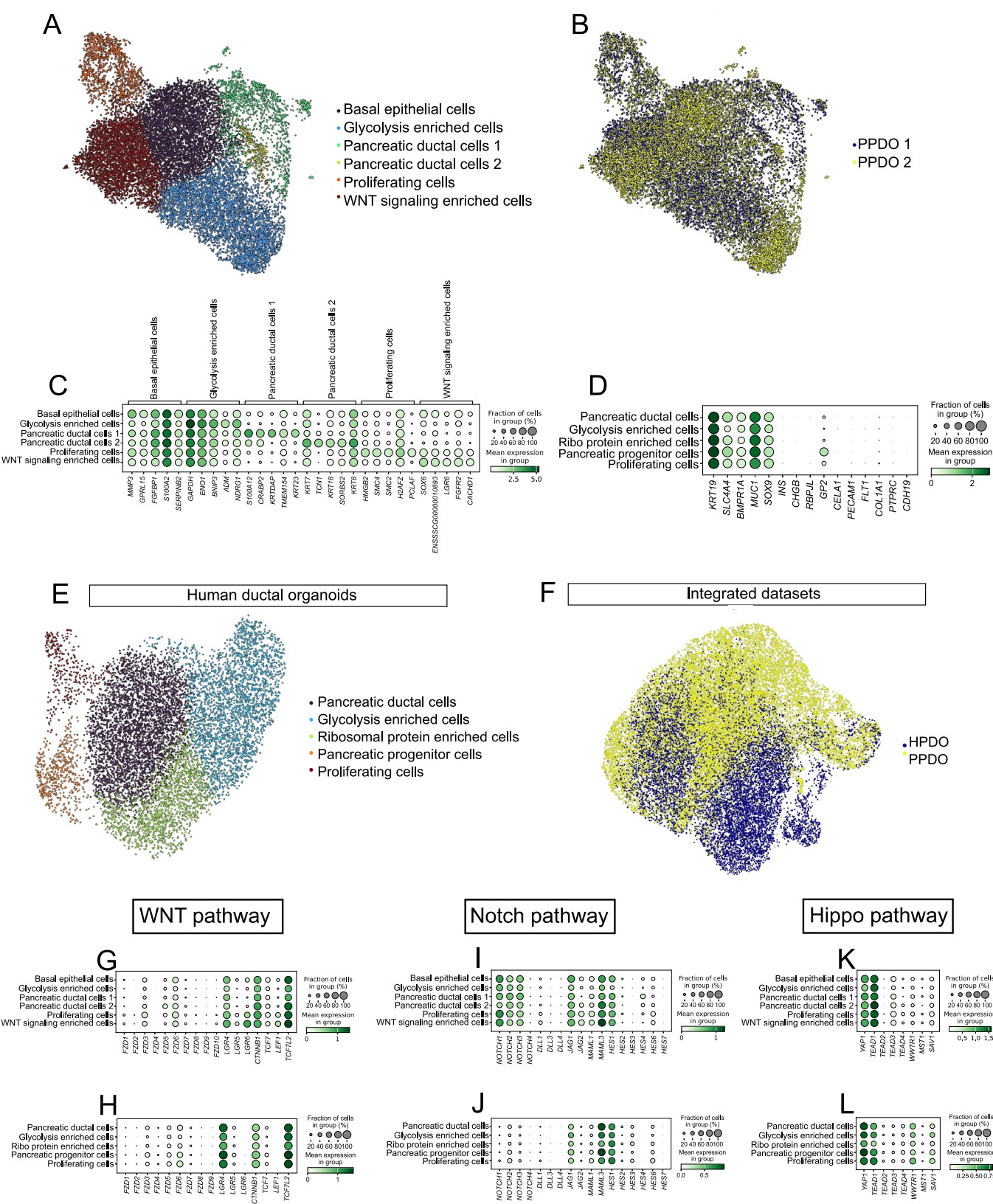

Figure 3.   Deep-phenotyping of PPDO and benchmarking to HPDO by scRNA-Seq.

(A, B) UMAPs showing the cell types/states (A) and sample composition (B) of the integrated scRNA-Seq data from two PPDO lines (1 LPN and 1 Ad) at passage 4 and 7. (C) Dot plot showing expression of the top 5 marker genes of each PPDO cluster. (D) Dot plot showing expression of key pancreatic gene markers for major cell types of the HPDO scRNA-Seq dataset including ductal (*KRT19-SLC4A4-BMPR1A-MUC1-SOX9*), endocrine (*INS-CHGB*), acinar/pancreatic progenitor (*GP2*), acinar (*RBPJL-CELA1*) endothelial (*PECAM1-FLT1*), stellate (*COL1A1*), immune (*PTPRC*), and schwann (*CDH19*) cell markers. (E) UMAP showing the cell types/states of scRNA-Seq data from HPDO. (F) UMAP showing the cross-species comparison of the integrated PPDO-HPDO scRNA-Seq datasets. (G–L) Dot plots of expression of important genes from the WNT (G, H), NOTCH (I, J) and Hippo (K, L) signaling pathways in the scRNA-Seq dataset of the integrated PPDO (G, I, K), and HPDO (H, J, L).

## Benchmarking PPDO to primary porcine pancreas

To benchmark PPDO to the primary tissue, we profiled porcine pancreata from two animals near reproductive maturity using scRNA-Seq and identified all major pancreatic cell lineages (Appendix Fig. S3A). First, we integrated and compared all cell types/states from the primary porcine pancreas and PPDO datasets. The two datasets did not cluster together in two-dimensional space, yet a small percentage of the PPDO cells were classified as primary pancreatic ductal cells (Appendix Fig. S3B,C). Integrating only the primary ductal and beta cell populations together with all the PPDO cells showed that the primary ductal cells clustered in latent space to the pancreatic ductal cells of the PPDO, showcasing the similarities between the two populations (Fig. 4A,B). To assess ductal identity, we checked the expression of the top five marker genes per PPDO cluster assigned as pancreatic ductal cells in the scRNA-Seq of the primary porcine pancreas. This comparison showed that the pancreatic ductal cell 2 population/state of PPDO was more similar to the primary ductal cells, while the pancreatic ductal cell 1 population appeared to be a culture-specific cell state/type (Appendix Fig. S3D). Importantly, we noticed that the porcine duct contains a cell population characterized by *LGR5* and anterior gradient 2, protein disulfide isomerase family (*AGR2*-recently shown to be expressed in WNT enriched population of mouse ductal cells (Fernández et al, 2024)) gene expression implying a WNT responsive population with progenitor characteristics (Fig. 4C–E). Immunostaining of porcine pancreas and PPDO against AGR2 indicated subpopulations of cells with AGR2 protein expression. This heterogeneity was more pronounced in the PPDO cultures (Fig. 4F,G).

To test the possibility that the WNT/BMP positive population gives rise to PPDO, we initiated PPDO cultures comparing complete medium and medium depleted of cytokines that modulate the WNT and BMP signaling, namely WNT3,-R-Spondin-1, Noggin (WRN). Initial organoid formation was unaffected in the two conditions suggesting that PPDO can form in the absence of a WNT/BMP signaling pulse (Fig. 4H,I). However, the cultures without WRN factors had limited expansion potential with PPDO decreasing progressively in number and disappearing between passage three to six (Fig. 4J,K). A similar phenotype in response to WRN depletion was observed in HPDO (Fig. 4L–O). To directly assess the WNT signaling role in organoid survival/proliferation, we cultured PPDO and HPDO for one passage without WNT3a/R-Spondin-1 and treated them with a WNT pathway inhibitor in a dose–response manner. In both PPDO and HPDO proliferation was completely halted upon WNT inhibition and the drug was toxic for the cultures (Fig. EV2A–H). In summary, our data benchmarked the transcriptional similarities and differences of in vitro derived PPDO to the in vivo primary porcine pancreas,

pointing to both WNT/BMP$^+$ and WNT/BMP$^-$ populations with organoid forming capacity.

## Differentiation potential of PPDO

Embryonic ductal epithelium contains cells with bipotent potential towards ductal and endocrine lineages and are derived from the multipotent pancreatic progenitors with tri-lineage potential (acinar-ductal-endocrine cells). To explore the utility of PPDO for studying pancreatic progenitor biology, we tested different endocrine differentiation protocols to assess PPDO's developmental capacity towards the endocrine lineage. First, we complemented the basal organoid culture medium with porcine serum, aiming to mimic the in vivo milieu. Gene expression analysis did not show any significant induction of genes linked to endocrine cell fate apart from induction of neuronal differentiation 1 (*NEUROD1*) (Fig. EV3A,B,E–J). Next, we tested various combinations of protocols for differentiating human embryonic stem cells to β-cells, since a similar approach has been recently shown to differentiate primary mouse pancreatic organoids towards endocrine fate (Fernández et al, 2024; Bakhti et al, 2019). However, this approach, in particular S5 medium used for endocrine progenitor induction, was not tolerated by the PPDO, inducing massive cell death (Fig. EV3C,D). Testing another proposed differentiation protocol efficient for human organoid differentiation (Loomans et al, 2018) also yielded high organoid toxicity and no endocrine cells (Fig. EV3K,L).

Next, we tested removing the proliferating cytokines from the PPDO medium could lead to cell fate changes, similar to the protocol reported for gastric organoid differentiation to insulin producing cells (Huang et al, 2023). To address the changes on a global scale, we performed bulk transcriptome analysis of different PPDO lines cultured for seven days in complete and cytokine-depleted media (see Methods for details). Of note, this protocol was toxic to HPDO suggesting that the effects are restricted to PPDO (Fig. EV3M,N). Differential gene expression analysis showed a clear transcriptional segregation of PPDO cultured in the two media. PPDO cultures from different developmental stages clustered together (Fig. EV4A). This demonstrated that the developmental stage that the PPDO was derived from did not have a substantial effect on cell identity and the transcriptional signature (Fig. EV4A). To further support the ductal cell identity of PPDO, we deconvoluted the bulk RNA-Seq signature against our scRNA-Seq dataset. Using this approach, we validated with a different method the mature ductal cell identity of early passage PPDO with approximately 80% of the cells assigned to the pancreatic ductal cell 2 cluster (Fig. EV4B).

We observed that most endocrine signature genes were not expressed in these cultures (Dataset EV2). However, we detected

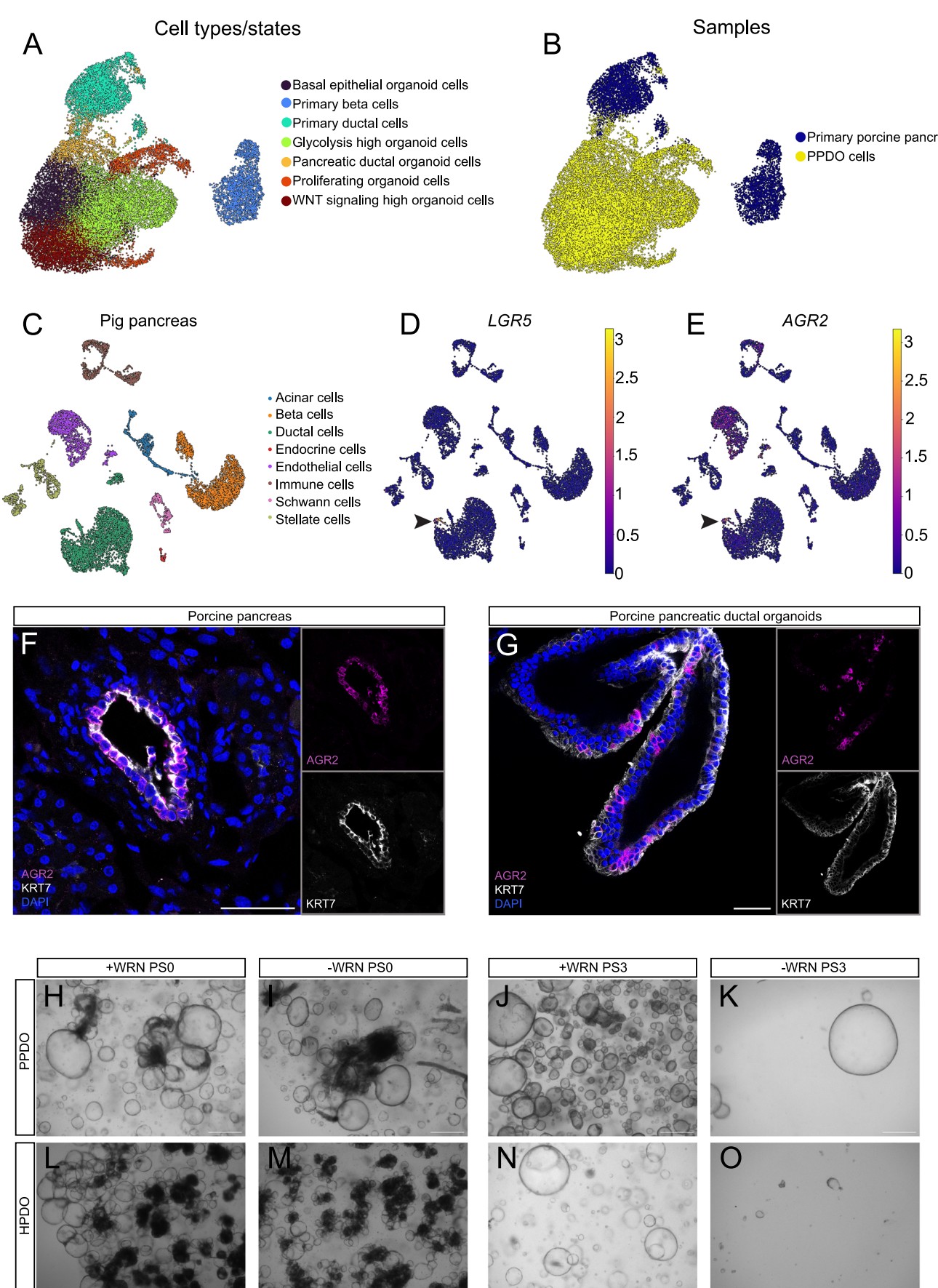

**Figure 4. Benchmarking PPDO to primary porcine pancreas.**

(A, B) UMAP representation of the integrated PPDO and primary porcine pig ductal and β-cells scRNA-Seq datasets (2 LPN pancreata) with the corresponding annotated cell types/states (A) as well as the corresponding individual samples used for integration (B). sysVI integration is shown. (C–E) UMAP representation of the pig pancreas dataset showing the corresponding cell types (E) and the expression level of the *LGR5* (F) and *AGR2* (G) genes. Arrowheads point to the *LGR5* and *AGR2* positive ductal cell population. (F) Single-plane confocal images of EPN porcine pancreas immunostained against KRT7 (gray) and AGR2 (magenta) and counterstained with DAPI. Insets show the individual channels of the merge image. Scale bar: 50 μm. (G) Single-plane confocal images of PPDO immunostained against KRT7 (gray) and AGR2 (magenta) and counterstained with DAPI. Insets show the individual channels of the merge image. Scale bar: 50 μm. (H, I) Brightfield images of PPDO cultured in medium supplemented with (H) or without (I) WNT3a/R-spondin-1/Noggin factors (WRN) 7 days following duct isolation and culture. Scale bar 500 μm. PS = Passage. (J, K) Brightfield images of PPDO cultured in medium supplemented with (J) or without (K) WNT3a/R-spondin-1/Noggin factors (WRN) after three passages at 7 days of culture post passage. Scale bar 500 μm. Experiments in (H–K) have been independently reproduced using PPDO from three different animals (1 EPN and 2 Ad). (L, M) Brightfield images of HPDO cultured in medium supplemented with (L) or without (M) WNT3a/R-spondin-1/Noggin factors (WRN) 7 days following duct isolation and culture. Scale bar 500 μm. (N, O) Brightfield images of HPDO cultured in medium supplemented with (N) or without (O) WNT3a/R-spondin-1/Noggin factors (WRN) after three passages at 7 days of culture post passage. Scale bar 500 μm. Experiments in (N, O) have been independently reproduced using 4 different HPDO preparations. Source data are available online for this figure.

minimal expression of endocrine genes in the embryonic PPDO sample, which suggested that embryonic PPDO might retain more plasticity and potency compared to adult-derived PPDO (Fig. EV4C). Differential gene expression analysis revealed that genes upregulated upon differentiation relate to a mature ductal gene signature, predominantly carbonic anhydrases and retinoic acid signaling pathway genes (Fig. 5A). Genes in the WNT pathway were among the most significantly downregulated during differentiation, suggesting that the WNT proliferative progenitor phenotype can be dynamically regulated in vitro (Fig. 5A). Using pathway analysis, we identified established molecular pathways of ductal cell identity including one-carbon (Karampelias et al, 2021) and retinoic acid (Rovira et al, 2010) metabolic processes among the most significantly upregulated transcripts. Pathways enriched in the downregulated genes included among others vascular endothelial growth factor (VEGF) related processes, glycolysis/hypoxia pathways and epithelial-to-mesenchymal transition processes (Fig. 5B,C). Therefore, the RNA-Seq based comparison of the differentiation medium suggested that removal of proliferating agents from the PPDO medium promotes a more mature epithelial/ductal fate, but a more thorough comparison is needed to further substantiate this claim.

Using the RNA-Seq results as a guide, we modified the PPDO medium to induce endocrine cell fate. We treated PPDO with a cytokine-depleted media supplemented with inhibitors for NOTCH, retinoic acid and insulin receptor signaling for 5 days. This treatment induced gene expression of endocrine genes but with a large variability between PPDO lines (Fig. 5D–G). Still, no hormone expressing cells were detectable as assessed by immunocytochemistry on the protein level. To induce protein hormone expressing cells, we dispersed PPDO to single/clumps of cells (to manually disperse them from the epithelium) and plated them in laminin-coated chambers (as laminin/integrin signaling is important for endocrine differentiation during embryogenesis (Mamidi et al, 2018; Cozzitorto et al, 2020; Jin et al, 2013)), followed by treatment with the endocrine induction medium. Of note, the insulin signaling inhibitor was omitted as it was inducing overt cell death over prolonged treatment. Under these conditions, we observed few scattered INS$^+$/GCG$^+$ (co-stained for both markers) cells in early but never in late passaged PPDO (Fig. 5H–K). Moreover, the induction on the protein level was less than 1% of total cells assessed suggesting that this protocol is not optimized for endocrine induction for therapeutic applications. Our results showcase the utility of PPDO as a platform to study ductal cell

maturation in vitro and exhibit variability in induction of endocrine cell fate with very low efficiency.

## Chemical screen for safety/toxicology assessment of FDA drugs using PPDO

Organoid cultures hold great translational potential for drug discovery/repurposing studies. We designed and performed a PPDO-based chemical screen to reveal FDA-approved drugs that could be a potential safety risk for ductal/progenitor cells by inducing proliferation (carcinogenesis) or cell death (pancreatitis). We plated PPDO in 96-well plates, cultured for 2 days for PPDO formation and treated them for 3 days with 176 chemicals from an FDA-approved library (see Methods for screening details). We used high-content imaging to identify chemicals that induced mitosis as assayed by positivity for the mitosis marker phosphorylated serine 10 of histone 3 (pHH3) (Fig. 6A). In parallel, we benchmarked the image-based method to a luciferase-based readout of cell survival coupled to ATP (Fig. EV5A). Image-based quantifications provided less variability compared to the luciferase-based readout. Our primary screen showed that removal of WRN factors from the medium reduced proliferation (Fig. 6B,C). Eight chemicals exhibited a z-score >1 suggesting that they might stimulate proliferation compared to the complete medium (Fig. 6B,C). Primary hits of our chemical screen were: flumazenil (γ-aminobutyric acid type a receptor (GABAa) receptor antagonist), cimetidine (histamine receptor H2 (HRH2) antagonist), tranylcypromine hydrochloride (lysine demethylase 1A (LSD1) inhibitor), megestrol acetate (progesterone receptor agonist), stavudine (nucleoside analog) and losartan/telmisartan/candesartan (angiotensin AT1 receptor (AGTR1) antagonist). Therefore, our primary screen provided potential candidates with proliferative induction capacity in ductal cells.

To validate the results of the primary screen, we performed a secondary screen with the most promising primary hits using three independent PPDO lines. Additionally, we included a second chemical targeting the same protein to validate the molecular pathway of the potential hits (Fig. 6D–M). Our secondary screen revealed that none of the primary hits could significantly increase proliferation of PPDO when added to complete medium (Fig. 6N). The GABAa receptor antagonists, flumazenil and picrotoxin could revert proliferation levels almost to the level of the complete medium, but with great variability between PPDO lines, and therefore the results did not reach statistical significance (Fig. 6O).

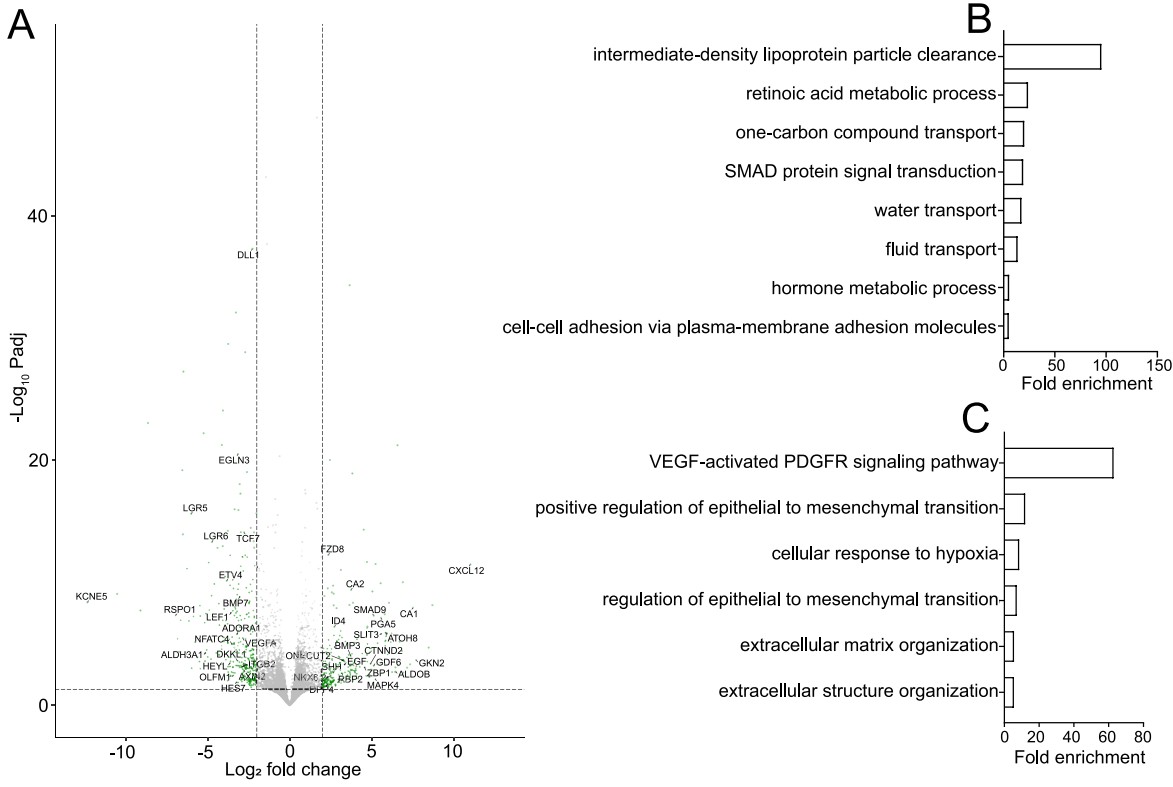

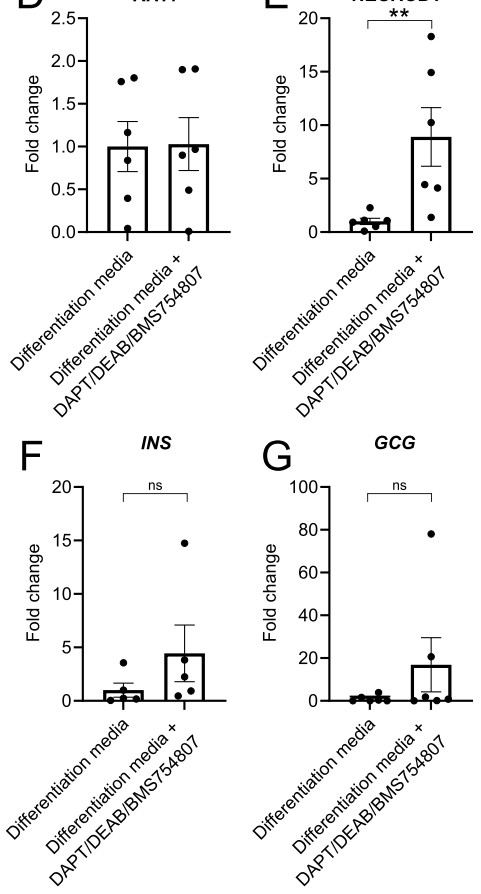

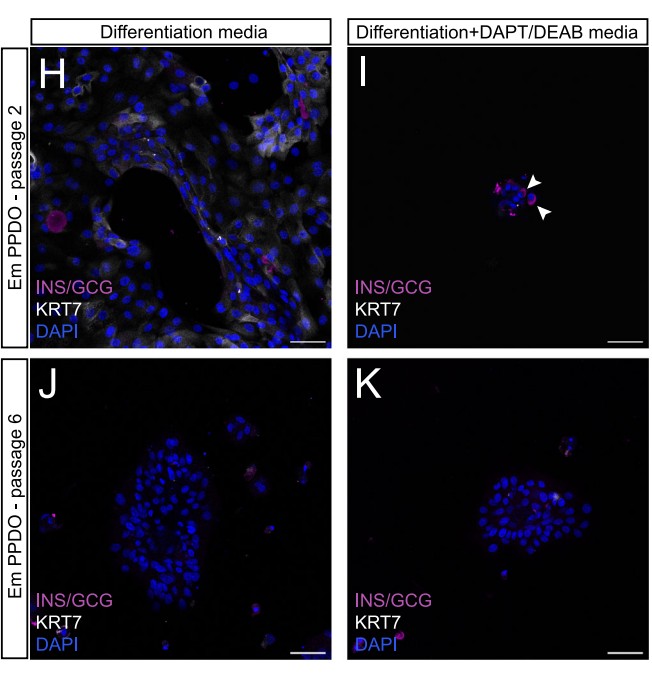

**Figure 5. Differentiation potential of PPDO to pancreatic lineages.**

(A) Volcano plot indicating the number of significantly changed genes following the 7 days of culturing in the basal differentiation medium. Genes important for developmental processes are highlighted in the graph. Bulk RNA-Seq was performed on 4 biological replicate PPDO from 1 Em, 2 EPN and 1 LPN samples. Statistical analysis was employed as described in DESEQ2 package using a Wald test with Benjamin and Hochberg adjustment for multiple testing. (B, C) Bar plots showing the fold enrichment of significantly affected biological processes for the upregulated and downregulated genes of the RNA-Seq dataset shown in (A). (B) shows selected enriched GO terms for the significantly upregulated genes and (C) shows selected GO terms for the significantly downregulated genes. (D–G) Plots showing the fold change of gene expression for *KRT7* (D), *NEUROD1* (E), *INS* (F) and *GCG* (G) genes between differentiation medium alone or supplemented with DAPT-DEAB-BMS754807. Data are shown as mean ± SEM. Mann–Whitney test was used to assess significance with **$P = 0.0043$ for *NEUROD1*, $^{ns}P = 0.1508$ for *INS* and ns $^{ns}P = 0.3095$ for *GCG*. $n = 6$ different PPDO lines were assayed (2 Em, 3 EPN, 1 Ad). (H–K) Single-plane confocal images of PPDO derived from Em pig pancreas from passage 2 (H, I) or passage 6 (J, K) cultures and treated with differentiation only (H–J) or differentiation media supplemented with the NOTCH inhibitor DAPT and the aldehyde dehydrogenase inhibitor DEAB (I–K). PPDO were immunostained against KRT7 (gray)-INS/GCG (magenta) and counterstained with DAPI. Arrowheads in I indicate the insulin positive cells. Similar staining patterns were obtained from $n = 4$ different biological replicate PPDO lines (1 Em, 2 EPN, 1 Ad). Scale bar: 50 μm. Source data are available online for this figure.

To benchmark against HPDO, we validated that the molecular targets of the primary hits were indeed expressed in the PPDO and HPDO (Fig. EV5B,C). The histamine receptor, GABA receptor and LSD1 demethylase targets but not the angiotensin receptor genes were present in HPDO, and therefore the respective chemicals were excluded from downstream experiment. Compared to the PPDO, HPDO showed great variability in the incorporation of the mitotic marker pHH3 across lines highlighting an inter-donor variability of proliferation dynamics. None of the chemicals altered significantly the proliferation percentage when added to the complete medium (Fig. 6P). None of the chemicals reverted the proliferation rate of HPDO to the control condition, unlike what we observed with the GABAa receptor antagonists in the PPDO cultures. Picrotoxin had the more pronounced effect on stimulating proliferation compared to the screening medium, but without reaching significant levels (Fig. 6Q). Lastly, we performed a dose–response test of picrotoxin treatment in HPDO. None of the concentrations significantly changed proliferation rate of HPDO cells (Fig. EV5D). Overall, our chemical screen benchmarked the use of PPDO for an FDA relevant pharmacological application and all tested drugs showed a safe profile in terms of uncontrolled proliferation induction.

## Alpha-adrenergic receptor inhibitors induce ductal cell toxicity

On the opposite spectrum, several chemicals showed aberrant toxicity of PPDO as assessed by brightfield imaging during the screening. Grouping the toxic chemicals according to their proposed mechanism of action, we observed a clear enrichment of chemicals that inhibit DNA synthesis relating processes possibly by inhibiting cell proliferation (Fig. 7A). Other prominent chemical classes inducing toxicity included adrenergic receptor inhibitors, 3-hydroxy-3-methyl-glutaryl-CoA reductase (HMG-CoA) inhibitors (statins) and estrogen receptor targeting molecules (Fig. 7A). These results were consistent between the PPDO and HPDO models (Fig. 7B–I). Next, we assessed early effects (18 h post treatment) following chemical treatments of PPDO and HPDO with etoposide (DNA synthesis inhibitor), lovastatin (HMG-CoA inhibitor) and prazosin hydrochloride (adrenergic receptor inhibitor), which were the most consistent toxic drug classed in our screen. In both PPDO and HPDO, there was a clear effect of prazosin hydrochloride treatment to induce apoptosis, with all organoids almost dying following the short treatment. Lovastatin and etoposide induced apoptosis but not to similar levels as prazosin hydrochloride treatment suggesting that prolonged treatment is needed for

observing the initial effects (Fig. 7J–S). Our screen of FDA-approved chemical compounds using ductal organoids showed a conserved, strong toxic effect of adrenergic receptor inhibitors on ductal cells highlighting the suitability of PPDO as surrogate model.

## Discussion

Organoid cultures are a valuable tool for studying adult progenitor biology and pancreatic disease. In this work, we established and performed an in-depth characterization of PPDO across distinct developmental stages of porcine pancreas development. These samples provide information on the full range of pancreas development in a relevant large animal model (pig), samples that are difficult to obtain from human donors. Our phenotyping reveals a WNT enriched signaling progenitor signature with basal cell characteristics in vitro that correlates with a porcine ductal subpopulation in vivo. No morphological, transcriptional, or protein differences were observed in PPDO from distinct developmental stages, i.e., fetal, early post-natal and adult pancreas. Contrary to HPDO that maintained a similar phenotype across cultures, PPDO transformed to dense, WNT/basal cell signaling enriched spheres upon prolonged culturing highlighting a difference between the two species. Moreover, we show that PPDO can acquire either a more mature ductal fate or differentiate/transdifferentiate to an endocrine fate upon chemical and extracellular matrix manipulations. Finally, we demonstrate the translation potential of this system for FDA-approved drug safety application by identifying novel conserved pathways for ductal/progenitor cell toxicity/proliferation between pigs and humans.

PPDO transcriptional and functional phenotyping confirms the ductal lineage of these cultures that was progressively lost over passaging in favor of a low proliferating, WNT/basal cell signaling enriched population/state. Two studies recently reported the derivation of PPDO for the study of exocrine-related pathologies corroborating the initial pancreatic ductal fate of these cultures (O'Malley et al, 2024; Angyal et al, 2024). Building on that, we report here the first comprehensive proteomics profile of PPDO across developmental stages and HPDO compared to exocrine extract to corroborate the ductal cell fate of early passage PPDO and their similarity to HPDO. This resource will be useful to the community for future benchmarking approaches of pancreatic organoids. Contrary to PPDO, HPDO cultures retained their ductal fate over passaging as evident by our scRNA-Seq datasets in line with a recent scRNA-Seq study of HPDO (Cherubini et al, 2024).

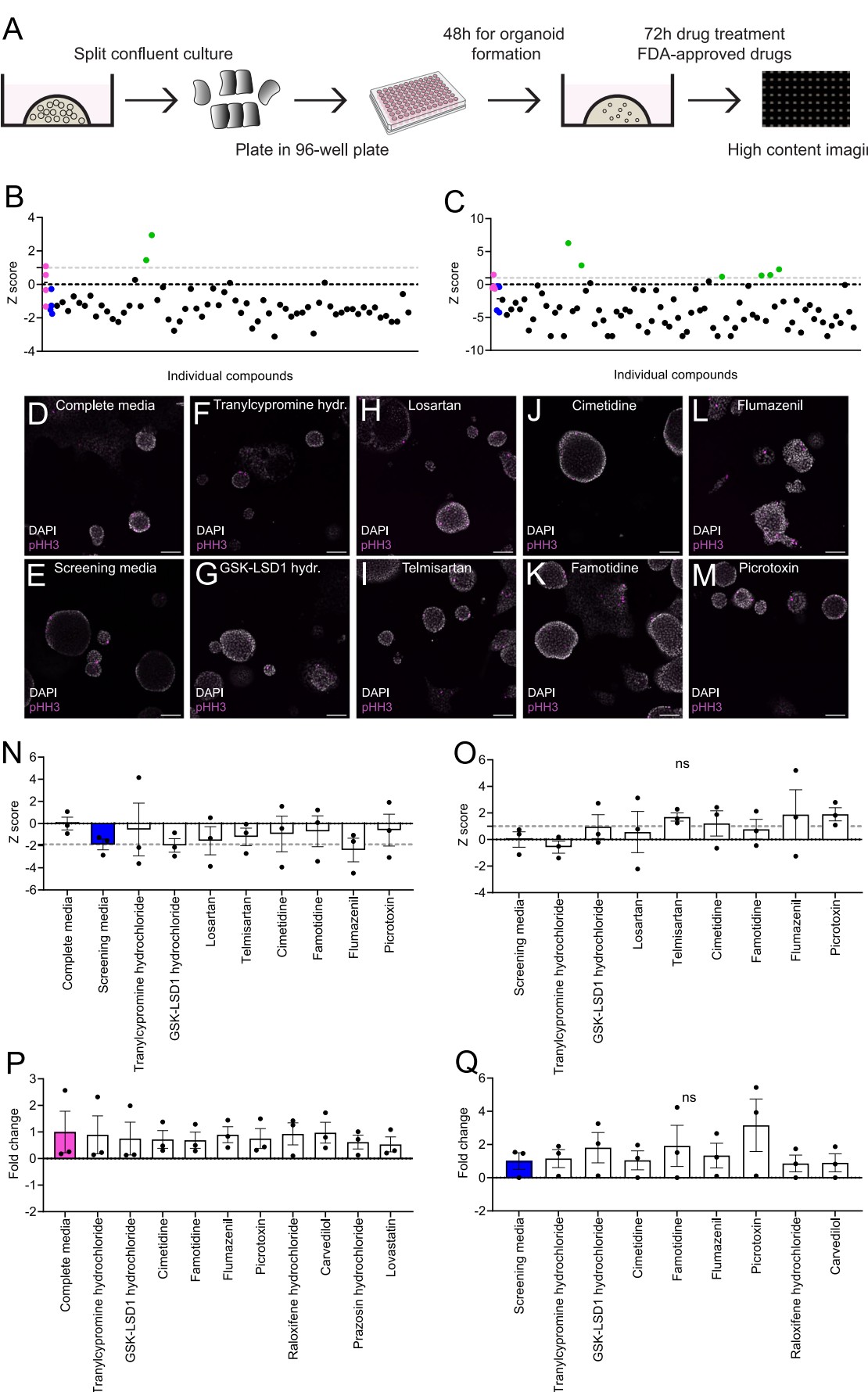

**Figure 6. Chemical screen for safety/toxicology assessment of FDA drugs using PPDO.**

(A) Schema documenting the chemical screen approach using PPDO cultures. (B, C) Scatter plots reporting the z-scores of the proliferation percentage of the primary screen results compared to the complete medium control wells for two PPDO lines, Em (B) and Ad (C). Each dot represents a single well of the 96-well plate treated with a single chemical. Magenta dots show the individual wells of PPDOs treated in the complete medium with DMSO (control) and blue dots show the individual wells of PPDOs treated in screening medium (complete medium -WRN) with DMSO and the line for these two wells corresponding to the median of the four wells/condition. Gray dotted line demarcates z-score of 1, black dotted line a z-score of 0 and green dots positive hits of primary screen. Different chemicals of the library were screened in each PPDO culture and thus the plots do not represent technical replicates. (D–M) Single-plane confocal images of PPDOs from the secondary screen cultured in complete (D) or screening media (E–M) and treated with DMSO (D, E), tranylcypromine hydrochloride (F), GSK-LSD1 hydrochloride (G), losartan (H), telmisartan (I), cimetidine (J), famotidine (K), flumazenil (L) and picrotoxin (M). Proliferating cells are marked by the marker pHH3 (magenta) and nuclei are counterstained with DAPI (gray). Scale bar 100 μm. (N, O) Bar plots reporting the z-scores of the proliferation percentage of the secondary screen. In (N), the z-score of the treatments of the chemicals added to the complete medium in reference to the complete medium are shown. In (O), the z-score of the treatments of the chemicals added to the screening medium in reference to the screening medium are shown. Gray dotted line shows the mean of the screening medium z-score for (N) and the z-score of 1 for (O). $n = 3$ individual PPDO lines (2 LPN and 1 Ad). Error bars show the ±SEM. ns = not significant as assessed by a Kruskal–Wallis statistical test, exact $P$val = 0.5011. (P, Q) Bar plots reporting the fold change of the proliferation percentage of the HPDO chemical screen. In (P), fold change in reference to the proliferation rate of the complete medium is shown. In (Q), fold change in reference to the proliferation rate of the screening medium is shown. $n = 3$ individual HPDO lines. Error bars show the ±SEM. ns = not significant as assessed by a Kruskal–Wallis statistical test, exact $P$val = 0.8310. Source data are available online for this figure.

This species difference could be explained by diverse starting culture populations or variable growth factor efficacies in the PPDO cultures. Additionally, PPDO originating from distinct porcine developmental stages do not vary morphologically, functionally, or transcriptionally, which differs to mouse intestinal organoids that show a clear morphological difference between embryonic and adult cultures (Pikkupeura et al, 2023). Importantly, removal of proliferation-related cytokines from PPDO induces a mature ductal fate, but all endocrine-inducing protocols tested were largely inefficient. This limited endocrine potential of PPDO differs from the endocrine induction capacity of early postnatal porcine islet cultures, in which ductal cells differentiate to endocrine cells with high efficiency (Zhang et al, 2021; Seeberger et al, 2023; Hassouna et al, 2018; Karampelias et al, 2022, 2021). Overall, we have performed a deep-phenotyping of PPDO according to recent suggestions for benchmarking organoid cultures (Marsee et al, 2021) and we argue that a cross-species comparisons in terms of molecular profile and differentiation capacity/conditions is needed to provide a holistic overview of these models (Xu et al, 2025).

We generated PPDO across developmental stages with the aim to temporally resolve and study adult pancreatic progenitor biology, a much-debated topic in pancreas regeneration field (Goode et al, 2023; Karampelias et al, 2025). Since expansion of pancreatic progenitors is a much-sought advancement that the field is working for mass endocrine cell production (Jarc et al, 2023; Gonçalves et al, 2021; Ma et al, 2022), PPDO can serve as a model to study progenitor dynamics. Our results indicate a WNT enriched transcriptional signature present in both PPDO and primary porcine ductal cells. Prolonged passaging of PPDO lead to enrichment of the WNT population with basal cell characteristics (KRT5$^+$) as shown by the scRNA-Seq and proteomics datasets. A rare ductal population with basal cell characteristics has been recently identified to be important for pancreas homeostasis and carcinogenesis (Martens et al, 2022; Coolens et al, 2024) and PPDO provides a great model to study this population in vitro. Two studies proposed that a WNT responsive population characterized by LGR5 expression could be surfacing under injury/high metabolic demand conditions (Huch et al, 2013; Rodriguez et al, 2021). A recent scRNA-Seq study of the adult mouse ductal epithelium clearly showed a WNT responsive transcriptional signature in the healthy mouse pancreas (Fernández et al, 2024), a population that, if present, was not highlighted in another mouse ductal scRNA-seq

dataset (Hendley et al, 2021). No definitive WNT enriched population was shown/discussed in recent transcriptional profiles of human embryonic pancreas (Ma et al, 2023; Migliorini et al, 2024) or in a scRNA-Seq of adult human ductal cells (Qadir et al, 2020) but that could be attributed to the enrichment strategy chosen in the latter study. A recent finding suggests that LGR5$^+$ marks a multipotent progenitor population in the human embryonic pancreas, but the prevalence of this population in the scRNA-Seq datasets of human embryonic progenitors is not clear (Andersson-Rolf et al, 2024). Nevertheless, these human embryonic organoids differentiated to all pancreatic epithelial lineages, similar to observations from mouse embryonic explant cultures (Andersson-Rolf et al, 2024; Bonfanti et al, 2015). Moreover, our scRNA-Seq and proteomics of HPDO did not show such pronounced WNT transcriptional signature but rather progenitors marked by *GP2* expression, a multipotent embryonic progenitor marker in humans, highlighting another important difference to the PPDO (Ramond et al, 2017; Cogger et al, 2017; Ameri et al, 2017). Still, this population expressed hallmarks of WNT responsive cells including olfactomedin 4 (*OLFM4*) and *LGR5* (van der Flier et al, 2009; Muñoz et al, 2012). Our data hint towards a WNT signaling-dependent adult pancreatic progenitor population with organoid forming capacities, unipotent ductal progenitor potential and limited endocrine differentiation potential.

Chemical screens using organoids as a platform hold great promise for identifying new therapeutic leads or assessing the safety profile of already commercialized drugs. We utilized our PPDO system and performed a chemical, high-content imaging-based screen to assess the safety of FDA-approved chemicals for pancreatic ductal/progenitor cells. Given the need for non-rodent animal model research for regulatory approval of drugs, our PPDO platform can be utilized to screen for pancreatitis or pancreatic cancer drug applications. The addition of technical replicates could further improve the robustness of such image-based approaches, given the observed variability between biological replicates especially in the HPDO. Our approach implicates adrenergic receptor inhibitors and HMG-CoA inhibitors in ductal/progenitor cell toxicity. While neurotransmitters are implicated in pancreatic cancer biology, no extensive studies have been performed using the α adrenergic receptor inhibitor (Schuller and Al-Wadei, 2010). Angiotensin-converting enzyme inhibitors (ACEi) have been involved in drug-induced pancreatitis previously, suggesting that

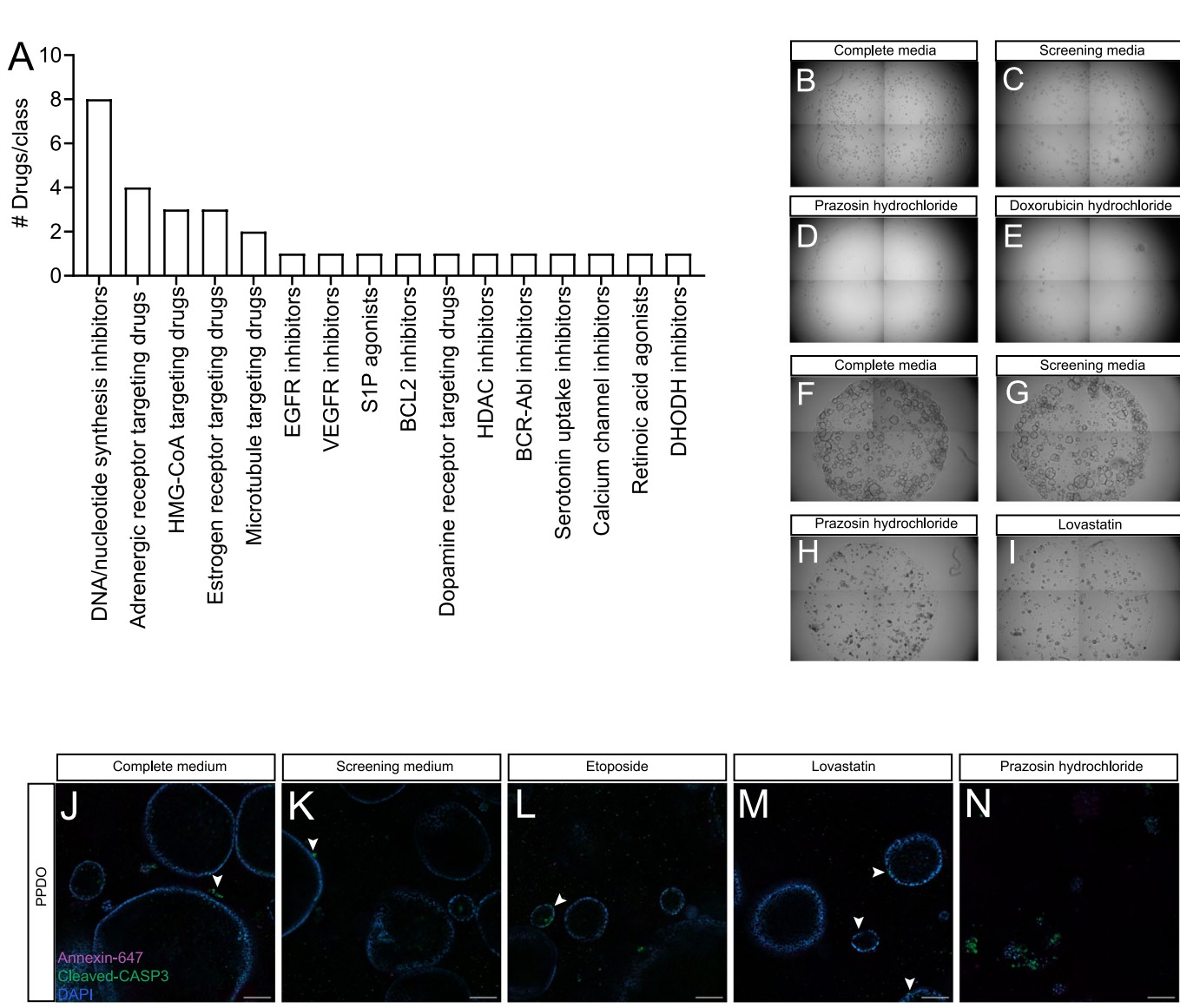

**Figure 7. Alpha-adrenergic receptor inhibitors induce ductal cell toxicity.**

(A) Bar plot showing the summary of the number of chemicals/chemical class that induced toxicity in the primary screen. (B–E) Brightfield images of the whole well of a 96-well plate picturing PPDO in complete medium (B), screening medium (C) and PPDOs treated with prazosin hydrochloride (D) and doxorubicin hydrochloride (E), two drugs that induced cell death. Four individual images were stitched together to get the overview of the well. (F–I) Brightfield images of the whole well of a 96-well plate picturing HPDO in complete medium (F), screening medium (G) and HPDOs treated with prazosin hydrochloride (H) and lovastatin (I), two drugs that induced cell death. Four individual images were stitched together to get the overview of the well. (J–N) Single-plane confocal images of PPDOs cultured in complete (J) or screening medium (K–N) and treated with DMSO (J–K), etoposide (L), lovastatin (M), prazosin hydrochloride (N). Apoptotic cells are marked by the cleaved-CASP3 (green) and annexin V (magenta) and nuclei are counterstained with DAPI (blue). Arrowheads point to the cleaved-CASP3 positive cells in the organoids. Scale bar 100 μm. Results were similar between $n = 3$ biological replicates (1 Em, 1 EPN, 1 Ad). (O–S) Single-plane confocal images of HPDO cultured in complete (O) or screening medium (P–S) and treated with DMSO (O–P), etoposide (Q), lovastatin (R), prazosin hydrochloride (S) Apoptotic cells are marked by the cleaved-CASP3 (green) and annexin V (magenta), proliferating cells by Ki67-570 and nuclei are counterstained with DAPI (blue). Scale bar 100 μm. Results were similar between $n = 3$ biological replicates.

this pathway can be involved in pancreas damage (Rouette et al, 2022; Twohig et al, 2021). On the contrary, there is a wealth of clinical data regarding potential roles of statins in exocrine disease such as pancreatitis and pancreatic cancer (Ba et al, 2023; Badalov et al, 2007; Gong et al, 2017), and our study points to ductal cells as a target for this drug class. Whether statins can induce toxicity in patient-derived pancreatic ductal adenocarcinoma organoids remains to be seen. Moreover, our screening approach highlights the absence of significant proliferative phenotypes for FDA-approved drugs. Of note, GABA treatment itself has been previously involved in pancreatic β-cell proliferation and in contradictory studies regarding α- to β-cell fate conversion (Li et al, 2016; Purwana et al, 2014; Ackermann et al, 2018; Ben-Othman et al, 2016; Espes et al, 2021; Shin et al, 2018; Martin et al, 2022). The absence of proliferative phenotypes is of importance for pancreatic cancer safety concerns in the presence of pre-cancerous lesions and highlights the use of the easily obtainable PPDO for this type of studies.

In summary, we performed an in-depth benchmarking of PPDOs across developmental stages and used them as a platform for high-content chemical screen. The wider accessibility of pig pancreas tissue across development compared to human and the validation of PPDO as a screening platform with comparable results to HPDO makes it an attractive model for the study of mammalian ductal/progenitor biology at scale and for pancreatitis/pancreatic cancer applications. Meanwhile, we acknowledge that biological results using pancreatic organoids can vary greatly depending on the (i) starting culture material (primary vs stem cell derived), (ii) developmental stage (embryo vs adult), (iii) model organism (mouse vs pig vs human) and (iv) culture conditions (medium composition/passaging). Thus, clear reporting of such parameters will be necessary to clarify the biologically relevant results. Better characterized pancreatic organoid systems can pave the way for a holistic understanding of pancreatic biology and disease, and generation of new therapy approaches.

# Methods

### Reagents and tools table

| Reagent/Resource | Reference or Source | Identifier or Catalog Number |
|---|---|---|
| **Experimental models** | | |
| Primary porcine ductal organoids | N/A | N/A |
| Primary human ductal organoids | N/A | N/A |
| Wild-type Landrace Pigs | N/A | N/A |
| INS-eGFP Landrace Pigs | N/A | N/A |
| **Antibodies** | | |
| Rabbit a-KRT5 | Abcam | ab53121 |
| Rabbit a-SOX9 | Millipore | AB3555 |
| Mouse a-KRT7 | Agilent Technologies | M701829-2 |
| Guinea pig a-KRT8/18 | Origene | BP5007 |
| Rabbit a-PAN-KRT | Agilent Technologies | Z0622 |

| Reagent/Resource | Reference or Source | Identifier or Catalog Number |
|---|---|---|
| Mouse a-BMPR1A | LSBio | LS-C191759 |
| Mouse a-CDH1 | BD Biosciences | 610181 |
| Rat a-CDH1 | Takara | M108 |
| Rabbit a-CFTR | Cell Signaling | 78335 |
| Mouse a-GP2 | MBL Life Science | D277-3 |
| Goat a-PDX1 | R&D Systems | AF2419 |
| Rabbit a-AGR2 | Cell Signaling | 13062S |
| Mouse a-GCG | Sigma-Aldrich | G2654 |
| Mouse a-NKX6-1 | Developmental Studies Hybridoma Bank | F55A10 |
| Sheep a-NEUROG3 | R&D Systems | AF3444 |
| Rabbit a-phospho (Ser10) histone 3 | Cell Signaling | 3377S |
| Rabbit a-pancreatic amylase | Abcam | 21156 |
| Guinea pig a-INS | LSBio | LS-C85862-1 |
| Guinea pig a-GCG | Takara | M182 |
| Rabbit a-cleaved-caspase3-ASP175 | Cell Signaling | 9661S |
| Phalloidin Alexa Fluor 546 | Invitrogen | A22283 |
| a-Ki670 Vio G570 | Miltenyi Biotec | 130-133-796 |
| Streptavidin Alexa Fluor 647 | Life Technologies | S21374 |
| Multiple Alexa Fluor conjugated secondary antibodies | ThermoFischer Scientific | N/A |
| **Oligonucleotides and other sequence-based reagents** | | |
| *INS* TaqMan probe | ThermoFischer Scientific | Ss03386682_u1 |
| *GCG* TaqMan probe | ThermoFischer Scientific | Ss03384069_u1 |
| *NEUROD1* TaqMan probe | ThermoFischer Scientific | Ss03373333_s1 |
| *SST* TaqMan probe | ThermoFischer Scientific | Ss03391856_m1 |
| *ACTB* TaqMan probe | ThermoFischer Scientific | Ss03376563_Uh |
| *KRT7* TaqMan probe | ThermoFischer Scientific | Ss06890421_Mh |
| *CFTR* TaqMan probe | ThermoFischer Scientific | Ss03389420_m1 |
| **Chemicals, enzymes and other reagents** | | |
| Collagenase V | Merck-Millipore | C9263 |
| HBSS | Gibco | 24020-091 |
| TrypLE Express 1X | Gibco | 12605-010 |
| Advanced DMEM/F12 | Life Technologies | 12634-010 |
| Penicillin-Streptomycin | Gibco | 15140163 |
| Primocin | InvivoGen | ant-pm-2 |
| GlutaMAX | ThermoFischer Scientific | 35050038 |
| N-2 supplement | ThermoFischer Scientific | 17502048 |

| Reagent/Resource | Reference or Source | Identifier or Catalog Number |
|---|---|---|
| B-27 supplement – Vitamin A | ThermoFischer Scientific | 12587010 |
| A83-01 | Tocris Bioscience | 2939 |
| Prostaglandin E2 | Tocris Bioscience | 2296 |
| rhWNT3a | Tocris Bioscience | 5036-GMP-010 |
| hR-Spondin 1 | PeproTech | 120-38-100 |
| hNoggin | PeproTech | 120-10C-100 |
| hFGF10 | PeproTech | 100-26-25 |
| hEGF | PeproTech | AF-100-15-100 |
| [Leu15] human gastrin I | Sigma-Aldrich | G9145-1MG |
| Y-27632 dihydrochloride | Sigma-Aldrich | Y0503-1MG |
| Nicotinamide | Sigma-Aldrich | N0636-100G |
| N-acetyl-L-cysteine | Sigma-Aldrich | A9165-5G |
| 3dGRO L-WRN | Sigma-Aldrich | SCM105 |
| Cultrex RGF basement membrane extract Type 2 | Bio-Techne | 3533-010-02 |
| Porcine serum | Sigma-Aldrich | P9783-500ML |
| DAPT | Tocris Bioscience | 2634 |
| 4-Diethylaminobenzaldehyde | Sigma-Aldrich | D86256 |
| BMS-536924 | Sigma-Aldrich | TA9947273392 |
| Mouse laminin | Sigma-Aldrich | CLS354232-1EA |
| MCDB131 | Gibco | 10372019 |
| BSA fraction V, fatty acid free | Roche | 232-936-2 |
| Glucose | Sigma-Aldrich | |
| ITS-X | Gibco | 51500-056 |
| ALK5 inhibitor II | Enzo Life Sciences | ALX-270-445 |
| Heparin | Sigma-Aldrich | H3149-10KU |
| SANT-1 | Sigma-Aldrich | S4572-5mg |
| Retinoic acid | Sigma-Aldrich | |
| Gamma secretase inhibitor XXI | Merck-Millipore | 565790-1MG |
| T3 | Sigma-Aldrich | T6397 |
| Retinoic acid | Sigma-Aldrich | R2625 |
| Betacellulin | Bio-Techne | 261-CE-250/CF |
| Vitamin C | Sigma-Aldrich | A4544 |
| D-(+)- Glucose | Sigma-Aldrich | G7528 |
| DMSO | Sigma-Aldrich | D5879-100ML |
| BSA Fraction V, Fatty Acid Free, for cell culture media | Roche | 10775835001 |
| MEM nonessential Amino Acid Solution | Corning | 15333581 |
| Zinc Sulfate Heptahydrate | Sigma-Aldrich | Z0251 |
| Trace Elements A, 1000x solution | Corning | 15333641 |
| Trace Elements B, 1000x solution | Corning | 15343641 |
| Sodium bicarbonate | Carl Roth | 8551.1 |
| Albumin from pig serum | Sigma-Aldrich | A1830-1G |

| Reagent/Resource | Reference or Source | Identifier or Catalog Number |
|---|---|---|
| IWR-1-endo | Merck-Millipore | 681669-10 mg |
| Cultrex organoid harvesting solution | R&D Systems | 3700-100-01 |
| Annexin V, Alexa FluorTM 647 conjugate-500 uL | Life Technologies | A23204 |
| Bovine serum albumin | Sigma-Aldrich | A3311-100G |
| DBA-biotin | Vector Laboratories | B-1035-5 |
| Chromium Single Cell 3' Reagent Kits v3.1 | 10X Genomics | 1000268 |
| Arcturus PicoPure RNA isolation kit | Applied Biosystems | KIT0204 |
| Illumina Stranded mRNA Prep | Illumina | 20040532 |
| 7-AAD | ThermoFischer Scientific | A1310 |
| DAPI | ThermoFischer Scientific | D1306 |
| SuperScript VILO cDNA synthesis kit | ThermoFischer Scientific | 11754050 |
| TaqMan fast advanced Master Mix | ThermoFischer Scientific | 4444558 |
| Tocriscreen FDA-approved drugs library | Tocris Bioscience | 7200 |
| Triton X-100 | Sigma-Aldrich | T8787 |
| Tween 20 | ThermoFischer Scientific | 85113 |
| Losartan potassium | Sigma-Aldrich | PHR1602-1G |
| Telmisartan | Santa Cruz Biotechnology | sc-204907 |
| Cimetidine | Sigma-Aldrich | C4522-5G |
| Famotidine | Sigma-Aldrich | F6889-500MG |
| Tranylcypromine hydrochloride | Cayman chemicals | Cay10010494-250 |
| GSK-LSD1 hydrochloride | Tocris Bioscience | 5361/10 |
| Picrotoxin | Sigma-Aldrich | P1675 |
| Flumazenil | Cayman chemicals | Cay14252-10 |
| CellTiter-Glo luminescent cell viability assay | Promega | G7570 |
| Vectashield | Vector Laboratories | H-1000-10 |
| Forskolin | Tocris Bioscience | 1099 |
| Human secretin | CliniSciences | hor-273-25mg |
| **Software** | | |
| Fiji (ImageJ) | https://imagej.net/software/fiji/ | N/A |
| GraphPad Prism 10 | https://www.graphpad.com/ | N/A |
| Cellpose 2.0 | https://www.cellpose.org/ | N/A |
| R studio (v2024.04.2) | https://posit.co/products/open-source/rstudio/?sid=1 | N/A |
| R v4.3.3 | https://www.r-project.org/ | N/A |

| Reagent/Resource | Reference or Source | Identifier or Catalog Number |
|---|---|---|
| Spectronaut v18&20 | Biognosys | |
| Adobe Illustrator | Adobe | N/A |
| CellRanger v7.1.0 | 10x Genomics | |
| Microsoft Excel | Microsoft | |
| All R and Python-based packages for analysis are detailed in the Methods section with proper citation to original source | N/A | N/A |
| **Other** | | |
| Leica TCS SP5 confocal microscope | Leica Microsystems | N/A |
| Stellaris confocal microscope | Leica Microsystems | N/A |
| Zeiss LSM 880 | Zeiss | N/A |
| Cell Discoverer 7 | Zeiss | N/A |
| EVOS M5000 | ThermoFischer Scientific | N/A |
| Zeiss Observer widefield microscope | Zeiss | N/A |
| FACSAria III | BD Biosciences | N/A |
| Q Exactive HF mass spectrometer | ThermoFischer Scientific | N/A |
| timsTOF HT mass spectrometer | Bruker | N/A |
| NovaSeq6000 | Illumina | N/A |
| NovaSeqX+ | Illumina | N/A |
| ViiA 7 Real-Time PCR system | ThermoFischer Scientific | N/A |

## Porcine pancreatic ductal organoid culture

Wild-type and *INS*-eGFP pigs (German landrace background) were housed and bred at the state-of-the-art, pathogen-free pig facility of LMU Munich (Center for Innovative Medical Models; www.lmu.de/cimm/). *INS*-eGFP pigs exhibit beta-cell specific eGFP transgene expression under the control of the porcine insulin promoter (Kemter et al, 2017). The animals were generally kept in groups, with the exception of boars. Up to an age of 28 days, piglets were housed in a farrowing pen together with the mother, and a heated nest was offered to the piglets. Pigs were fed a commercial complete feed (Vilstalmühle and Deuka) twice daily, and had free access to water ad libitum from nipple drinkers, bowl drinkers, and trough flushers. Computer-assisted ventilation system ensured proper air exchange, humidity and temperature control. For enrichment, straw/hay, balls, play chains, ropes made of a sisal-cotton mix, wooden or lucerne sticks, and bite hedgehogs are alternately provided. The animals always have access to organic, deformable material (straw/hay, wood, chew ropes, hay cobs, rooting soil). Regular health monitoring was performed according to FELASA guidelines. Work described in this paper was performed under the Permission No. 55.2-2532.Vet_02-17-136 and 55.2-2532.Vet_02-19-195 approved by the licensing committee from the responsible authority (Government of Upper Bavaria). All experiments were conducted

according to the German Animal Welfare Act and Directive 2010/63/EU on the protection of animals used for scientific purposes. In this work, we defined the developmental stages of the pigs that the pancreas was dissected from as: embryonic (Em) referring to embryos before birth, early postnatal (EPN) referring to pigs after birth and immediately before/after weaning (postnatal days 1–50), late postnatal (LPN) referring to adolescent pigs after weaning and before reproductive maturity (postnatal day 51–300) and adult (Ad) pigs (postnatal day 300 and onwards).

To establish PPDO, the isolated pig pancreas was dissected, cut into small pieces and digested with 1 mg/ml collagenase V (diluted with HBSS 1X buffer) for 20' at 37 °C. Digestion reaction was stopped with 1% BSA containing buffer in PBS 1X and ductal structures were manually picked under the microscope. Ductal structures were washed five times with what we refer to as basal medium constituted of Advanced DMEM/F12 (Life Technologies) medium supplemented with Pen/Strep (1X), primocin (100 μg/ml), GlutaMAX (1X) and HEPES (100 mM). PPDO were maintained with the previously published complete medium for HPDO (Broutier et al, 2016) containing the basal medium supplemented with: N2 supplement (1X - ThermoFischer Scientific), B27-vitamin A supplement (1X - Life Technologies), A83-01 (5 μM - Tocris Biosciences), prostaglandin E2 (3 μM), rhWNT3a (60 ng/μl), human r-spondin1 (250 ng/ml), noggin (25 ng/ml), FGF10 (100 ng/ml), human EGF (50 ng/ml), human gastrin I (10 nM), Y-27632 dihydrochloride (10 μM), nicotinamide (10 mM - Sigma-Aldrich) and N-acetyl-L-cysteine (1 mM - Sigma-Aldrich). For maintaining the PPDO, rhWNT3a, human r-spondin1 and noggin were substituted with the 3dGRO L-WRN conditioned medium supplement (Sigma-Aldrich - SCM105). Cells were plated using the Cultrex RGF basement membrane extract Type 2 (Bio-Techne) as the matrix substance diluted 1:3 ratio with what we refer to as complete medium. Differentiation medium that was used for differentiation of organoids consisted of basal medium supplemented with N2 supplement (1X - ThermoFischer Scientific), B27-vitamin A supplement (1X - Life Technologies), A83-01 (5 μM - Tocris Biosciences), Y-27632 dihydrochloride (10 μM), nicotinamide (10 mM - Sigma-Aldrich) and N-acetyl-L-cysteine (1 mM - Sigma-Aldrich). Screening medium used for the chemical screening experiments constituted of basal medium supplemented with: N2 supplement (1X - ThermoFischer Scientific), B27-vitamin A supplement (1X - Life Technologies), A83-01 (5 μM - Tocris Biosciences), prostaglandin E2 (3 μM), FGF10 (100 ng/ml), human EGF (50 ng/ml), human gastrin I (10 nM), Y-27632 dihydrochloride (10 μM), nicotinamide (10 mM - Sigma-Aldrich) and N-acetyl-L-cysteine (1 mM - Sigma-Aldrich). PPDO and HPDO were routinely checked for mycoplasma contamination and all tests showed no mycoplasma contamination.

Subculturing PPDO and HPDO was performed 7–10 days following previous splitting. Briefly, the Cultrex domes were dissociated with cold basal medium and organoids, washed 1X with basal medium and organoids were initially broken down manually with the P200 pipette. Following manual dissociation, organoids were incubated with TrypLE express (GIBCO) for 4 min at 37 °C to further break the structures into smaller clamps and single cells and organoids were plated at the desired density with fresh Cultrex and medium. All brightfield images across the manuscript were acquired with an EVOS M5000 cell imaging

system with the 4X objective in brightfield mode, unless otherwise stated.

For differentiation test of PPDO, three different medium compositions were tested: (1) Basal organoid medium supplemented with 20% pig serum (Sigma-Aldrich). (2) Basal organoid medium supplemented with N2 supplement (1X - ThermoFischer Scientific), B27-vitamin A supplement (1X - Life Technologies), A83-01 (5 µM - Tocris Bioscience), Y-27632 dihydrochloride (10 µM), nicotinamide (10 mM - Sigma-Aldrich) and N-acetyl-L-cysteine (1 mM - Sigma-Aldrich) further supplemented with DAPT (1 µM - Tocris Bioscience), 4-Diethylaminobenzaldehyde (DEAB) (10 µM - Sigma-Aldrich) with/without BMS-536924 (5 µM - Sigma-Aldrich). For this medium combination, treatments performed either in Cultrex cultured organoids, or in dispersed cells cultured in mouse laminin (Corning laminin Sigma-Aldrich) coated ibidi chambers at 270 µg/ml final concentration. (3) S5 (Basal media: MCDB131 (Gibco) supplemented with: 1X Glutamax (Gibco), 2% BSA fraction V, fatty acid free (Roche), 35 mmol/L NaHCO$_3$ (Carl Roth), 25.5 mmol/L Glucose (Sigma-Aldrich), 0.5X ITS-X (Gibco), 100 U/ml Penicillin/Streptomycin (Gibco) + Supplements: 0.25 mM Vitamin C (Sigma-Aldrich), 0.01 g/L Heparin (Sigma-Aldrich), T3 1 µmol/L (Sigma-Aldrich), 10 µmol/L Alk5i (Enzo Life Sciences), 0.25 µmol/L SANT-1 (Sigma-Aldrich), 0.1 µmol/L retinoic acid (Sigma-Aldrich), 20 ng/ml BTC (Novus Biologicals) and 10 µmol/L GSiXXl (Millipore))and S6 media (Basal media: MCDB131 (Gibco) supplemented with: 1X Glutamax (Gibco), 2% BSA fraction V, fatty acid free (Roche), 14 mmol/L NaHCO$_3$ (Carl Roth), 8 mmol/L Glucose (Sigma-Aldrich), 0.5X ITS-X (Gibco), 100 U/ml Penicillin/Streptomycin (Gibco), 0.01 g/L Heparin (Sigma-Aldrich), 1X Trace elements A (Corning), 1X Trace elements B (Corning), 1 µmol/L ZnSO$_4$ (Sigma-Aldrich), 1X MEM Nonessential Amino Acid solution (Corning)) supplemented in PPDOs as described previously (Velazco-Cruz et al, 2019). 4-Differentiation media as described in Loomans et al (Loomans et al, 2018) with the basal organoid media supplemented with 1X Insulin-Transferrin-Selenite-X (Gibco), 10 mM nicotinamide, 2 g/l pig albumin (different than original protocol that used human albumin). For the WNT inhibition experiments, complete PPDO media without WNT3a and r-spondin1 was supplemented with IWR-1-endo (Sigma-Aldrich) WNT inhibitor at the concentrations indicated for 7 days directly after splitting.

## Human pancreatic ductal organoid culture

Exocrine pancreatic extracts were obtained following human islet isolation process from the Alberta Islet Distribution Program and the work was conducted under approval of the Health Research Ethics Board at the University of Alberta (Pro-00001620). Work with primary human tissue and HPDO generation and processing was performed under the ethical permit 2022-637-S-KH provided by the TUM ethics committee. Informed consent was obtained from all donors and the experiments conformed to the principles set out in the WMA Declaration of Helsinki and the Department of Health and Human Services Belmont Report. All HPDO were derived from adult donors and all information can be found on Table EV1. HPDO were generated either by handpicking ducts when available (death due to acinar enzyme tissue degradation) or by growing the single cell

suspension from the byproduct of human islet isolation methods enriched in exocrine tissue (Kin et al, 2024). HPDO were passaged at least three times before any experimentation. Complete, screening, differentiation and treatment media as well as passaging methodology used were identical to the PPDO cultures stated above.

## Immunostaining of PPDO and HPDO

PPDO or HPDO were released from Cultrex and washed once with cold PBS 1X followed by incubation with the Cultrex organoid harvesting solution (Bio-Techne) for 30 min. Organoids were fixed with 4% PFA for 45 min at 4 °C, followed by incubation with PBS 1X supplemented with 0.1% Tween20 (Sigma-Aldrich). Cells were blocked in a solution of 0.2% w/v BSA (Sigma-Aldrich), 0.1% Triton X-100 (Sigma-Aldrich) in PBS1X for 1 h at 4 °C. Primary antibodies were incubated overnight in blocking solution at 4 °C. Organoids were washed 2X for at least 1 h each in blocking solution and secondary antibodies were added overnight at 4 °C together with DAPI to counterstain the nuclei. Following two washes for at least 1 h each at 4 °C to remove unspecific binding of secondary antibodies, organoids were mounted with elvanol in 8-well chambers and imaged using a Leica SP5 or a Zeiss 880 laser scanning confocal microscope. Primary antibodies used in this study were: a-SOX9 (rabbit, 1:800, Millipore AB3555), a-KRT5 (rabbit, 1:200, abcam ab53121), a-KRT7 (mouse, 1:200, Agilent Technologies M701829-2), a-KRT8/18 (guinea pig, 1:1000, Origene BP5007), a-PAN-KRT (rabbit, 1:500, Agilent Z0622), a-BMPR1A (mouse, 1:400, LSBio LS-C191759), a-CDH1 (mouse, 1:800, BD Biosciences 610181), a-CDH1 (rat, 1:500, Takara M108), a-CFTR (rabbit, 1:100, Cell Signaling 78335), a-GP2 (mouse, 1:100, MBL D277-3), a-PDX1 (goat, 1:200, R&D Systems AF2419), a-GCG (mouse, 1:200, Sigma-Aldrich G2654), a-AGR2 (rabbit, 1:200, Cell Signaling 13062S), a-GCG (guinea pig, 1:800, Takara M182), a-NKX6-1 (mouse, 1:100, Developmental Studies Hybridoma Bank F55A10), a-NEUROG3 (sheep, R&D Systems AF3444), a-Pancreatic amylase (rabbit, 1:200, abcam 21156), a-INS (guinea pig, 1:200, LSBio (BIOZOL), LS-C85862-1), a-GCG (guinea pig, 1:1000, Takara, M182), Phalloidin Alexa Fluor 546 (Invitrogen A22283), a-cleaved-caspase3-ASP175 (1:200, Cell Signaling, 9661S), a-Ki67-Vio G570 (1:100, Miltenyi Biotec, 130-133-796). For Annexin staining, we added 1:100 dilution of Annexin V, Alexa FluorTM 647 (Thermo Fisher Scientific) to live organoid cultures for 30 min before fixation. Uptake was observed live before fixation.

## Immunostaining of porcine pancreas

Cryosections (12 µm) of porcine pancreas were rehydrated in PBS 1X followed by permeabilization (0.3% Triton X-100), blocking (BSA-donkey serum- in PBS 1X) and overnight incubation with primary antibodies at 4 °C. After washing primary antibodies, tissue was incubated with secondary antibodies plus DAPI to counterstain the nucleus for 2 h at room temperature followed by washes and mounting with elvanol and 1.5 coverglass thickness. Images were obtained with a TCS SP5 confocal microscope. Primary antibodies against KRT7 (mouse, 1:200, Agilent Technologies M701829-2) and AGR2 (rabbit, 1:200, Cell Signaling 13062S) were used.

## CFTR assay

PPDOs and HPDOs were plated following standard splitting procedure mentioned above in 8-well glass-bottom chambers (Ibidi). Following four to five days of culture, PPDOs were incubated with forskolin (10 μM final concentration), secretin (1 μg/ml final concentration) and a DMSO control and were imaged live using a Zeiss Observer microscope using brightfield lamp. HPDOs were incubated with forskolin and DMSO as control and imaged live using a Stellaris confocal microscope. Images from five independent positions/treatment were captured every 15 min over the span of 4 h. Lumen area was segmented with a custom-made model using the cellpose 2.0 algorithm (Stringer et al, 2021) and ROI area was calculated with Fiji (Schindelin et al, 2012) for the first and last images of the acquisition.

## Proteomics sample preparation and processing

Both for PPDO from different developmental stages and for HPDO samples, 2 confluent wells of a 24-well plate were collected and washed once with PBS 1X. Organoid Cultrex domes were dissociated for 30 min at 4 °C Cultrex organoid harvesting solution (Bio-Techne) and organoids were dispersed into single cell suspension using TrypLE Express (Gibco) at 37 °C for 8–12 min with pipetting every 4 min. The single cell suspension was stained with DAPI for 5 min at room temperature to label dead cells. 100,000–150,000 live cells were sorted with a FACSAria III (BD Biosciences) in protein LoBind tubes (Eppendorf). Similarly, human exocrine extracts were washed twice with PBS 1X, dispersed into single cell suspension with TrypLE and alive cells were sorted on a FACSAria III instrument (BD Biosciences). All sorted cells were washed five times with sterile PBS 1X and pellets were snap-frozen in liquid nitrogen and stored at −80 °C until processing.

Cell pellets were tryptic digested using the PreOmics iST Kit (Preomics GmbH, Martinsried, Germany) according to the manufacturer's specifications. For the human samples, the mass-spec (MS) data were acquired in DIA mode on a Q Exactive HF mass spectrometer (Thermo Fisher Scientific Inc., Waltham, MA, USA). Equal amounts of peptides were automatically loaded to the online coupled RSLC (Ultimate 3000, Thermo Fisher Scientific Inc.) HPLC system. A Nano-Trap column was used (300-μm inner diameter (ID) × 5 mm, packed with Acclaim PepMap100 C18, 5 μm, 100 Å from LC Packings, Sunnyvale, CA, USA, before separation by reversed-phase chromatography (Acquity UPLC M-Class HSS T3 Column 75 μm ID × 250 mm, 1.8 μm from Waters, Eschborn, Germany) at 40 °C. Peptides were eluted from the column at 250 nl/min using increasing ACN concentration in 0.1% formic acid from 3 to 40% over a 90-min gradient. The DIA method consisted of a survey scan from 300 to 1500 $m/z$ at 120,000 resolution and an automatic gain control (AGC) target of 3e6 or 120-ms maximum injection time. Fragmentation was performed via higher-energy collisional dissociation with a target value of 3e6 ions determined with predictive AGC. Precursor peptides were isolated with 37 variable windows spanning from 300 to 1650 $m/z$ at 30,000 resolution with an AGC target of 3e6 and automatic injection time. The normalized collision energy was 28, and the spectra were recorded in profile type. For the PPDO proteomics dataset, the DIA-PASEF mode on a timsTOF HT mass spectrometer (Bruker, Bremen, DE) was utilized. Equal amounts of peptides were automatically loaded to the online coupled VanquishNeo (Thermo Fisher Scientific, Dreieich, DE) HPLC system. A Nano-Trap column was used (300-μm inner diameter (ID) × 5 mm), packed with Acclaim PepMap100 C18, 5 μm, 100 Å from LC Packings, Sunnyvale, CA, USA, before separation by reversed-phase chromatography (Aurora 75 μm ID × 250 mm, 1.8 μm from Ionoptics, Fitzroy, AUS) at 40 °C. Peptides were eluted from the column at 250 nl/min using increasing ACN concentration in 0.1% formic acid from 3 to 40% over a 45-min gradient. The DIA-PASEF method covered a mass range from 300 to 1250 $m/z$ and a mobility range from 0.65 to 1.35 1/ko. Precursor peptides were isolated with 34 equal windows. Collision energy for 0.6 1/ko was set to 20 and for 1.6 1/ko to 59. Estimated cycle time was 1.91 s.

## Proteomics data analysis

Both PPDO and HPDO datasets were processed as a direct DIA analysis with the Spectronaut software (version 20 for PPDO and 18 for HPDO, Biognosys, Schlieren, Switzerland). The Swissprot human database using BSG factory settings for Pulsar search was used for the human dataset (Release 2020_02, 20432 sequences) and a custom-built pig protein database comprised of SwissProt pig proteins and non-redundant Trembl sequences selected for longest sequences per protein entry based on human genes (16534 sequences; version 1 generated February 2024) was used for the pig dataset using BSG factory settings for Pulsar search. For DIA analysis, default settings were applied. Quantity MS level was set to MS2; Quantity type was area, and protein quantification is based on top three peptide intensities. For HPDO unique and for PPDP protein-group specific peptides were allowed for quantification. For quantification, precursor filtering was set on Qvalue, with cross run normalization. Quantity MS level was set to MS2; Quantity type was area, and protein quantification is based on the summed-up peptide intensities. Data were exported with cross run normalization.

Data from Spectronaut were processed using the DEP2 pipeline (Feng et al, 2023) in R. Protein intensities were $\log_2$-transformed and filtered, allowing for missing values in up to one replicate per condition. Variance stabilizing normalization was applied. Missing values, primarily showed to be missing not at random, were imputed using the GSimp deep learning-based approach (Wei et al, 2018) with parameters hi_q = 0.1, iters_each = 100, and iters_all = 20. Differential expression analysis was performed using protein-wise linear models with empirical Bayes moderation as implemented in the limma package (Ritchie et al, 2015). False discovery rate was used to adjust $P$-values using the Benjamini-Hochberg method at an alpha level of 0.05, with a $\log_2$ fold change threshold of 1.

## ScRNA-Seq

For the primary porcine pancreas dataset, pancreata from two pigs (WT and *INS*-eGFP) aged at PN150 were processed separately at two different days. Pancreata were cleaned from excess fat and connective tissue, chopped into fine pieces with scissors and incubated with 2 mg/ml of collagenase V for 10 min at 37 °C. For the first pancreas, we manually isolated ductal fragments to enrich for the ductal cell population of the pancreas. Both isolated ducts and whole lysates from the second pancreas were incubated for 10 min with TrypLE Express (Gibco) to generate the single cell

suspension. Single cell suspension was stained with 7-AAD (Invitrogen) as a live/dead cell marker. 20,000 alive cells were sorted for sample #1. Sample #2 which contained the *INS*-eGFP transgene was stained with a conjugated Dolichos Biflorus Agglutinin (DBA)-biotin (1:100, Vector Laboratories B-1035-5) followed by incubation with Streptavidin Alexa Fluor 647 (1:500, Life Technologies S21374) to label ductal cells. Then 10,000 DBA$^+$ cells together with 10,000 *INS*-eGFP cells and 20,000 cells of the whole cell suspension were sorted using the FACSAria III (BD Biosciences) and all cells were pooled together.

For the organoid cultures, one confluent well of a 24-well plate containing PPDOs or HPDOs were detached from the Cultrex using cold basal medium. The cells were washed once using cold basal medium and were incubated for 30 min in Cultrex organoid harvesting solution to dissolve the extracellular matrix. PPDOs and HPDOs were dissociated to single cell suspension with TrypLE Express incubation for 10 min at 37 °C, followed by inactivation with a solution containing 1% BSA in PBS 1X. Cells were stained with 7-AAD (Invitrogen) as a live cell marker and 20,000 live single cells were sorted using the FACSAria III (BD Biosciences).

The cell suspension was immediately used for single-cell RNA-seq library preparation with a target recovery of 10,000 cells. Libraries were prepared using the Chromium Single Cell 3′ Reagent Kits v3.1 (10x Genomics, 1000268) according to the manufacturer's instructions. Libraries were pooled and sequenced on an Illumina NovaSeq6000 with a target read depth of 50,000 reads/cell. FASTQ files were aligned to the GRCh38 human genome with Ensembl release 111 annotations or to the in-house improved pig genome annotation (Yang et al, 2025) and pre-processed using the CellRanger software v7.1.0 (10x Genomics) for downstream analyses.

## scRNA-Seq analysis

For downstream analysis, ambient RNA correction was performed using the SoupX tool (Young and Behjati, 2020) using the autoEstCont function, except for the HPDO sample which was not deemed necessary to perform the correction. The corrected matrix was used for all the scRNA-Seq dataset analysis that was done with Scanpy (Wolf et al, 2018) suit of tools. For the PPDOs we filtered away genes expressed in less than 3 cells, cells with more than 3% mitochondrial gene counts, and cells containing more than 9000 and less than 600 (PN100) or 1000 (PN240). Counts were normalized to 10,000 and log transformed using natural log and pseudocount 1, top 2000 highly variable genes were calculated using the 'cell ranger' flavor in the Scanpy package and clustering was performed using the top 20 PCs of the PCA with the Leiden algorithm with a resolution of 0.3 and 0.5 for the PN100 and PN240 samples, respectively. Marker genes per cluster were identified using the Wilcoxon test and cell types/states were manually annotated based on marker genes and GO enrichment terms using the PANTHER (Thomas et al, 2022) database (Dataset EV2).

For the HPDO sample we filtered away genes expressed in less than 3 cells, cells with more than 15% mitochondrial gene counts, and cells containing more than 9000 and less than 1000 genes and 5000 counts. Counts were normalized to 10,000 and log transformed using natural log and pseudocount 1, top 2000 highly variable genes were calculated using the 'cell ranger' flavour in the

Scanpy package and clustering was performed using the top 25 PCs of the PCA with the Leiden clustering algorithm with a resolution 0.5. Marker genes per cluster were identified using the Wilcoxon test and cell types/states were manually annotated based on marker genes and GO enrichment terms using the PANTHER (Thomas et al, 2022) database.

The two primary pancreas datasets were filtered separately before merging using 8% mitochondrial gene expression cutoff for both datasets followed by including cells for sample #1: with more than 400 and less than 9000 genes and for sample #2: with more than 800 and less than 9000 genes and more than 3000 UMI counts. For both samples, doublets were identified and excluded using the scrublet (Wolock et al, 2019) tool using a doublet threshold of 0.25 and 0.15, respectively.

For integration PPDO and primary porcine datasets, we tested just aggregating the matrix, and integration using the scanorama (Hie et al, 2024), scVI (Lopez et al, 2018), and sysVI (Hrovatin et al, 2025) tools. The scanorama and sysVI integration results are shown in the manuscript.

Ligand receptor interactions were inferred using the LIANA (Dimitrov et al, 2024) framework using the rank_aggregate function. Pig orthologues were used based on the human reference list using the built-in get_hcop_orthologs function from the HCOP database.

For mapping the HPDO to the tabula sapiens pancreas dataset (Jones et al, 2022), we used the web interface of the scArches (Lotfollahi et al, 2022) (https://www.archmap.bio/#/sequencer/genemapper) to upload our filtered HPDO scRNA-Seq matrix and used the scVI (Lopez et al, 2018) and kNN approach to map our HPDO dataset to the atlas. Visualization using the UMAP was performed using the Scanpy suit.

## RNA-Seq

For the bulk RNA-Seq analysis, PPDOs cultured in complete or differentiation media were collected after seven days of culture and washed twice with PBS 1X. RNA was extracted using the Arcturus PicoPure RNA isolation kit (Applied Biosystems). RNA quality was assessed using the bioanalyzer and all samples showed a RIN value more than 9 and therefore included in library preparation using the mRNA prep ligation kit (Illumina) with poly-A tail selection, following the kit's instructions. The libraries were sequenced using the NovaseqX+ sequencer (Illumina) with a depth of ≥30 million reads per sample (paired-end mode 2 × 100 bases). Library preparation and sequencing were performed at Helmholtz Zentrum München (HMGU) by the Genomics Core Facility. Processing of the sequencing that generated the aligned reads was performed using the nf-core pipeline (v3.14) with Nextflow (23.10.1) and the salmon tool used for quantification of transcripts (Di Tommaso et al, 2017; Patel, 2024; Patro et al, 2017; Quinlan and Hall, 2010; Li et al, 2009; Dobin et al, 2013; Love et al, 2020). The reads were aligned to the custom annotated Sus Scrofa genome.

The gene length scaled counts output file from salmon was used for differential expression analysis. Raw counts from transcripts with –iso and –ext were aggregated to get the summed counts per gene transcript. The DESeq2 (Love et al, 2014) package was used for differential gene expression analysis with the apeglm (Zhu et al, 2019) log fold change shrinkage applied for effect size shrinkage and the EnhancedVolcano package for visualization. GO

enrichment analysis was performed with the PantherDB (Thomas et al, 2022) database. Bulk RNA-Seq deconvolution was performed using the dampened weighted least squares (DWLS) method (Tsoucas et al, 2019) that was recently benchmarked (Dietrich et al, 2024) to be as one of the best for this computational task. Deconvolution was assigned against the scRNA-Seq clusters of the PPDO dataset generated in this manuscript and the DWLS implementation of the algorithm according to package instructions.

## Quantitative PCR

For gene expression analysis, one well (20 µl culture) of a 8-well Ibidi glass-bottom plate (Ibidi) was collected with cold PBS, Cultrex was dissolved as described above and, washed twice with PBS1X. RNA was extracted using the Arcturus PicoPure RNA isolation kit (Applied Biosystems) and eluted with 15 µl Nucleic-acid free water according to manufacturer's instructions. 100–200 ng of RNA was used to generate cDNA the SuperScript VILO cDNA synthesis kit. RT-PCR was performed with 0.5 µl of initial RNA using the TaqMan Fast Advanced Mix and the following TaqMan probes coupled with 6-carboxyfluorescein (FAM): *INS* (Ss03386682_u1), *GCG* (Ss03384069_u1), *NEUROD1* (Ss03373333_s1), *SST* (Ss03391856_m1), *ACTB* (Ss03376563_Uh), *KRT7* (Ss06890421_Mh), and *CFTR* (Ss03389420_m1). Results were analyzed using the ∆∆ct method (Livak and Schmittgen, 2001).

## Chemical screen

PPDOs/HPDOs were dissociated into single cells as described above for passaging and 7 µl droplets were plated in a glass-bottom, 96-well plates. We cultured PPDOs/HPDOs for 2 days in complete media for organoid formation. After 2 days, cultures were washed once with PBS 1X, and chemicals from a Tocris-FDA-approved drugs library (Tocris-7200) were added in a final concentration of 20 µM in medium without WNT3a, R-Spondin-1, noggin factors. Four control wells/plate were used with complete and media without the 3 factors supplemented with DMSO (Sigma-Aldrich). PPDOs were incubated with the drugs for 3 days, followed by centrifugation of the plate for 10', at $900 \times g$ at 10 °C and fixation with 4% paraformaldehyde. PPDOs/HPDOs were permeabilized with 0.5% Triton X-100 solution for (1h-room temperature), followed by blocking (1h-room temperature) with 3% donkey serum supplemented with 0.1% Triton X-100 blocking buffer. PPDOs/HPDOs were incubated with pHH3 (1:1000, Cell Signaling) primary antibody overnight at 4 °C followed by secondary antibody incubation together with DAPI to counterstain nuclei (2h-room temperature). PPDOs/HPDOs were mounted with Vectashield (Vector Laboratories) and the whole 96-well plate was imaged using a Cell Discoverer 7 microscope (Zeiss) using the laser-scanning confocal mode and imaging two random positions/well. The whole plate was also imaged using the brightfield mode to assess the survival of the cultures. To count the percentage of proliferating cells both the DAPI and pHH3 channels were segmented in 3D using the Cellpose 2.0 algorithm (Stringer et al, 2021) on custom-trained models (trained on the 2D images of the experiment), the proliferation percentage calculated as the number of pHH3$^+$ to total DAPI$^+$ cells and the z-score was calculated in reference to complete media using the mean and standard deviation measures. The counts from the two positions/well imaged were averaged to get an overall proliferation percentage/well. For the

secondary screen, all chemicals were purchased from different vendors and diluted fresh to avoid any confounding effects using the same batch of drugs and focus on the most promising biological data. The following chemicals were purchased and screened at a final concentration of 20 µM: losartan potassium (Sigma-Aldrich), telmisartan (Santa Cruz Biotechnology), cimetidine (Sigma-Aldrich), famotidine (Sigma-Aldrich), tranylcypromine hydrochloride (Cayman Chemicals), GSK-LSD1 hydrochloride (Tocris Biosciences), picrotoxin (Sigma-Aldrich), flumazenil (Cayman Chemicals).

For the luciferase screen, we used the CellTiter-Glo(R) Luminescent Cell viability assay (Promega) according to manufacturer's protocol and measured luminescence in a standard luminometer.

## Statistics

Statistics were calculated using the GraphPad Prism software and individual statistical tests and *p* values are mentioned in the respective figure legends. For proteomics, scRNA-Seq and bulk RNA-Seq datasets statistics and tests are described in the respective Methods section. No sample size determination was done prior to experiments. Researchers were not blinded during the experiments except the chemical screen experiments during which the analysis was performed in a double-blind manner. No samples were excluded from the analysis. Individual PPDO and HPDO were randomly allocated to described experimental designs presented through this work.

### The paper explained

#### Problem
Organoids are 3D culture systems recapitulating the function and structure of the tissue of origin. Human pancreatic ductal organoids can have multiple applications in clinical practice from diagnostics to drug discovery and toxicology. Given the shortage of donors, new properly benchmarked and easily accessible pancreatic organoid models are needed to scale-up clinical relevance of pancreatic ductal organoids.

#### Results
Here, we established porcine pancreatic ductal organoids and benchmarked them to the human system. We performed an extensive characterization of the porcine and human organoid systems using single-cell transcriptome, proteome profiles and functionality assays to demonstrate the similarities shared by the two systems. We designed a high-content chemical screen using FDA-approved chemicals to evaluate porcine pancreatic ductal organoids as a toxicology tool to identify safety concerns. Our results show a conserved toxic effect of alpha adrenergic receptor antagonists to porcine and human ductal organoids.

#### Impact
Pigs share great similarities with human pancreas morphology and physiology as well as metabolism making them a great alternative for clinically relevant studies. We demonstrate that the porcine pancreatic ductal organoid systems can be used as an alternative to the human system for clinically relevant and high-throughput toxicology/drug discovery applications. Our results indicate a role for alpha adrenergic receptor antagonist drug class in pancreatic ductal cell toxicity, establishing it as a potential candidate for drug-induced pancreatitis in clinical practice.

## Data availability

Raw sequencing data are available from SRA with project accession number: PRJNA1312453. Processed bulk and scRNA-seq files can be accessed from GEO: GSE307147, GSE307148, GSE307295. Raw data were uploaded to PRIDE (Proteomics Identification Database, EMBL 2025) with the identifier PXD068063. Source data for Fig. 7 have been uploaded to BioImage Archive with accession number: S-BIAD2280. Code used to analyze scRNA-Seq and proteomics is available here: https://github.com/chrika2/Pig_organoids.

The source data of this paper are collected in the following database record: biostudies:S-SCDT-10_1038-S44321-025-00330-3.

## Peer review information

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

## Acknowledgements

We would like to thank Kerstin Diemer, Susanne Badeke, Ines Kunze and Christina Blechinger for their excellent technical support as well as Drs Alessandro Dema, Mostafa Bakhti and Elke Schlüssel for critical comments on the manuscript. We acknowledge the technical support of the Genomics Core Facility at Helmholtz Munich. Next-Generation Sequencing was also carried out at the Competence Centre for Genomic Analysis (Kiel). CK was supported by a postdoctoral fellowship from the Alexander von Humboldt Foundation. MMvDB was supported by the Helmholtz Research School for Diabetes (HRD), which is funded by the Helmholtz Association - Initiative and Networking Fund (IVF). The study was supported by the Helmholtz Society, Deutsche Forschungsgemeinschaft (DFG, German Research Foundation) – Project number 458958943, DFG Research Infrastructure NGS_CC (project 407495230) as part of the Next Generation Sequencing Competence Network (project 423957469), the German Center for Diabetes Research (DZD; 82DZD08D03), the Juvenile Diabetes Research Foundation (Breakthrough T1D; 3-SRA-2023-1420-S-B), and—in part—by the Innovative Health Initiative *NHPig* project 101165643).

## Author contributions

**Christos Karampelias**: Conceptualization; Data curation; Formal analysis; Supervision; Validation; Investigation; Visualization; Methodology; Writing—original draft; Project administration. **Kaiyuan Yang**: Data curation; Formal analysis; Investigation; Methodology; Writing—review and editing. **Falk J Farkas**: Data curation; Formal analysis; Investigation; Visualization; Methodology; Writing—review and editing. **Michael Sterr**: Data curation; Formal analysis; Investigation; Writing—review and editing. **Mireia Molina van Den Bosch**: Data curation; Investigation; Writing—review and editing. **Simone Renner**: Resources; Investigation; Writing—review and editing. **Janina Fuß**: Resources; Data curation; Investigation; Writing—review and editing. **Christine von Toerne**: Data curation; Formal analysis; Methodology; Writing—review and editing. **Sören Franzenburg**: Data curation; Formal analysis; Supervision; Investigation; Writing—review and editing. **Tatsuya Kin**: Resources; Investigation; Project administration; Writing—review and editing. **Eckhard Wolf**: Resources; Supervision; Project administration; Writing—review and editing. **Elisabeth Kemter**: Resources; Data curation; Formal analysis; Supervision; Visualization; Methodology; Project administration; Writing—review and editing. **Heiko Lickert**: Conceptualization; Resources; Supervision; Funding acquisition; Methodology; Project administration; Writing—review and editing.

Source data underlying figure panels in this paper may have individual authorship assigned. Where available, figure panel/source data authorship is listed in the following database record: biostudies:S-SCDT-10_1038-S44321-025-00330-3.

## Funding

## Disclosure and competing interests statement

The authors declare no competing interests.

# Expanded View Figures

**Figure EV1.   Pancreatic marker expression in PPDO from different developmental stages.**

(**A–C**) Single-plane confocal images of PPDO derived from one Em pig pancreas immunostained against Pan-Cytokeratin (PAN-CK)-BMPR1A-CDH1 (**A**), SOX9-Phalloidin-NKX6-1 (**B**), CFTR-GP2-PDX1 (**C**) and counterstained with DAPI. Insets show the individual channels of the merge image. PPDO were stained at passage 2. Scale bar: 50 μm. (**D–F**) Single-plane confocal images of PPDO derived from an LPN pig pancreas immunostained against KRT5-BMPR1A-CDH1 (**D**), SOX9-GCG/NKX6-1-Phalloidin (**E**), CFTR-GP2-PDX1 (**F**) and counterstained with DAPI. Insets show the individual channels of the merge image. PPDO were stained at passage 4. Scale bar: 50 μm. (**G–I**) Single-plane confocal images of PPDO derived from an Ad pig pancreas immunostained against CFTR-GP2-PDX1 (**G**), SOX9-Phalloidin-CDH1 (**H**), AMY3A-KRT7-NEUROG3 (**I**), and counterstained with DAPI. PPDO were stained at passage 3. Insets show the individual channels of the merge image. Scale bar: 50 μm. (**J–L**) Single-plane confocal images of HPDO immunostained against KRT7-AGR2-CDH1 (**J**), CFTR-BMPR1A-NKX6.1 (**K**), SOX9-Phalloidin-PDX1 (**L**), and counterstained with DAPI. Insets show the individual channels of the merge image. HPDO were stained at passage 3. Scale bar: 50 μm. (**M, N**) Single-plane confocal images of porcine pancreas cryosections (Ad) immunostained against NKX6.1-SOX9-CDH1 (**M**) and phalloidin-555-PDX1 (**N**), and counterstained with DAPI. Scale bar: 50 μm. (**O**) Quantification of the organoid area of late passage (>5) PPDO following 4 h of live imaging and treatment with forskolin. $n = 3$ independent experiments with 3 PPDO lines (2 EPN and 1 Ad). Data are shown as mean ± SD. One-way ANOVA followed by Dunnett's multiple comparisons test was used to assess significance. Not-significant (ns) $P = 0.9010$.

◀

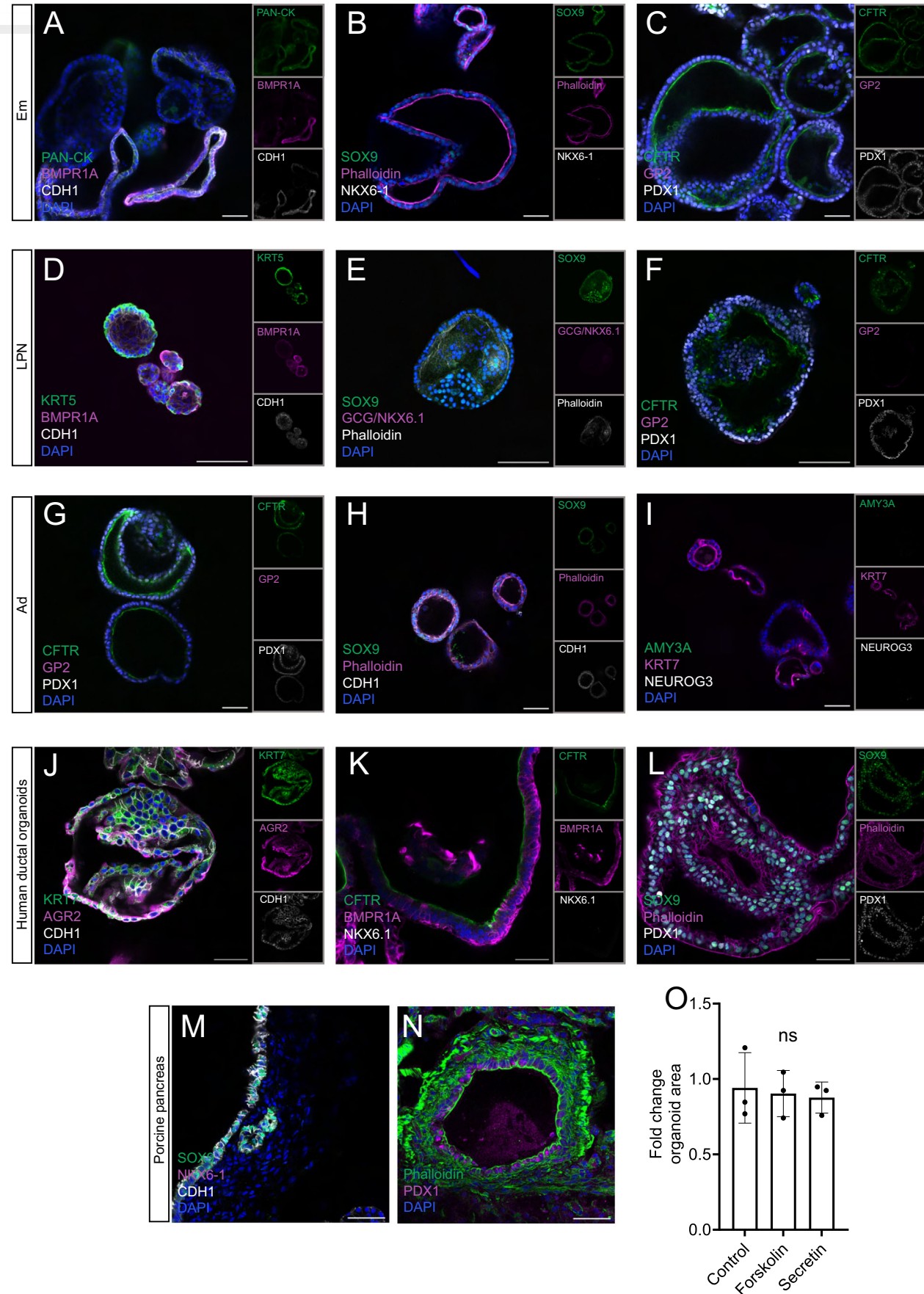

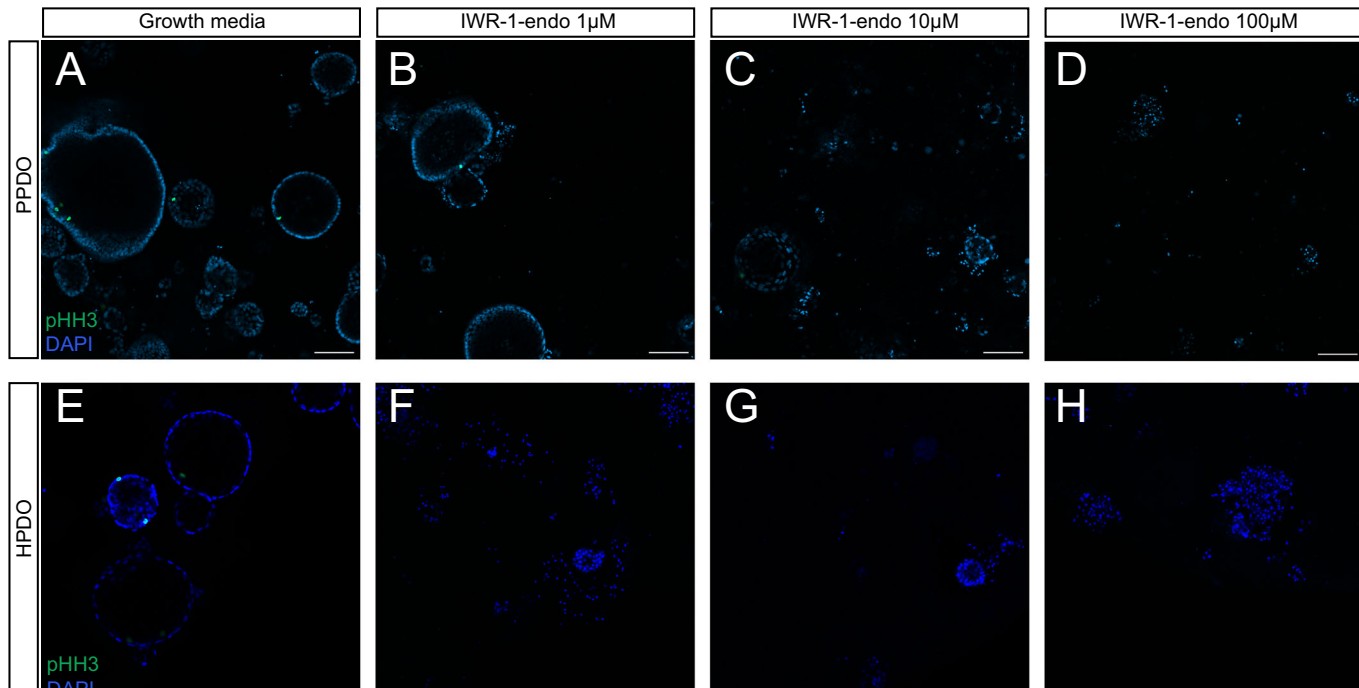

**Figure EV2.  PPDO and HPDO proliferation arrest upon WNT inhibition.**

(**A–D**) Single-plane confocal images of PPDO in growth media (**A**) or in media lacking WNT3a, R-Spondin-1 treated with 1 µM (**B**), 10 µM (**C**) or 100 µM (**D**) IWR-1-endo WNT signaling inhibitor. PPDO were immunostained for pHH3 and counterstained with DAPI. Experiment was repeated with $n = 4$ biological replicates (1 Em, 1 EPN, 2 Ad). Scale bar: 100 µm. (**E–H**) Single-plane confocal images of HPDO in growth media (**E**) or in media lacking WNT3a, R-Spondin-1 treated with 1 µM (**F**), 10 µM (**G**) or 100 µM (**H**) IWR-1-endo WNT signaling inhibitor. HPDO were immunostained for pHH3 and counterstained with DAPI. Experiment was repeated with $n = 3$ biological replicates. Scale bar: 100 µm.

    

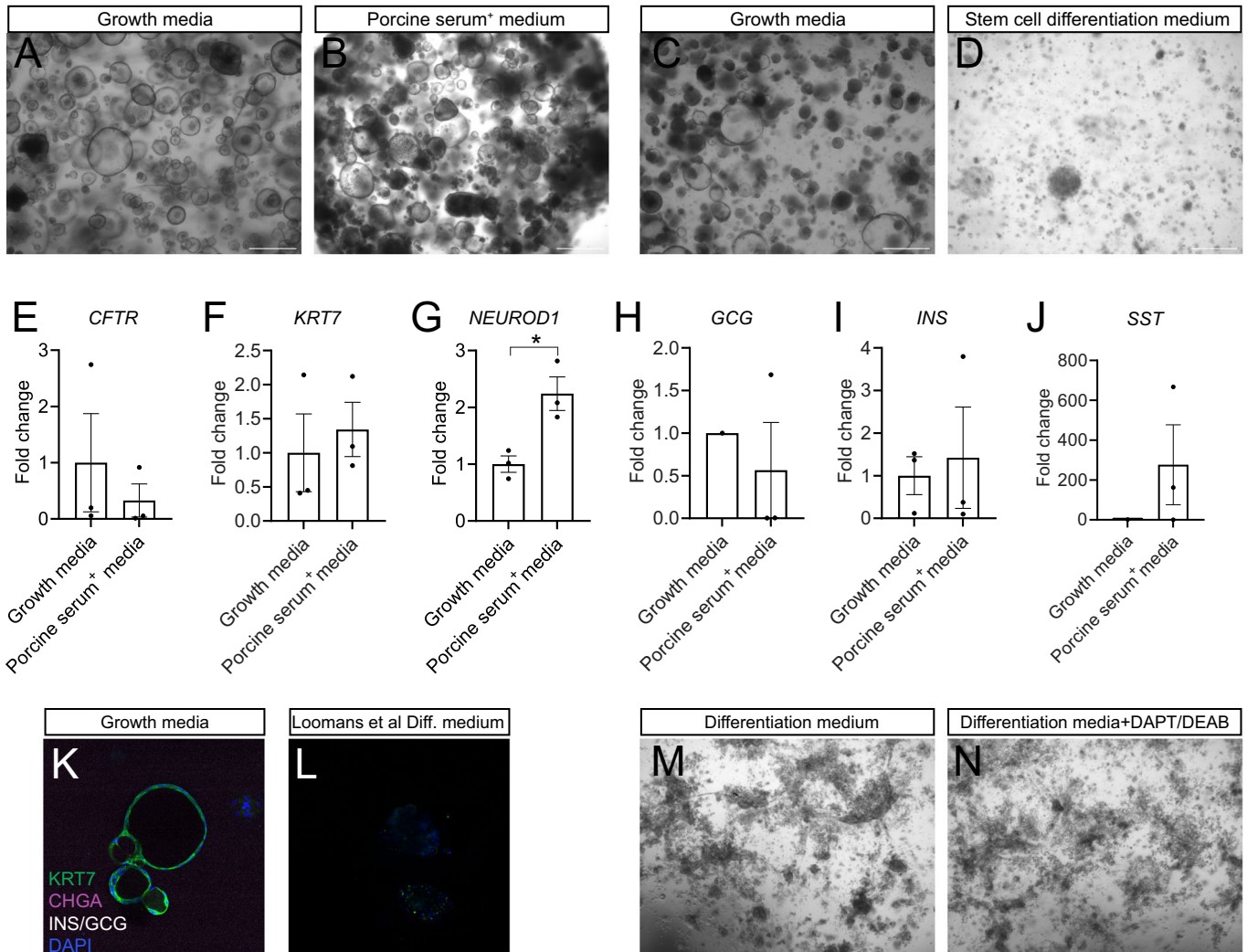

**Figure EV3. Endocrine differentiation protocol testing.**

(A, B) Brightfield microscopy images of PPDO in complete (A) or porcine serum supplemented (B). Scale bar 500 μm. (C, D) Brightfield microscopy images of PPDO at the end of differentiation in complete (C) or at the end of the differentiation using S5 + S6 combination (D) media. Scale bar 500 μm. (E–J) Bar plots showing the fold change of gene expression analysis at the end of the differentiation after treatment with porcine serum. Gene expression was measured for *CFTR* (E), *KRT7* (F), *NEUROD1* (G), *GCG* (H), *INS* (I), and *SST* (J). n = 3 independent PPDO lines (1 Em and 2 EPN). Absence of samples from the plots indicate non-detectable amplification following the RT-PCR. Data are shown as mean ± SEM. Unpaired Student's t-test was used to assess significance with *P = 0.0196 for *NEUROD1*. (K, L) Single-plane confocal images of PPDO in growth media (K) or in differentiation media from Loomans et al (L) (see Methods). PPDO were immunostained against INS-KRT7-CHGA and counterstained with DAPI. Experiment was repeated with n = 3 biological replicates (1 Em-1 EPN-1 Ad). Scale bar: 50 μm. (M, N) Brightfield microscopy images of HPDO in differentiation media (M) or differentiation media supplemented with DAPT/DEAB small molecules (N). Experiment was repeated with n = 3 biological replicates. Scale bar 500 μm.

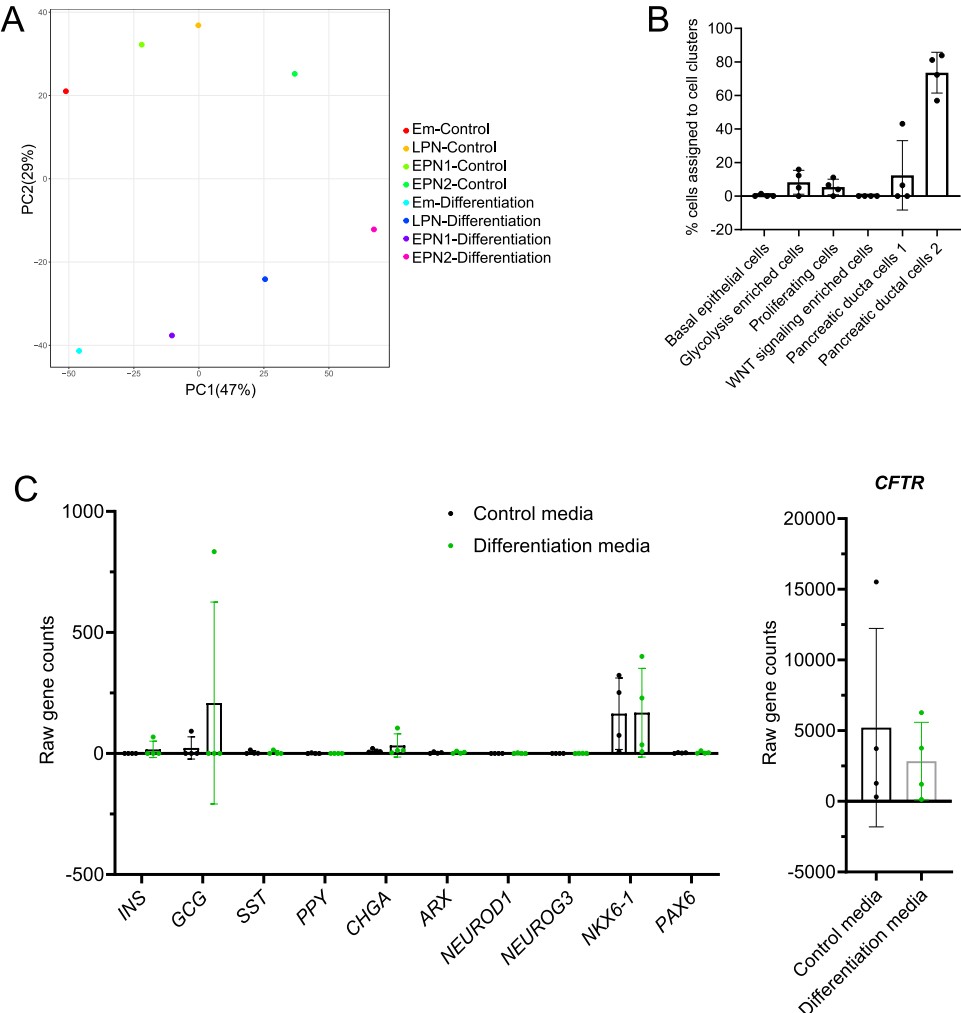

**Figure EV4. Quality assessment of the RNA-Seq differentiation analysis.**

(A) PCA plot of the bulk RNA-Seq results from the differentiation test showing the first two components. (B) Bar plot showing the percentage for each cell type found of the PPDO scRNA-Seq dataset in each of the 4 biological replicates of the bulk RNA-Seq dataset. Data show the mean ± SD. (C) Bar plot showing the raw counts (gene length scaled) of the RNA-Seq dataset for control and differentiation media. Each dot represents each of the biological replicate PPDO sequenced from 1 Em, 2 EPN and 1 LPN samples ($n = 4$). Data show the mean ± SD.

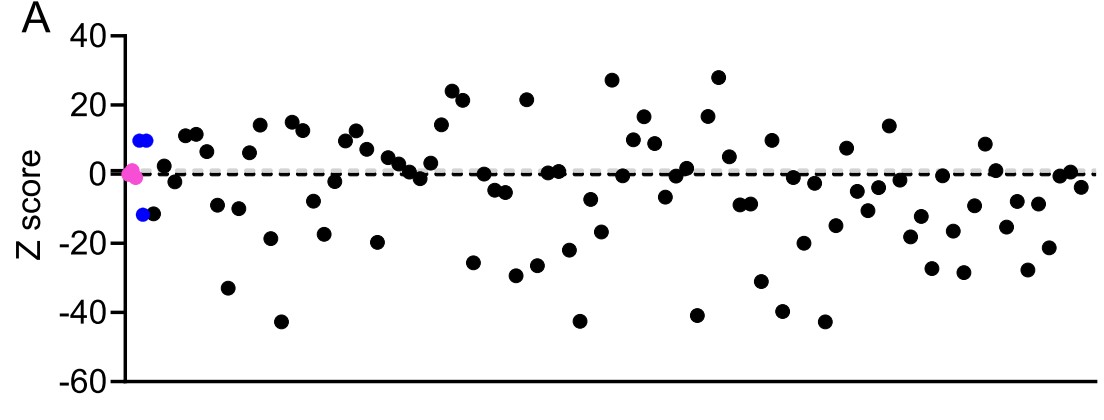

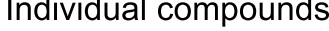

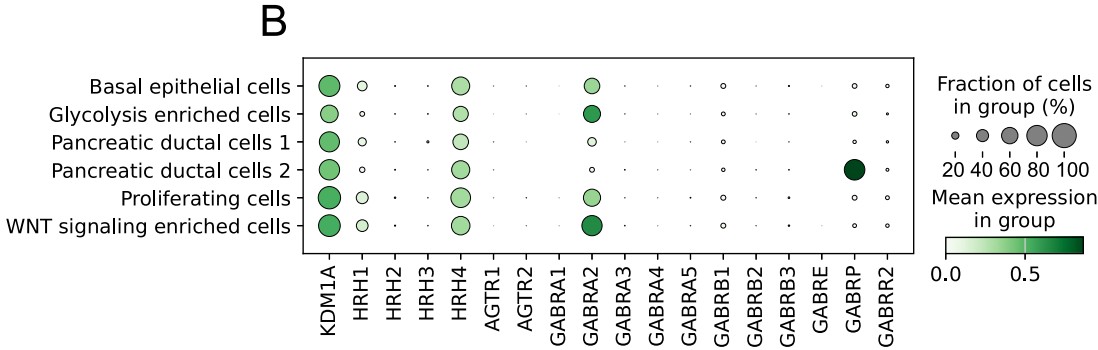

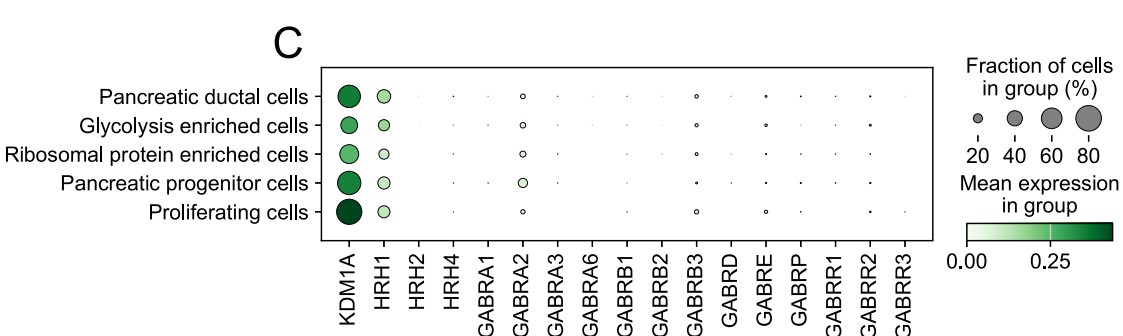

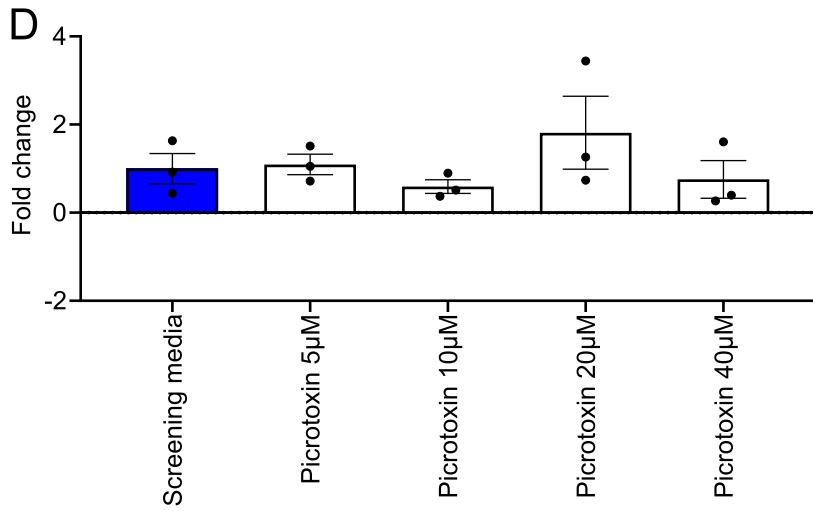

◄  **Figure EV5.  Validating primary chemical screen data.**

(**A**) Scatter plots reporting the z-scores of the ATP-luciferase assay of the primary screen results compared to the complete medium control wells for one PPDO line (Em as in main figure). Each dot represents a single well of the 96-well plate treated with a single chemical. Magenta dots show the individual wells of PPDOs treated in the complete medium with DMSO (control) and blue dots show the individual wells of PPDOs treated in screening medium (complete medium -WRN) with DMSO and the line for these two wells corresponding to the median of the 3 wells/condition. Gray dotted line demarcates z-score of 1, black dotted line a z-score of 0. (**B, C**) Dot plots showing gene expression of the genes in the pathways targeted by the primary hits of the chemical screen in the PPDO (**B**) and HPDO (**C**) scRNA-Seq dataset. (**D**) Bar plot reporting the fold change of the proliferation percentage of the HPDO dose-response experiment with picrotoxin treatment. $n = 3$ individual HPDO lines. Error bars show the ±SEM. Not significant as assessed by a Kruskal–Wallis statistical test.

