## [Peer Review File · EMBO Molecular Medicine]

Benchmarking porcine pancreatic ductal organoids for drug screening applications

Christos Karampelias, Kaiyuan Yang, Falk Farkas, Michael Sterr, Mireia Molina van Den Bosch, Simone Renner, Janina Fuß, Christine von Toerne, Soren Franzenburg, Tatsuya Kin, Eckhard Wolf, Elisabeth Kemter, and Heiko Lickert

Corresponding author: Heiko Lickert (heiko.Lickert@helmholtz-muenchen.de)

Review Timeline:

Submission Date:	10th Mar 25
Editorial Decision:	11th Apr 25
Revision Received:	11th Sep 25
Editorial Decision:	7th Oct 25
Revision Received:	10th Oct 25
Accepted:	17th Oct 25

Editor: Lise Roth

Transaction Report:

11th Apr 2025

Dear Heiko,

Thank you for submitting your manuscript to EMBO Molecular Medicine and please accept my apologies for the delay in getting back to you as one of the referees needed more time to complete his/her review. We have now heard back from the referees who have agreed to review your manuscript. As you will see below, the referees raise significant concerns about your manuscript, which unfortunately preclude its publication in EMBO Molecular Medicine in its current form.

The reviewers mention the interest of the findings but also raise a number of partially overlapping major concerns that should be addressed to strengthen the conclusions and increase the robustness of the model. After further cross-commenting with the referees, we agreed that experimental revisions should focus on:

1. More robust comparison between PPDOs and HPDOs
2. Strengthening of drug screening
3. Confirmation of duct identity
4. Better characterization of organoids from diabetic animals

If you feel that you can satisfactorily address these points, you may wish to submit a revised version of your manuscript. Please include a covering letter detailing how you have addressed each of the points raised by the reviewers. A revised manuscript will be reviewed again, and we cannot guarantee at this stage that the final outcome will be favorable.

Revising the manuscript according to the reviewers' recommendations appears to require a lot of additional work and experimentation, and given the potential interest of your findings, we are prepared to extend the deadline to 6 months.

If you would like to discuss further the points raised by the referees or your revision plan, I am available to do so via email or video. Let me know if you are interested in this option.

EMBO Molecular Medicine encourages a single round of revision only and therefore, acceptance or rejection of the manuscript will depend on the completeness of your responses included in the next, final version of the manuscript. For this reason, and to save you from any frustrations in the end, I would strongly advise against returning an incomplete revision. Should you find that the requested revisions are not feasible within the constraints outlined here and prefer, therefore, to submit your paper elsewhere, we would welcome a message to this effect.

4) A .docx formatted letter INCLUDING the reviewers' reports and your detailed point-by-point responses to their comments. As part of the EMBO Press transparent editorial process, the point-by-point response is part of the Review Process File (RPF), which will be published alongside your paper.

5) A complete author checklist, which you can download from our author guidelines (<https://www.embopress.org/page/journal/17574684/authorguide#submissionofrevisions>). Please insert information in the checklist that is also reflected in the manuscript. The completed author checklist will also be part of the RPF.

6) All Materials and Methods need to be described in the main text using our 'Structured Methods' format. According to this format, the Methods section includes a Reagents and Tools Table (listing key reagents, experimental models, software and relevant equipment and including their sources and relevant identifiers) followed by a Methods and Protocols section describing the methods, ideally using a step-by-step protocol format. The aim is to facilitate adoption of the methodologies across labs. Please download and fill our Reagents and Tools Table template (.docx), which you can find in our author guidelines: <https://www.embopress.org/page/journal/14693178/authorguide#structuredmethods>.

An example of a Method paper with Structured Methods can be found here:
<https://www.embopress.org/doi/10.15252/msb.20178071>

7) Please note that all corresponding authors are required to supply an ORCID ID for their name upon submission of a revised manuscript.

8) It is mandatory to include a 'Data Availability' section after the Materials and Methods. Before submitting your revision, primary datasets produced in this study need to be deposited in an appropriate public database, and the accession numbers and database listed under 'Data Availability'. Please remember to provide a reviewer password if the datasets are not yet public (see <https://www.embopress.org/page/journal/17574684/authorguide#dataavailability>).

9) For data quantification: please specify the name of the statistical test used to generate error bars and P values, the number (n) of independent experiments (specify technical or biological replicates) underlying each data point and the test used to calculate p-values in each figure legend. The figure legends should contain a basic description of n, P and the test applied. Graphs must include a description of the bars and the error bars (s.d., s.e.m.). Please provide exact p values.

10) Our journal encourages inclusion of *data citations in the reference list* to directly cite datasets that were re-used and obtained from public databases. Data citations in the article text are distinct from normal bibliographical citations and should directly link to the database records from which the data can be accessed. In the main text, data citations are formatted as follows: "Data ref: Smith et al, 2001" or "Data ref: NCBI Sequence Read Archive PRJNA342805, 2017". In the Reference list, data citations must be labeled with "[DATASET]". A data reference must provide the database name, accession number/identifiers and a resolvable link to the landing page from which the data can be accessed at the end of the reference. Further instructions are available at .

11) We replaced Supplementary Information with Expanded View (EV) Figures and Tables that are collapsible/expandable online. EV Figures should be cited as 'Figure EV1, Figure EV2' etc... in the text and their respective legends should be included in the main text after the legends of regular figures.

12) The paper explained: EMBO Molecular Medicine articles are accompanied by a summary of the articles to emphasize the major findings in the paper and their medical implications for the non-specialist reader. Please provide a draft summary of your article highlighting

13) Author contributions: CRediT has replaced the traditional author contributions section because it offers a systematic machine readable author contributions format that allows for more effective research assessment. Please remove the Authors Contributions from the manuscript and use the free text boxes beneath each contributing author's name in our system to add specific details on the author's contribution. More information is available in our guide to authors.

Please also suggest a visual abstract to illustrate your article as a PNG file 550 px wide x 300-600 px high. A cropped portion of this image will serve as thumbnail for the table of content on our webpage.

16) As part of the EMBO Publications transparent editorial process initiative (see our Editorial at <http://embomolmed.embopress.org/content/2/9/329>), EMBO Molecular Medicine will publish online a Review Process File (RPF) to accompany accepted manuscripts.

In the event of acceptance, this file will be published in conjunction with your paper and will include the anonymous referee reports, your point-by-point response and all pertinent correspondence relating to the manuscript. Let us know whether you agree with the publication of the RPF and as here, if you want to remove or not any figures from it prior to publication. Please note that the Authors checklist will be published at the end of the RPF.

I look forward to receiving your revised manuscript.

With kind regards,

Lise

***** Reviewer's comments *****

Referee #1 (Comments on Novelty/Model System for Author):

No ethical issues detected.

The advantages of a porcine organoid model versus the well-established human ones remain unclear and its suitability for translational applications is not thoroughly assessed.

Referee #1 (Remarks for Author):

In this manuscript, Karampelias et al. established porcine pancreatic ductal organoids (PPDO) as a model system for investigating pancreatic biology and disease. They generated PPDOs from different stages of porcine embryonic development, including embryonic (E) and post-natal (PN) stages, as well as from diabetic porcine genotypes. The authors then benchmarked the PPDOs to primary porcine pancreatic tissue and human adult duct organoids. Moreover, they assessed PPDOs suitability for translational applications by evaluating their differentiation potential and using them as a platform screening to assess the safety of FDA-approved drugs.

Cross-species comparisons are interesting and can provide valuable insights in biology. Here, the authors leverage porcine tissue to compare porcine duct organoids versus human organoids. However, the advantages of a porcine organoid model versus the well-established human ones remain unclear and its suitability for translational applications is not thoroughly assessed. Overall, the findings presented in this study are mostly inconclusive and, perhaps, the study would benefit from a more robust comparison of in vivo pancreatic tissues between the two species to better define key differences and similarities that could enhance the development of improved PPDO models.

Additional specific concerns are listed below:

-One main concern with this study is that the authors repeatedly state that there are no changes across developmental stages, but most of the comparison focuses only on two PN time points. This does not sufficiently address developmental differences,

and a more comprehensive overview would be better.

-Also, the authors generated organoids from both healthy and diabetic porcine animals, observing no morphological differences between them. However, all downstream analyses were conducted exclusively on healthy samples. If diabetic organoids are not characterized further, their inclusion seems unnecessary. The authors should either remove them or provide additional characterization. Figure 1B-D only presents brightfield microscopy images of wild-type and diabetic genotypes at 2 time points, which suggest morphological heterogeneities across cultures. However, without any quantitative analyses it cannot be ruled out if diabetic PPDOs exhibit any morphological changes and if the 'phenotypic switch' occurs earlier than in wild-type condition. Moreover, it would be important to test if the diabetic organoids exhibit differences in proliferation, differentiation, or functionality compared to controls or if they respond to drugs similarly to control organoids.

-The study includes a very elegant assay to assess organoid functionality via ductal cell bicarbonate excretion at early passages. However, it is unclear which samples were tested here (E or PN and which genotype?). Does organoid functionality change across time points, passages, or genotypes?

-The single-cell RNASeq analysis appears to have been conducted exclusively on PN samples. It would have been valuable to assess whether and how cell populations change across developmental stages, including embryonic samples. This is particularly relevant given that the E110 PPDO was the only sample to exhibit differentiation potential. The authors state that "the transcriptional signature was similar between the two PPDO lines assayed, demonstrating the similarity of PPDO cultures regardless of developmental stage," but this comparison is limited to PN stages. Additionally, comparison of transcriptome changes between early and late passages of the organoid cultures should have been analysed within the same line rather than comparing two different PPDO lines at different passages.

-The authors report that the mature pancreatic ductal cell state decreases with passaging. Would it be possible to identify a precursor population within the culture and investigate pathways involved in this transition? Given that WNT/BMP inhibition did not affect the proportion of mature cells, an *in silico* analysis could provide further insights into this.

-Based on the single-cell RNAseq analysis, the authors defined a WNT/BMP-responsive duct population as a putative progenitor state at the origin of the PPDOs. Experiments based on modulating WNT/BMP signalling were designed to test this hypothesis [depletion of WNT3a-RSpondin1-Noggin (WRN)]. There are some problems in the interpretation of these results and, possibly, flaws in the experimental design. First the expression of LGR5 is not visible in the UMAP shown in Fig 3D, only AGR2 expression is visible, and it was also validated by IF. Second, it is unclear on which basis the authors postulated the presence of a WNT/BMP-responsive population. Which is the BMP-signature examined here? and why this pathway was modulated? Perhaps, instead of a simultaneous inhibition of both pathways in the PPDO culture, it would be more helpful to assess any possible individual effects of WNT inhibition, separately from BMP inhibition, as only WNT signalling seems to be enhanced in the PPDOs.

-Related to that, was the WNT signalling enrichment more prominent in embryonic organoids, or was it consistent across developmental stages?

-In Figure 4, the authors claim that E110 organoids have a higher differentiation capacity but very few endocrine genes were tested shown and only the upregulation of NEUROD1 was statistically significant, none of the other markers. Moreover, only the results for E110 PPDOs, without any comparisons to PN organoids, were shown (Figure 4D-G). In Figure S6B, the bar plot shows one PPDO in control and differentiation media or four different PPDOs (biological or technical replicates)? In conclusion, the lack of endocrine differentiation is in line with the absence of GP2 expression in PPDO organoids and suggests that PPDO only display an adult duct identity.

-In the drug screening, authors used pHH3 immunostaining as readouts of both cell death and proliferation. This is suboptimal. It is essential to specify how the drug screening data were normalized, as in organoid cultures cell counts are difficult to obtain. Would an ATP assay provide a more objective measurement instead of immunofluorescence?

Minor points:

-The authors tested the impact of ECM proteins in endocrine differentiation by culturing the organoids in coated chambers with laminins. However, they do not specify which laminins were used, which is important since different laminins may have distinct effects.

-The manuscript should more clearly indicate which time points were used for each experiment. Are they consistent across assays?

-A table summarizing the relationship between each organoid type and its codification should be included and will be helpful for the reader. What does it refer PN100 or PN240? It is not clear from the manuscript.

-What is 'PS' referring to in Figure 3?

-Do the graphs in Supplementary Figure 2A-C refer to passage 4 or 7? This information should be specified in figure legend.

Referee #2 (Comments on Novelty/Model System for Author):

The actual data provided in the manuscript does not evidence that the porcine organoid system derived from pancreatic ducts maintain its duct identity in culture, as only <5% of the cells in 2 organoid lines characterized are identified as ductal cells with an in vivo profile. Additionally, the data supporting its differentiation strategies towards ductal or beta cell fate is weak. Furthermore, the screening results show a big variability as admitted by the authors.

Referee #2 (Remarks for Author):

In the manuscript titled "Benchmarking porcine pancreatic ductal organoids for drug screening application" C. Karampelias and colleagues aim at establishing and characterizing porcine-derived adult stem cell organoids from pancreatic ducts, benchmark their transcriptional profile against primary porcine pancreas and human pancreatic duct organoids, and finally use them as a drug screening platform that potentially mimics responses observed with human ductal organoids. Since porcine material is much more readily available than human pancreas, showing the potential of these porcine organoids in toxicology studies seems an interesting approach. However, there are critical issues with the manuscript. The actual data provided in the manuscript does not evidence that the porcine organoid system derived from pancreatic ducts maintain its duct identity in culture, as only <5% of the cells in 2 organoid lines characterized are identified as ductal cells with an in vivo profile. Additionally, the data supporting its differentiation strategies towards ductal or beta cell fate is weak. Furthermore, the screening results show a big variability as admitted by the authors. I would also argue that none of the molecules tested show a clear effect in porcine organoids when repeated in the secondary screen and therefore it is difficult to translate these hits into the human lines. Of note, there is again great variability and no clear effect in the screening on human ductal organoids, but the experimental setup (1 run, n=1 per line in 3 lines) does not allow to assess if the variability is biological or technical. Additionally, there are many questions remaining regarding both the experimental setups and the transcriptomic characterization of the lines presented in the manuscript. As there are no line number indicators I point to these issues by section:

Generation and characterization of PPDO across developmental stages

- 1) They name 3 genotypes in the first paragraph of the results section and figure 1B-E', but the authors never mention them again. In case that these different genotypes are used in the data described in the manuscript, the authors should clearly indicate the genotype of each organoid line they are displaying the results from.
- 2) The authors should define the naming conventions they use for embryonic and postnatal, and whether the subsequent number means days."
- 3) No explanation in the text of what is INS-C94Y genotype and how this is a diabetic pig model.
- 4) The stainings show irregular results between different lines and do not adjust to the statements in the text. For example, CFTR staining in Figure 1F seems to be negative, but the authors claim that is CFTR-positive. If you zoom in in the image there seems to be a very faint signal, but it overlaps with high intensity phalloidin staining. This suggest that CFTR signal observed can be an artifact. Since anti-CFTR antibodies rarely work adequately in immunofluorescence (and are rarely used by CFTR/Cystic fibrosis experts), the authors should consider performing FISH against CFTR to demonstrate its expression. Additionally, authors should show the CFTR expression level in the sc and bulk RNA samples.
- 5) In general, for the stainings, showing a primary tissue comparison will help assessing how similar/dissimilar the expression pattern is from native porcine pancreas tissue.
- 6) For figure 1F to 1H the genotype and age of the organoid line/lines used is not described. It is not clear whether all the lines correspond to the same genotype/developmental stage or not. It would be interesting to know if this line is PN19 with very low CFTR expression according to the staining or E110 (high staining signal), PN100 (middle staining signal) or PN1245 (high staining signal).
- 7) Additionally, since CFTR antibody staining is challenging, and the fold change increase in the forskolin swelling assay is modest, the authors should include a condition with specific CFTR inhibitors to highlight that CFTR-specific nature of the swelling.
- 8) According to the methods section, the authors perform the swelling assay on medium containing PGE2. This molecule has the same effect on organoid swelling as forskolin and will likely obscure any forskolin-induced swelling. As observed in Fig1 I-K', the organoids are already swollen prior adding forskolin. I recommend the authors repeating the experiment without basal PGE2 in the medium, this will likely help to assess the full swelling potential of these organoids and evaluate the CFTR functionality better.

Deep phenotyping of PPDO and benchmarking to HPDO by scRNAseq

- 9) The authors compare 1 early and 1 late passage organoid for both porcine and human lines. The early and late lines are from

different donors. Because of the inherent donor-specific variability in the samples this comparison cannot yield representative results and more donors per time point should be included. Additionally, this should be done with samples at early and late passages from the same donors.

10) Cell cluster annotation methodology is not clear. PPDO and HPDO samples are presented here with several cell type-specific clusters. However, how the authors assign cell types/states to the clusters is not clear. The methods section indicates that these were "manually annotated based on marker genes and GO enrichment terms using the PANTHER database". However, there is no figure showing which were the genes and GO terms enriched by cluster. Therefore, it is not possible to judge the basis of these annotations. Additionally, the dotplots shown in Figure2 show a rather homogeneous gene expression pattern of the selected genes across cell clusters, instead of cell type specific ones. This suggests that cells from different clusters are rather similar. In, Fig2B and D, only KRT7 is more expressed in one ductal cell cluster. SLCA4A and BMPR1A are homogeneously expressed across clusters and MUC1 and SOX9 barely expressed. Additionally, most of the genes shown in dotplots B and D (PPDOs) are not shown in F (HPDOs) (not expressed?), making the comparison and benchmarking to human organoids difficult.

11) The authors should integrate the cells from the 2 PPDO lines sequenced into 1 dataset.

12) Do ductal cells from porcine organoids express the markers detected in ductal cells from primary porcine samples (SuppFig4; PKHD1, SPP1, CFTR, CLDN10)?

13) Again, in relation to the clustering and cell type annotation, it seems that only a minority of cell from porcine ductal organoids are ductal cells (<10% of which only <5% represent a population observed in vivo, SuppFig 2A and B, SuppFig 4D and E). Thus, it is not clear what most cells in these organoids are. In contrast, HPDO present 50% of ductal cells (SuppFig 2C).

14) It would be interesting to see how the protein stainings in Figure 1 correlate with the RNA expression profiles per samples.

15) In supplementary figure 3 legend there is a typo. "(A-D)" should read "(A-B)"

Overall, Figure2, SuppFig 2 and SuppFig 3 show that there is not a great overlap between the expression profiles of porcine and human ductal organoids.

Benchmarking PPDO to primary porcine pancreas

16) In Figure4b the sample labels are swapped.

17) Figures SuppFig 4B and C do not show how some organoid cells cluster together with those of primary samples as stated in the text. If this is the case, authors should change the UMAP display to enable this visualization.

18) Why did the authors only used ductal and beta cell populations for the re-clustering with PPDO? What changes if also considering acinar cells? Are PPDO cells also different from these?

19) Again here, Supplementary figure 4D shows how the expression pattern of one of the two ductal cell types from organoids does not match that of porcine primary ductal cells. This is problematic, since ductal cell cluster 2 only encompasses <5% of all cells sequenced from the two PPDOs samples (Fig 2 and SF 2). As the authors write in the text, the ductal population 1 seems to be an in vitro artifact. Thus, I am not sure if organoids that present <5% ductal cells in culture can be called ductal organoids. Authors should characterize better most cells in PPDO cultures to have a better understanding of what cell types they represent and how do these cells match in vivo porcine pancreas biology.

Differentiation potential of PPDOs

20) The authors should include a brief description of S5 and S6 media composition/vendor in the methods and not only reference previous publications.

21) In the legend of SF5, the authors should indicate which line(s) they are using for the differentiation experiments and their developmental stage.

22) The authors should clarify how many samples (replicates) per condition they have performed bulk RNA Sequencing on. From the PCA plot on SF6 it can be deduced that it is only one sample per condition and that these are pooled in the DEG analysis of Figure4. Authors should clarify this in the text and legend of the figures. If pooled, the authors should demonstrate that there is really no stage-specific effect on the differentiation process. This does not seem to be the case, as shown in SF6B the developmental age has an impact on the beta cell differentiation (again only using 1 embryonic line with only 1 replicate). Besides, SF6A shows that the PC1 explaining most of the transcriptional difference, segregates samples by developmental stage. Lastly, their data on Figure4D-G show that there are differences. Therefore, the authors should include more replicates per developmental age or justify better how the samples can be pooled together to perform the analysis in Figure4 or if

embryonic samples (with the right replicate number) should be considered separately. If so, there should be at least 3 replicates per category.

23) Additionally, the developmental age should be included in the label of plot SF6B.

24) SF6C and D do not support the claims of the authors in the text referring to this figure, as most of the markers are homogeneously expressed between the different clusters and only few downregulated genes seem to be depleted in ductal cells.

25) The authors should include references on how one-carbon and retinoic acid metabolic processes are markers of ductal cell identity. The characterization of the ductal differentiation is superficial since the authors only show the volcano plot and GO enrichment analysis of the up- and down-regulated genes. How are these genes and GO categories expressed in the different cell types identified in their primary porcine single cell RNAseq dataset? Are they enriched in ductal cells? This simple cross-comparison would have made the analysis better.

26) The authors reason that culturing the organoids in laminin gels will benefit the endocrine differentiation process of these organoids. However, the Cultrex RGF basement membrane extract Type 2 (Bio-technie) used to grow organoids is composed mostly of laminins, and no systematic comparison is performed. In this experiment, the authors should include a quantification of the positive cell stained for INS/GCG. Are these Insulin producing cells merely being carried over into culture from the tissue biopsy for 2 passages? Or are they truly generated in vitro from progenitor cells? Again, the quantification of basal medium vs differentiation medium vs differentiation+DAPT+DEAB medium will help addressing this point and comparing early and late passages.

27) The authors should have a triplicate of E110 to claim that PPDOs of embryonic origin have a stronger potential for Beta cell differentiation than post-natal organoid lines.

28) Additionally, the cell viability of these cultures at the differentiation endpoint should be assessed as this would be key for the usability of these Beta cells in future research (Figure4 D-G). The small cell clumps depicted in Figure4 I and K might indicate that the organoids are under severe stress or dying.

Chemical screen identifies regulators of PPDO proliferation/survival

29) As it is one of the two main readouts of their screening, authors should determine cell killing by performing active caspase3 or live/dead staining. Bright field images are not enough since they only enable a qualitative measurement not compatible with a compound screening. Growth arrest could take place without inducing cell death (as indicated by the authors in the text) and it will be difficult to distinguish these two scenarios in a compound screening without a quantitative measurement of cell death.

30) From the text and and Figure5G-H legend, it is not clear that the screening medium is -WRN medium. It is clear only after checking the method section. Authors should write this clearly in the text/legend for readability.

31) In Figure5G-H it is not clear what the black dotted line indicates.

32) It is not defined in the legend or labels what green dots indicate in Figure5G-H

33) None of the hits obtained for proliferation in the primary screen showed reproducible results in the secondary screen. This precludes making any conclusion about the effect of any the compounds tested.

34) Including a positive and negative control, inducing cell death is recommended
HMG-CoA and alpha-adrenergic receptor inhibitors induce toxicity in HPDO

35) Again, cleaved caspase or live/death staining should be performed to determine cell death/proliferation arrest.

36) In figure 6, authors show that organoids are smaller in lovastatin (Fig6E) than in screening media (Fig6C). However, the proliferation screening Fig6R and S show that organoids proliferate more with lovastatin than without. In fact, they proliferate in average almost like the full medium condition. These two data pieces are not coherent. Additionally, the authors did not indicate if the Fig6D-G organoids were grown in medium with or without WRN.

37) The authors have not repeated the screening. With such standard deviations and with only 1 run, it is not possible to make any conclusion about the potential effect of the drugs on the organoids. It seems that there is no effect, and a statistical test would most likely show this. The screening needs to be repeated to confirm the results.

Referee #3 (Comments on Novelty/Model System for Author):

In this study, the authors present a thorough characterisation of porcine pancreatic ductal organoids (PPDOs) as a potential

surrogate model for human pancreatic ductal organoids (HPDOs). HPDOs are highly relevant for studies of pancreatic biology and disease but are often difficult to access in sufficient quantities. The study focuses on an impressive range of analyses, including scRNA sequencing, immunofluorescence marker profiling, and a series of functional assays aimed at defining the properties of the PPDOs. Although the characterisation is detailed and technically well executed, the extent to which PPDOs truly recapitulate the molecular and functional features of HPDOs is not entirely clear. The comparative benchmarking to human organoids is present but somewhat limited, and further analyses would strengthen the argument for the relevance of this model.

Referee #3 (Remarks for Author):

In this study, the authors present a thorough characterisation of porcine pancreatic ductal organoids (PPDOs) as a potential surrogate model for human pancreatic ductal organoids (HPDOs). HPDOs are highly relevant for studies of pancreatic biology and disease but are often difficult to access in sufficient quantities. The study focuses on an impressive range of analyses, including scRNA sequencing, immunofluorescence marker profiling, and a series of functional assays aimed at defining the properties of the PPDOs. Although the characterisation is detailed and technically well executed, the extent to which PPDOs truly recapitulate the molecular and functional features of HPDOs is not entirely clear. The comparative benchmarking to human organoids is present but somewhat limited, and further analyses would strengthen the argument for the relevance of this model.

Main Findings:

- scRNA-Seq shows that the transcriptomic signatures of PPDOs are consistent across ages but do not fully compare with HPDOs or primary porcine pancreas
- Phenotypic and molecular signatures of PPDOs change after prolonged culture
- The authors show that the PPDOs express ductal fate markers but limited endocrine differentiation is restricted to PPDOs derived from certain embryonic stages.
- The authors demonstrate the use of the PPDO system for high throughput drug screening, validating some of the targets in HPDOs.

Overall feedback:

- Well written and easy to follow
- The PPDOs are shown to be a working model but they seem to be significantly different to HPDOs. Further analyses would strengthen the relevance of this model as a potential surrogate model to HPDOs.

Major concerns

1. The authors claim to benchmark their PPDOs against HPDOs but have few experiments that compare these organoids side by side. HPDO data should be added to fig 1 F-H, 1I-L, 3 F-G, 4, 4D-K.
2. The authors generated PPDO from primary porcine pancreatic ductal cells from different genotypes. It is not always clear when PPDOs from diabetic pigs are used in figures, and more importantly the authors do not further discuss the differences between different genotypes. Differences should be explored and explained.
3. Integration of the data from figure 5 and 6 should be implemented, as this is part of the benchmarking analysis.
4. The authors should provide more experimental support to demonstrate that PPDOs are a useful addition to HPDOs.

Minor concerns

1. In the introduction, the authors mention other groups have generated HPDO from both exocrine pancreatic tissue and from pluripotent stem cell-derived pancreatic progenitors. It would be useful to discuss the differences between these, so that the comparison of PPDO to pluripotent stem cell-derived HPDO can also be discussed.
2. It is not always clear in the figures which PPDO type was used. The authors should provide more intuitive labelling for the reader to keep track of the different PPDOs used in each figure. Additionally, a table of all the different organoids used in this study could be helpful for the reader.
3. Regarding the change in PPDO phenotype in prolonged culture - have quality control experiments been performed (for example karyotyping) to investigate genome stability in these culture conditions over time?
4. The authors note that cultures without WRN factors have limited expansion potential with PPDO decreasing in number and disappearing between passage 3-6. These passage numbers correlate with the change in phenotype of the organoids - have any experiments been performed to assess whether these two phenomena are connected?

5. Please provide a rationale for removing proliferating cytokines from the PPDO medium as a differentiation protocol, for example providing a reference to other groups which have previously tried this approach.
6. The authors should try to adapt other ductal organoids endocrine differentiation protocols published (Loomans et al. 2018, Fernandez et al. 2024)
7. The resolution or colour definition of UMAP plots should be adjusted. Distinction between colours is difficult (Fig 2A, C, E, Fig 3B, D, E, S2D, E)
8. The authors should integrate the different datasets in figure 2A and 2C so differences and similarities are clearer.
9. Is Figure 3B legend correct? As it is it seems that PPDO are mainly formed by primary beta cells and primary ductal cells
10. The scale on Figure 3D-E does not seem appropriate and makes it difficult for the reader to capture any differences in expression.
11. Figure 3G shows a few Agr2-positive cells in the PPDO. This is quite different to the expression of Agr2 in the porcine pancreas (Figure 3F), where most of the ductal cells are expressing Agr2. What are those Agr2-negative cells?
12. The authors should add Chromogranin and SST to figure 4D-K.
13. Statistics should be added to S5E, F, H, I, J and include description of those in the figure legend.
14. For the lumen expansion experiment, what happens if you use late passage organoids? How does it compare to human organoids?
15. For Figure 4A, several highly significantly regulated genes are not annotated - what are they? Can there be an alternative representation where the pathways shown in the enrichment graph are highlighted in different colours?
16. In Figure 5, is PN1300 PPDO a type for PN130 PPDO?

***** Reviewer's comments *****

Referee #1 (Comments on Novelty/Model System for Author):

No ethical issues detected.

The advantages of a porcine organoid model versus the well-established human ones remain unclear and its suitability for translational applications is not thoroughly assessed.

Referee #1 (Remarks for Author):

In this manuscript, Karampelias et al. established porcine pancreatic ductal organoids (PPDO) as a model system for investigating pancreatic biology and disease. They generated PPDOs from different stages of porcine embryonic development, including embryonic (E) and post-natal (PN) stages, as well as from diabetic porcine genotypes. The authors then benchmarked the PPDOs to primary porcine pancreatic tissue and human adult duct organoids. Moreover, they assessed PPDOs suitability for translational applications by evaluating their differentiation potential and using them as a platform screening to assess the safety of FDA-approved drugs.

Cross-species comparisons are interesting and can provide valuable insights in biology. Here, the authors leverage porcine tissue to compare porcine duct organoids versus human organoids. However, the advantages of a porcine organoid model versus the well-established human ones remain unclear and its suitability for translational applications is not thoroughly assessed. Overall, the findings presented in this study are mostly inconclusive and, perhaps, the study would benefit from a more robust comparison of in vivo pancreatic tissues between the two species to better define key differences and similarities that could enhance the development of improved PPDO models.

We thank the reviewer for the thorough comments on our work. We have addressed most of the concerns raised in the revised version of our manuscript.

Additional specific concerns are listed below:

-One main concern with this study is that the authors repeatedly state that there are no changes across developmental stages, but most of the comparison focuses only on two PN time points. This does not sufficiently address developmental differences, and a more comprehensive overview would be better.

To address this important point, we performed and analyzed proteomics characterization of 20 new proteomics samples from PPDO across developmental stages (embryonic, early postnatal and adult as well as the same early postnatal PPDO in late passages that the phenotypic switch already occurred – 12 samples) and HPDO (4 samples of exocrine pancreatic fraction and 4 HPDO samples). Our analysis presented in the new Figure 2 shows that there is not a single significant protein changed between PPDO from different developmental stages as it is evident by the PCA plot as well. We provide supplementary file 1 with all the comparisons and statistical tests of this analysis for the readers. As suggested by the reviewer, testing the same PPDO in early and late passages identifies a clear switch on the proteome level to a WNT/basal cell signature, important for studying this cell type in vitro. This is an advantage of PPDO as HPDO do not enrich for this important mammalian population with implications in carcinogenesis. We also used this dataset to provide strong evidence for the similarity of PPDO and HPDO. By addressing this comment we have generated the first comprehensive proteomics characterization of pancreatic organoids, a

useful resource for the community.

-Also, the authors generated organoids from both healthy and diabetic porcine animals, observing no morphological differences between them. However, all downstream analyses were conducted exclusively on healthy samples. If diabetic organoids are not characterized further, their inclusion seems unnecessary. The authors should either remove them or provide additional characterization. Figure 1B-D only presents brightfield microscopy images of wild-type and diabetic genotypes at 2 time points, which suggest morphological heterogeneities across cultures. However, without any quantitative analyses it cannot be ruled out if diabetic PPDOs exhibit any morphological changes and if the 'phenotypic switch' occurs earlier than in wild-type condition. Moreover, it would be important to test if the diabetic organoids exhibit differences in proliferation, differentiation, or functionality compared to controls or if they respond to drugs similarly to control organoids.

As per suggestion, we removed the only instance that a diabetic PPDO was mentioned in the manuscript (former Fig 1B-B') to avoid confusion and since it is not a major part of our manuscript.

-The study includes a very elegant assay to assess organoid functionality via ductal cell bicarbonate excretion at early passages. However, it is unclear which samples were tested here (E or PN and which genotype?). Does organoid functionality change across time points, passages, or genotypes?

We introduced a sentence to clarify the developmental age that the PPDO were derived from as Em (embryonic) – early postnatal (EPN) – late postnatal (LPN) – adult (Ad) and updated all figure legends to reflect these changes. We further clarify these distinct stages in the methods section of our manuscript. No change in functionality was observed across developmental stages but a clear absence of functionality was observed when we assayed PPDO at late passages. The new data are shown in Supplementary Figure 1O.

-The single-cell RNASeq analysis appears to have been conducted exclusively on PN samples. It would have been valuable to assess whether and how cell populations change across developmental stages, including embryonic samples. This is particularly relevant given that the E110 PPDO was the only sample to exhibit differentiation potential. The authors state that "the transcriptional signature was similar between the two PPDO lines assayed, demonstrating the similarity of PPDO cultures regardless of developmental stage," but this comparison is limited to PN stages. Additionally, comparison of transcriptome changes between early and late passages of the organoid cultures should have been analysed within the same line rather than comparing two different PPDO lines at different passages.

In the revised version of the manuscript (Fig. 2D), we include the proteomics comparison of the same stage PPDO in early and late passage and highlight the important proteins in heatmap (Fig. 2E). We observed that the passaging affects proliferation, which is greatly diminished and the cells acquire a basal cell type identity shown by KRT5+ as well as express developmentally relevant progenitor markers including PROCR and JAG1. Additionally, we have integrated the two PPDO scRNA-Seq datasets (Fig 3A-B) to show transcriptional similarity of the two ages as requested by the other reviewers. Finally, we expanded our RT-PCR analysis of the differentiation protocol with additional samples and did not observe a higher differentiation capacity for the embryonic PPDO. Rather, we argue that the endocrine differentiation varies a lot in PPDO cultures given the inefficiency of the induction and have updated our results (Fig. 5D-G) and discussion to address the limitations of organoid differentiation for cell therapy.

-The authors report that the mature pancreatic ductal cell state decreases with passaging. Would it be possible to identify a precursor population within the culture and investigate pathways involved in this transition? Given that WNT/BMP inhibition did not affect the proportion of mature cells, an in silico analysis could provide further insights into this. Perform in silico analysis pseudotime and correlate with literature

We performed a pseudotime analysis using the widely accepted Palantir algorithm but given the homogeneous nature of the populations, we could not reach a safe conclusion from this data. We include the results of this analysis here for completeness, using the WNT+ population as the starting node and the pancreatic ductal cell population as the terminal state. The diffusion map shows a pseudotime trajectory from WNT+ cells to proliferating cell population and to Pancreatic ductal cell 1 populations.

-Based on the single-cell RNAseq analysis, the authors defined a WNT/BMP-responsive duct population as a putative progenitor state at the origin of the PPDOs. Experiments based on modulating WNT/BMP signalling were designed to test this hypothesis [depletion of WNT3a-RSpondin1-Noggin (WRN)]. There are some problems in the interpretation of these results and, possibly, flaws in the experimental design. First the expression of LGR5 is not visible in the UMAP shown in Fig 3D, only AGR2 expression is visible, and it was also validated by IF. Second, it is unclear on which basis the authors postulated the presence of a WNT/BMP-responsive population. Which is the BMP-signature examined here? and why this pathway was modulated? Perhaps, instead of a simultaneous inhibition of both pathways in the PPDO culture, it would be more helpful to assess any possible individual effects of WNT inhibition, separately from BMP inhibition, as only WNT signalling seems to be enhanced in the PPDOs.

We thank the reviewer for this insightful comment. To address these issues:

1- we updated the UMAP visualization and added arrows in new Fig. 4D-E to clarify the expression of LGR5 and AGR2 in the pig scRNA-Seq dataset.

2- We validated that the HPDO have similar behaviour to PPDO when removing the WNT/BMP signals from the media, forming organoids but failing to expand beyond passages 3-4 (New Fig 4H-O).

3- We performed a new experiment with both PPDO and HPDO leaving the BMP signaling intact (noggin) but removing the WNT signals (WNT3a-Rspondin) and additionally treating with a dose response of the WNT inhibitor IWR-1-endo. We observed that complete block of WNT signaling pathway led to organoid growth arrest (no proliferating cells) and toxicity at high concentrations (100uM). The data are included in the new revised EV Fig 2. Therefore, we provide strong evidence that clarifies the effect of WNT only inhibition on organoid growth across species.

-Related to that, was the WNT signalling enrichment more prominent in embryonic organoids, or was it consistent across developmental stages?

WNT signaling enrichment and distribution was similar across developmental stages.

-In Figure 4, the authors claim that E110 organoids have a higher differentiation capacity but very few endocrine genes were tested shown and only the upregulation of NEUROD1 was statistically significant, none of the other markers. Moreover, only the results for E110 PPDOs, without any comparisons to PN organoids, were shown (Figure 4D-G). In Figure S6B, the bar plot shows one PPDO in control and differentiation media or four different PPDOs (biological or technical replicates)?

In conclusion, the lack of endocrine differentiation is in line with the absence of GP2 expression in PPDO organoids and suggests that PPDO only display an adult duct identity.

We appreciate the opportunity to clarify this point. The results in the supplementary figure relating to the bulk RNA-Seq data has been performed on 4 biological replicates across different developmental stages. We added more biological replicates in our differentiation assay and we conclude that the high endocrine induction observed with the embryonic sample can be attributed to the inherent endocrine induction variability observed with the efforts to differentiate ductal organoids. We have updated the text to reflect the limitations of the differentiation assay (from us but also from the literature) and we agree that the PPDO and HPDO have limited differentiation capacity.

-In the drug screening, authors used pHH3 immunostaining as readouts of both cell death and proliferation. This is suboptimal. It is essential to specify how the drug screening data were normalized, as in organoid cultures cell counts are difficult to obtain. Would an ATP assay provide a more objective measurement instead of immunofluorescence?

We recognize that our phrasing in the text understandably was altering the meaning of our methods. We used the pHH3 marker only to assess proliferation of PPDO and not viability. Viability was assessed by the brightfield images acquired during the high-content screen. Moreover, our cellpose based segmentation and quantification of DAPI+ cells in 3D images was fairly accurate as reflected by the mask evaluation following segmentation and therefore we believe the validity of our chemical screen approach and normalization and are happy to share the segmentation model with the community. Lastly, we followed the reviewer's suggestion and performed the same chemical screen setup using the same organoid line as in Fig. 6B using the luciferase assay proposed. We observed a much higher noise level with the ATP-luciferase based assay compared to the image-based approach (New data included in EV Fig. 5A). Further, it was much easier to identify proliferation of cells using the image-based approach with Phh3 staining than the ATP luciferase-based assay highlighting the advantages of our approach.

Minor points:

-The authors tested the impact of ECM proteins in endocrine differentiation by culturing teh organoids in coated chambers with laminins. However, they do not specify which laminins were used, which is important since different laminins may have distinct effects.

We updated the methods with the company the laminin product was purchased but no information on the exact laminin composition could be provided by the manufacturer.

-The manuscript should more clearly indicate which time points were used for each experiment. Are they consistent across assays?

We added a sentence clarifying the developmental stages the PPDO are derived from in the beginning of the results' section and we updated all figure legends to reflect this change.

-A table summarizing the relationship between each organoid type and its codification should be included and will be helpful for the reader. What does it refer PN100 or PN240? It is not clear from the manuscript.

As mentioned above, we added a sentence clarifying the developmental stages the PPDO are derived from in the beginning of the results section and we updated all figure legends to reflect this change.

-What is 'PS' referring to in Figure 3?

PS refers to "passage" and now has been clarified in the text.

-Do the graphs in Supplementary Figure 2A-C refer to passage 4 or 7? This information should be specified in figure legend.

We updated the graph containing only the integrated dataset in Appendix Figure 1.

Referee #2 (Comments on Novelty/Model System for Author):

The actual data provided in the manuscript does not evidence that the porcine organoid system derived from pancreatic ducts maintain its duct identity in culture, as only <5% of the cells in 2 organoid lines characterized are identified as ductal cells with an in vivo profile. Additionally, the data supporting its differentiation strategies towards ductal or beta cell fate is weak. Furthermore, the screening results show a big variability as admitted by the authors.

Referee #2 (Remarks for Author):

In the manuscript titled "Benchmarking porcine pancreatic ductal organoids for drug screening application" C. Karampelias and colleagues aim at establishing and characterizing porcine-derived adult stem cell organoids from pancreatic ducts, benchmark their transcriptional profile against primary porcine pancreas and human pancreatic duct organoids, and finally use them as a drug screening platform that potentially mimics responses observed with human ductal organoids.

Since porcine material is much more readily available than human pancreas, showing the potential of these porcine organoids in toxicology studies seems an interesting approach. However, there are critical issues with the manuscript. The actual data provided in the manuscript does not evidence that the porcine organoid system derived from pancreatic ducts maintain its duct identity in culture, as only <5% of the cells in 2 organoid lines characterized are identified as ductal cells with an in vivo profile. Additionally, the data supporting its differentiation strategies towards ductal or beta cell fate is weak. Furthermore, the screening results show a big variability as admitted by the authors. I would also argue that none of the molecules tested show a clear effect in porcine organoids when repeated in the secondary screen and therefore it is difficult to translate these hits into the human lines. Of note, there is again great variability and no clear effect in the screening on human ductal organoids, but the experimental setup (1 run, n=1 per line in 3 lines) does not allow to assess if the variability is biological or technical.

Additionally, there are many questions remaining regarding both the experimental setups and the transcriptomic characterization of the lines presented in the manuscript. As there are no line number indicators I point to these issues by section:

We appreciate the detailed and constructive feedback on our manuscript from the reviewer that has improved all proposed aspects of our work.

Generation and characterization of PPDO across developmental stages

1) They name 3 genotypes in the first paragraph of the results section and figure 1B-E', but the authors never mention them again. In case that these different genotypes are used in the data described in the manuscript, the authors should clearly indicate the genotype of each organoid line they are displaying the results from.

We added a sentence clarifying the developmental stages the PPDO are derived from in the beginning of the results section and we updated all figure legends to reflect this change. Moreover, we explain the developmental stages in the Methods part of our manuscript.

2) The authors should define the naming conventions they use for embryonic and postnatal, and whether the subsequent number means days."

As mentioned above, we added a sentence clarifying the developmental stages the PPDO are derived from in the beginning of the results section and we updated all figure legends to reflect this change. We hope this generates clarity on the PPDO used.

3) No explanation in the text of what is INS-C94Y genotype and how this is a diabetic pig model.

As we explained in the introductory paragraph of our rebuttal, in the previous version of the manuscript, we only presented diabetic organoids in Figure 1 as a proof-of-principle that it is feasible to generate them without any obvious morphological phenotype. Following the suggestion of reviewer 1, we removed the mention of diabetic organoids from this work (only present in previous figure 1) and focus on the healthy organoid characterization and benchmarking (see also response to general point 4).

4) The stainings show irregular results between different lines and do not adjust to the statements in the text. For example, CFTR staining in Figure 1F seems to be negative, but the authors claim that is CFTR-positive. If you zoom in in the image there seems to be a very faint signal, but it overlaps with high intensity phalloidin staining. This suggest that CFTR signal observed can be an artifact. Since anti-CFTR antibodies rarely work adequately in immunofluorescence (and are rarely used by CFTR/Cystic fibrosis experts), the authors should consider performing FISH against CFTR to demonstrate its expression. Additionally, authors should show the CFTR expression level in the sc and bulk RNA samples.

This is an important topic raised by the reviewer. To address this issue, we have added new immunostainings of HPDO in Supplementary figure 1 that shows similar pattern of CFTR expression without co-staining with phalloidin. We also show the expression of CFTR in the bulk RNA-Seq (new EV. Fig. 4C) and in the scRNA-Seq PPDO dataset (new Appendix Fig. 3D) Similarly, we now show the protein level expression of CFTR in the new proteomics dataset across developmental stages (new supplementary file 1).

5) In general, for the stainings, showing a primary tissue comparison will help assessing how similar/dissimilar the expression pattern is from native porcine pancreas tissue.

We added immunostainings of pig pancreas for ductal markers in supplementary figure 1M-N.

6) For figure 1F to 1H the genotype and age of the organoid line/lines used is not described. It is not clear whether all the lines correspond to the same genotype/developmental stage or not. It would be interesting to know if this line is PN19 with very low CFTR expression according to the staining or E110 (high staining signal), PN100 (middle staining signal) or PN1245 (high staining signal).

We updated the developmental stage of the PPDO in the figure legends. In general, we do not observe changes in the functionality assay based on the developmental stage PPDO were derived from.

7) Additionally, since CFTR antibody staining is challenging, and the fold change increase in the forskolin swelling assay is modest, the authors should include a condition with specific CFTR inhibitors to highlight that CFTR-specific nature of the swelling.

We have now added a side-by-side comparison of HPDO to PPDO since HPDO are the gold standard for functionality as an additional control and benchmarking method. We observed the same fold induction rate of lumen expansion to the gold standard in our hands. (New Fig 1M). Further we highlight the absence of lumen induction in late passage organoids as a negative control assay showing absence of functionality (new supplementary figure 1O).

8) According to the methods section, the authors perform the swelling assay on medium containing PGE2. This molecule has the same effect on organoid swelling as forskolin and will likely obscure any forskolin-induced swelling. As observed in Fig1 I-K', the organoids are already swollen prior adding forskolin. I recommend the authors repeating the experiment without basal PGE2 in the medium, this will likely help to assess the full swelling potential of these organoids and evaluate the CFTR functionality better.

We did not observe any significant change on lumen expansion with/without PGE2 in our hands.

Deep phenotyping of PPDO and benchmarking to HPDO by scRNAseq

9) The authors compare 1 early and 1 late passage organoid for both porcine and human lines. The early and late lines are from different donors. Because of the inherent donor-specific variability in the samples this comparison cannot yield representative results and more donors per time point should be included. Additionally, this should be done with samples at early and late passages from the same donors.

We agree with the reviewer that this is an important point deserving of clarification. We addressed it with additional samples and a different methodology to complement our scRNA-Seq data. To this end and as mentioned to reviewer's 1 comment, we performed and analyzed proteomics characterization of 20 new proteomics samples from PPDO across developmental stages (embryonic, early postnatal and adult as well as the same early postnatal PPDO in late passages that the phenotypic switch already occurred – 12 samples) and HPDO (4 samples of exocrine pancreatic fraction and 4 HPDO samples). Our analysis presented in new figure 2 shows that there is not a single significant protein changed between PPDO from different developmental stages as it is evident by the PCA plot as well. We provide also supplementary file 1 with all the comparisons and statistical tests of this analysis for the readers. As suggested by the reviewer, testing the same PPDO in early and late passages identifies a clear switch on the proteome level to a WNT/basal cell signature, important for studying this cell type in vitro. This is a clear advantage of PPDO as HPDO do not enrich for this important mammalian population with implications in carcinogenesis. We also used this dataset to conclusively show the similarity of PPDO and HPDO. We believe

that by addressing this comment, we generated the first comprehensive proteomics characterization of pancreatic organoids, a useful resource for the community.

10) Cell cluster annotation methodology is not clear. PPDO and HPDO samples are presented here with several cell type-specific clusters. However, how the authors assign cell types/states to the clusters is not clear. The methods section indicates that these were "manually annotated based on marker genes and GO enrichment terms using the PANTHER database". However, there is no figure showing which were the genes and GO terms enriched by cluster. Therefore, it is not possible to judge the basis of these annotations. Additionally, the dotplots shown in Figure 2 show a rather homogeneous gene expression pattern of the selected genes across cell clusters, instead of cell type specific ones. This suggests that cells from different clusters are rather similar. In, Fig 2B and D, only KRT7 is more expressed in one ductal cell cluster. SLCA4A and BMPR1A are homogeneously expressed across clusters and MUC1 and SOX9 barely expressed. Additionally, most of the genes shown in dotplots B and D (PPDOs) are not shown in F (HPDOs) (not expressed?), making the comparison and benchmarking to human organoids difficult.

We appreciate the opportunity to further clarify our annotation strategy. As we presented in the methods section, we used a combination of marker genes per cluster together with GO enrichment terms to perform manual cluster annotation. To address the concerns, we include a new dot plot showing the expression of the top 5 marker genes per cluster for the integrated PPDO scRNA-Seq dataset (new Fig. 3C). As the reviewer points out correctly, these populations are rather homogeneous and that is the reason we refer to them in the text as "cell types/states" to address the fact that these might represent cells at different states e.g of cell cycle or different cell types.

Using marker genes, we annotated the "pancreatic ductal cell" clusters based on higher expression of mature ductal marker genes e.g SLC4A4, KRT7, SPP1. Basal cluster annotation was based on high expression of KRT5 and S100A2, known basal marker genes of human pancreatic ductal cells described here: PMID: 34330784 as well as a basal cluster assignment by Enrichr software against the database. The "Glycolysis enriched cluster" was annotated based on high levels of glycolysis related enzymes that was shown to be enriched also in the PantherGO Biological processes. WNT signaling enriched cluster was annotated based on the clear enrichment for LGR6 a prominent WNT signaling pathway mediator and known marker of WNT responsive stem cells in other tissues, as well as the WNT enrichment term in GO Biological term analysis. All the detailed GO terms are now included in the new supplementary file 2.

11) The authors should integrate the cells from the 2 PPDO lines sequenced into 1 dataset.

We agree with the suggestion by the reviewer and we now show the integrated dataset of the two PPDO scRNA-Seq experiments in new figure 3 and throughout our results section.

12) Do ductal cells from porcine organoids express the markers detected in ductal cells from primary porcine samples (SuppFig4; PKHD1, SPP1, CFTR, CLDN10)?

We now show that all marker genes from the primary porcine duct are expressed in the pancreatic ductal cell populations of the PPDO in updated Appendix Figure 3D.

13) Again, in relation to the clustering and cell type annotation, it seems that only a minority

of cell from porcine ductal organoids are ductal cells (<10% of which only <5% represent a population observed in vivo, SuppFig 2A and B, SuppFig 4D and E). Thus, it is not clear what most cells in these organoids are. In contrast, HPDO present 50% of ductal cells (SuppFig 2C).

As shown by our new proteomics dataset most cells in our early passage PPDO are of ductal nature. Late passage PPDO acquire a WNT+/basal cell type signature. Additionally, we have performed a computational deconvolution of our bulk RNA-Seq data to our scRNA-Seq cell type/state annotations and we identified that the majority of the cell types found in early passage PPDO are pancreatic ductal cells followed by glycolysis enriched and proliferating cell types. This new analysis is shown in EV Fig. 4B.

14) It would be interesting to see how the protein stainings in Figure 1 correlate with the RNA expression profiles per samples.

All the proteins are expressed in PPDO as assessed by proteomics analysis and levels can be found in supplementary file 1.

15) In supplementary figure 3 legend there is a typo. "(A-D)" should read "(A-B)"

We believe it was correctly assigned A-D since it refers to the 4 panels of the figure.

Overall, Figure2, SuppFig 2 and SuppFig 3 show that there is not a great overlap between the expression profiles of porcine and human ductal organoids.

Following our proteomics dataset, new bioinformatics analysis and thorough benchmarking of PPDO to HPDO with new experiments throughout the manuscript we demonstrated the similarities of PPDO and HPDO on the molecular level.

16) In Figure4b the sample labels are swapped.

Thank you for noticing, we corrected this labeling mistake on the figure.

17) Figures SuppFig 4B and C do not show how some organoid cells cluster together with those of primary samples as stated in the text. If this is the case, authors should change the UMAP display to enable this visualization.

This is a representation calculated on highly variable genes from all cell types in the integrated dataset (endocrine/exocrine/non pancreatic lineages). Given the small percentage of ductal cell highly variable genes, it is expected -given that all cell types are included- that the 2D representation look disconnected. For that reason, in main figure 4 (old figure 3), we only used beta and ductal cells for the integration to reflect the transcriptional similarities.

18) Why did the authors only used ductal and beta cell populations for the re-clustering with PPDO? What changes if also considering acinar cells? Are PPDO cells also different from these?

See reply in point 17 above.

19) Again here, Supplementary figure 4D shows how the expression pattern of one of the two ductal cell types from organoids does not match that of porcine primary ductal cells. This is problematic, since ductal cell cluster 2 only encompasses <5% of all cells sequenced from the two PPDOs samples (Fig 2 and SF 2). As the authors write in the text, the ductal population 1 seems to be an in vitro artifact. Thus, I am not sure if organoids that present

<5% ductal cells in culture can be called ductal organoids. Authors should characterize better most cells in PPDO cultures to have a better understanding of what cell types they represent and how do these cells match in vivo porcine pancreas biology.

As stated above our new proteomics dataset demonstrate the ductal lineage of PPDO and how it shifts with passaging.

Differentiation potential of PPDOs

20) The authors should include a brief description of S5 and S6 media composition/vendor in the methods and not only reference previous publications.

We updated our methods with a full description of the supplements used for the S5 and S6 media.

21) In the legend of SF5, the authors should indicate which line(s) they are using for the differentiation experiments and their developmental stage.

We updated the information.

22) The authors should clarify how many samples (replicates) per condition they have performed bulk RNA Sequencing on. From the PCA plot on SF6 it can be deduced that it is only one sample per condition and that these are pooled in the DEG analysis of Figure4. Authors should clarify this in the text and legend of the figures. If pooled, the authors should demonstrate that there is really no stage-specific effect on the differentiation process. This does not seem to be the case, as shown in SF6B the developmental age has an impact on the beta cell differentiation (again only using 1 embryonic line with only 1 replicate). Besides, SF6A shows that the PC1 explaining most of the transcriptional difference, segregates samples by developmental stage. Lastly, their data on Figure4D-G show that there are differences. Therefore, the authors should include more replicates per developmental age or justify better how the samples can be pooled together to perform the analysis in Figure4 or if embryonic samples (with the right replicate number) should be considered separately. If so, there should be at least 3 replicates per category.

To address the comments:

1- The bulk RNASeq was performed with 4 biological replicates and stages are clarified in the figure legends, so each dot on the plots reflects biological replicates. Since we established that all PPDO are similar regardless of developmental age they were derived from 4 biological replicates/treatment allows for statistical assessment of results

2- The PCA shows the segregation of samples based on differentiation protocol (PC1 47%) is strong while the effect on individual PPDO level most likely reflects subtle changes of the passaging.

3- We have added more biological replicates in Fig.4D-G (new fig 5D-G) and show that the variability is not correlating with the developmental stage but rather with the inefficiency of the protocol.

23) Additionally, the developmental age should be included in the label of plot SF6B.

Figure legend updated

24) SF6C and D do not support the claims of the authors in the text referring to this figure, as

most of the markers are homogenously expressed between the different clusters and only few downregulated genes seem to be depleted in ductal cells.

We agree with the reviewer that the scale of these plots were not adding meaningful evidence on the expression levels of the genes and therefore we removed them from the figure.

25) The authors should include references on how one-carbon and retinoic acid metabolic processes are markers of ductal cell identity. The characterization of the ductal differentiation is superficial since the authors only show the volcano plot and GOenrichment analysis of the up- and down-regulated genes. How are these genes and GO categories expressed in the different cell types identified in their primary porcine single cell RNAseq dataset? Are they enriched in ductal cells? This simple cross-comparison would have made the analysis better.

We updated the text to include the references showing the relevance of these pathways for ductal cel identity. Further most of these genes are highly expressed in our proteomics e.g CA2 and GUCA2B.

26) The authors reason that culturing the organoids in laminin gels will benefit the endocrine differentiation process of these organoids. However, the Cultrex RGF basement membrane extract Type 2 (Bio-techne) used to grow organoids is composed mostly of laminins, and no systematic comparison is performed. In this experiment, the authors should include a quantification of the positive cell stained for INS/GCG. Are these Insulin producing cells merely being carried over into culture from the tissue biopsy for 2 passages? Or are they truly generated in vitro from progenitor cells? Again, the quantification of basal medium vs differentiation medium vs differentiation+DAPT+DEAB medium will help addressing this point and comparing early and late passages.

PPDO are passaged at least two times before experimentation to ensure that no cell carryover from endocrine cells is present and we have validated this using the INS:GFP PPDO lines that reflect beta-cell expression. The different ECM composition of the pure laminin matrix and the Cultrex ECM used could potentially affect the signaling on the receptors and that is why we observe this phenotype only with the pure laminin product.

27) The authors should have a triplicate of E110 to claim that PPDOs of embryonic origin have a stronger potential for Beta cell differentiation than post-natal organoid lines.

We thank the reviewer for this critical observation. We updated our analysis with more replicates and indeed we observed that with a second embryonic PPDO line there were much milder changes in endocrine induction (Updated Figure 5D-G). Therefore, we have adapted our text and conclusions accordingly, toning down the language and conclusions made.

28) Additionally, the cell viability of these cultures at the differentiation endpoint should be assessed as this would be key for the usability of these Beta cells in future research (Figure4 D-G). The small cell clumps depicted in Figure4 I and K might indicate that the organoids are under severe stress or dying.

We appreciate the chance to explain our reasoning regarding this set of experiments. As an overall comment, we like to point that in the first version of the manuscript as well as the revised form we have clearly stated that organoid as a differentiation platform for regenerative therapy has limited potential to the far superior and already established stem-cell derived islets. Our study provides a proof-of-concept that some differentiation potential is

still present and can be used for basic developmental biology questions.

Chemical screen identifies regulators of PPDO proliferation/survival

29) As it is one of the two main readouts of their screening, authors should determine cell killing by performing active caspase3 or live/dead staining. Bright field images are not enough since they only enable a qualitative measurement not compatible with a compound screening. Growth arrest could take place without inducing cell death (as indicated by the authors in the text) and it will be difficult to distinguish these two scenarios in a compound screening without a quantitative measurement of cell death.

We thank the reviewer for the constructive feedback on our chemical screen. We now include in revised Fig 7 immunostainings of cleaved caspase 3 and annexin 647 incorporation with our most toxic chemicals following only 18 hours of treatment to assess early signs of cell death. We observed that the α -adrenergic inhibitor is very potent in inducing apoptosis with the rest of the chemicals having milder phenotypes. In the HPDO we also costained with Ki67-570 conjugate but the effects are strongly affected by the media composition (growth vs screening medium). Unfortunately, fixation really distorted the annexin V signal, which was added on live cells and hence the very weak signal presented.

30) From the text and and Figure5G-H legend, it is not clear that the screening medium is - WRN medium. It is clear only after checking the method section. Authors should write this clearly in the text/legend for readability.

We updated the figure legend with the necessary information.

31) In Figure5G-H it is not clear what the black dotted line indicates.

We updated the figure legend with the necessary information.

32) It is not defined in the legend or labels what green dots indicate in Figure5G-H

We updated the figure legend with the necessary information.

33) None of the hits obtained for proliferation in the primary screen showed reproducible results in the secondary screen. This precludes making any conclusion about the effect of any the compounds tested.

Our conclusions based on the proliferation screen is that none of the compounds stimulate ductal cell proliferation and therefore they do not pose any safety risk for carcinogenesis. To validate this claim, we performed a dose-response using the most promising candidate picrotoxin to address concentration-dependent phenotypes but no consistent phenotype was observed (New EV Fig. 5D).

34) Including a positive and negative control, inducing cell death is recommended
HMG-CoA and alpha-adrenergic receptor inhibitors induce toxicity in HPDO

In our revised manuscript we include Etoposide as a DNA inhibitor in the Caspase assay experiments (new Fig. 7J-S). The positive control for our proliferation assay is the growth media which includes the pro-proliferative WRN factors as shown also in revised Fig 4 and new EV. Fig. 2 in response to reviewer's 1 comments.

35) Again, cleaved caspase or live/death staining should be performed to determine cell death/proliferation arrest.

See reply to comment 29.

36) In figure 6, authors show that organoids are smaller in lovastatin (Fig6E) than in screening media (Fig6C). However, the proliferation screening Fig6R and S show that organoids proliferate more with lovastatin than without. In fact, they proliferate in average almost like the full medium condition. These two data pieces are not coherent. Additionally, the authors did not indicate if the Fig6D-G organoids were grown in medium with or without WRN.

We understand that the presentation of the results for this particular graph could affect interpretation of the results, therefore we removed the screening-only bar from the plot to allow for direct comparison to the complete growth media.

37) The authors have not repeated the screening. With such standard deviations and with only 1 run, it is not possible to make any conclusion about the potential effect of the drugs on the organoids. It seems that there is no effect, and a statistical test would most likely show this. The screening needs to be repeated to confirm the results.

We appreciate the chance to clarify our screening logic. Our reasoning behind performing the primary screen once with material that is tough to expand to large scale like the organoids is to quickly identify potential chemical of interest and move to the validation phase. If a chemical does not elicit a response in a primary screen then there is no incentive to further follow-up on it. We have validated our best hits with biological replicates i.e. different PPDO and HPDO lines in our secondary screen for a total of more than 6 biological replicates. This number of biological replicates provides confidence in our interpretations. However, and in response to reviewer's 1 comments, we performed the primary screen with a complimentary method (luciferase based readout) and show that our imaging-based approach gives less variable results. So, we conclude that for an imaging-based screen one replicate for the primary screen should suffice given the nature of the data, while for luciferase-based readouts more biological replicates are needed given the variability of the response.

Referee #3 (Comments on Novelty/Model System for Author):

In this study, the authors present a thorough characterisation of porcine pancreatic ductal organoids (PPDOs) as a potential surrogate model for human pancreatic ductal organoids (HPDOs). HPDOs are highly relevant for studies of pancreatic biology and disease but are often difficult to access in sufficient quantities. The study focuses on an impressive range of analyses, including scRNA sequencing, immunofluorescence marker profiling, and a series of functional assays aimed at defining the properties of the PPDOs. Although the characterisation is detailed and technically well executed, the extent to which PPDOs truly recapitulate the molecular and functional features of HPDOs is not entirely clear. The comparative benchmarking to human organoids is present but somewhat limited, and further analyses would strengthen the argument for the relevance of this model.

Referee #3 (Remarks for Author):

In this study, the authors present a thorough characterisation of porcine pancreatic ductal organoids (PPDOs) as a potential surrogate model for human pancreatic ductal organoids (HPDOs). HPDOs are highly relevant for studies of pancreatic biology and disease but are often difficult to access in sufficient quantities. The study focuses on an impressive range of

analyses, including scRNA sequencing, immunofluorescence marker profiling, and a series of functional assays aimed at defining the properties of the PPDOs. Although the characterisation is detailed and technically well executed, the extent to which PPDOs truly recapitulate the molecular and functional features of HPDOs is not entirely clear. The comparative benchmarking to human organoids is present but somewhat limited, and further analyses would strengthen the argument for the relevance of this model.

Main Findings:

- scRNA-Seq shows that the transcriptomic signatures of PPDOs are consistent across ages but do not fully compare with HPDOs or primary porcine pancreas
- Phenotypic and molecular signatures of PPDOs change after prolonged culture
- The authors show that the PPDOs express ductal fate markers but limited endocrine differentiation is restricted to PPDOs derived from certain embryonic stages.
- The authors demonstrate the use of the PPDO system for high throughput drug screening, validating some of the targets in HPDOs.

Overall feedback:

- Well written and easy to follow
- The PPDOs are shown to be a working model but they seem to be significantly different to HPDOs. Further analyses would strengthen the relevance of this model as a potential surrogate model to HPDOs.

We appreciate the positive and constructive feedback from the reviewer on our manuscript.

Major concerns

1. The authors claim to benchmark their PPDOs against HPDOs but have few experiments that compare these organoids side by side. HPDO data should be added to fig 1 F-H, 1I-L, 3 F-G, 4, 4D-K.

We agree with the reviewer that this is an important point raised. Therefore, we benchmarked all our PPDO experiments stated in the comment to HPDO and the new data can be found in the new Fig. 1M, Fig. 3F (Cross-species integration of the scRNASeq datasets to show their similarity), Fig. 4H-O (HPDO similarly rely on WRN for growth but not for establishing the cultures) and Supplementary Fig. 1J-L (immunostaining of HPDO), EV. Fig. 3M-N (PPDO differentiation protocol causes aberrant cell death of HPDO cultures).

As mentioned to the summary above, we now include 20 new proteomics samples from PPDO across developmental stages (Embryonic, early postnatal and adult as well as the same early postnatal PPDO in late passages that the phenotypic switch already occurred – 12 samples) and HPDO (4 samples of exocrine pancreatic fraction and 4 HPDO samples). This new analysis further confirms 1- the identical molecular nature of PPDO across developmental stage, 2- identifies the molecular switch of the passaging in the PPDO organoids and 3-demonstrates the shared ductal profile between PPDO and HPDO. Further, we have included side-by-side comparisons of PPDO-HPDO for all experiments presented in the manuscript and show their similarity by performing cross-species data integration of our scRNA-Seq data. In summary, we present new strong experimental

evidence to show the similarity of PPDO-HPDO.

2. The authors generated PPDO from primary porcine pancreatic ductal cells from different genotypes. It is not always clear when PPDOs from diabetic pigs are used in figures, and more importantly the authors do not further discuss the differences between different genotypes. Differences should be explored and explained.

As stated above, we agree that a deeper characterization of organoids from diabetic animals is an important research direction. However, a comprehensive benchmarking study, as suggested, would require a significantly larger cohort of diabetic pigs. Given the long gestation period and breeding timeline, this is logistically not feasible within the revision period for this manuscript. In the previous version of the manuscript, we only presented diabetic organoids in Figure 1 as a proof-of-principle that it is feasible to generate them without any obvious morphological phenotype. Following the suggestion of reviewer 1, we removed the mention of diabetic organoids from this work (only present in previous figure 1) and focus on the healthy organoid characterization and benchmarking, which is the core contribution of this work.

3. Integration of the data from figure 5 and 6 should be implemented, as this is part of the benchmarking analysis.

Per suggestion, we integrated the datasets of these two figures in the new figures 6 and 7 of the revised manuscript.

4. The authors should provide more experimental support to demonstrate that PPDOs are a useful addition to HPDOs.

See reply to comment 1 and additionally, we adopted the discussion and parts of the text to describe the advantages of PPDO. Especially the phenotypic switch to a more basal-like progenitor type is an important addition to the community to study this rare population.

Minor concerns

1. In the introduction, the authors mention other groups have generated HPDO from both exocrine pancreatic tissue and from pluripotent stem cell-derived pancreatic progenitors. It would be useful to discuss the differences between these, so that the comparison of PPDO to pluripotent stem cell-derived HPDO can also be discussed.

We updated the introduction on line 70 to briefly address this point.

2. It is not always clear in the figures which PPDO type was used. The authors should provide more intuitive labelling for the reader to keep track of the different PPDOs used in each figure. Additionally, a table of all the different organoids used in this study could be helpful for the reader.

We simplified the nomenclature used and mention it at the beginning of the results section. All figure legends were updated with this nomenclature.

3. Regarding the change in PPDO phenotype in prolonged culture - have quality control experiments been performed (for example karyotyping) to investigate genome stability in these culture conditions over time?

It was not feasible to culture the necessary amount of required material for the karyotyping experiments.

4. The authors note that cultures without WRN factors have limited expansion potential with PPDO decreasing in number and disappearing between passage 3-6. These passage numbers correlate with the change in phenotype of the organoids - have any experiments been performed to assess whether these two phenomena are connected?

This is an interesting observation raised. We speculate that the levels of WNT activation might play a role in this transition as we now also add data that complete WNT inhibition arrests PPDO and HPDO growth. However, more tedious experiments with different WNT ligands and concentrations need to be performed to conclusively assess the role of WNT signaling in this transition specifically in PPDO but not in HPDO.

5. Please provide a rationale for removing proliferating cytokines from the PPDO medium as a differentiation protocol, for example providing a reference to other groups which have previously tried this approach.

We updated the text with a reference to Huang et al. that used similar medium for gastric organoid differentiation and we inadvertently omitted in the first version. If there are any additional references that we have missed we are more than happy to provide proper credit.

6. The authors should try to adapt other ductal organoids endocrine differentiation protocols published (Loomans et al. 2018, Fernandez et al. 2024)

In the first version of the manuscript, we already tested an adapted version of the Fernandez protocol but was toxic to the PPDO. In the revised version, we tested the recipe of Loomans et al. (New EV. Fig. 6K-L) and while less toxic to the PPDO we did not see any endocrine cells, although KRT7 as a proxy for ductal cell identity was reduced.

7. The resolution or colour definition of UMAP plots should be adjusted. Distinction between colours is difficult (Fig 2A, C, E, Fig 3B, D, E, S2D, E)

We adjusted the UMAP colors to make the differences more clear and added arrows when specific populations were mentioned, but differences may still be not so easily discerned given that we maintain color blind friendly palettes.

8. The authors should integrate the different datasets in figure 2A and 2C so differences and similarities are clearer.

As also requested by reviewer 2, we integrated the two datasets and present them in new Fig 3A-B and throughout the manuscript.

9. Is Figure 3B legend correct? As it is it seems that PPDO are mainly formed by primary beta cells and primary ductal cells

We corrected the mistake on the figure.

10. The scale on Figure 3D-E does not seem appropriate and makes it difficult for the reader to capture any differences in expression.

We adjusted the UMAP colors to make the differences more clear and added arrows when specific populations were mentioned, but differences may still be not so easily discerned given that we maintain color blind friendly palettes.

11. Figure 3G shows a few Agr2-positive cells in the PPDO. This is quite different to the expression of Agr2 in the porcine pancreas (Figure 3F), where most of the ductal cells are expressing Agr2. What are those Agr2-negative cells?

We speculate that AGR2 reflects the WNT+ population in the PPDO cultures and that is why it has a heterogeneous expression pattern.

12. The authors should add Chromogranin and SST to figure 4D-K.

We performed experiments with SST primer but no amplification was observed that could reflect primer issue or negligible levels of SST.

13. Statistics should be added to S5E, F, H, I, J and include description of those in the figure legend.

We added statistics in all figure legends and notations in figures where appropriate.

14. For the lumen expansion experiment, what happens if you use late passage organoids? How does it compare to human organoids?

We performed the suggested experiments and include them in EV. Fig. 10. Our results indicate the loss of functionality of late passage PPDO.

15. For Figure 4A, several highly significantly regulated genes are not annotated - what are they? Can there be an alternative representation where the pathways shown in the enrichment graph are highlighted in different colours?

We agree with the reviewer's comment, and we provide the full list of differentially expressed genes for Figure 4A as supplementary file 3.

16. In Figure 5, is PN1300 PPDO a type for PN130 PPDO?

We have updated the annotation with our revised model in the figure legend.

7th Oct 2025

Dear Prof. Lickert, Dear Heiko,

Thank you for submitting your revised study to EMBO Molecular Medicine. We have now received the reports from the referees. As you will see, referees #1 and #3 are satisfied with the revisions, but referee #2 still has concerns regarding ductal identification confirmation, PPDO/HPDO comparison, and drug screening. Following further discussion within the team and consultation with the referees, we concluded that these issues could be resolved by rephrasing and editing the text, and that no further experiments are necessary at this stage.

I will therefore be able to accept your manuscript once the following minor issues have been addressed:

1/ Please address the remaining referees' concerns via discussion and text changes.

2/ Manuscript text:

- Please indicate in track changes mode any new modification in the text.
- Summary should be Abstract.
- We can accommodate a maximum of 5 keywords, please adjust accordingly.
- The manuscript sections should be in the following order: Title page - Abstract & Keywords - Introduction - Results - Discussion - Methods - Data Availability - Acknowledgments - Disclosure Statement & Competing Interests - References - Figure Legends - (Main Tables with legends if applicable) - Expanded View Figure Legends.
- Methods and Protocols should be renamed Methods.
 - o Animals: detail the origin of the animals, as well as the housing and husbandry conditions.
 - o Human material: include a statement confirming that informed consent was obtained from all subjects and that the experiments conformed to the principles set out in the WMA Declaration of Helsinki and the Department of Health and Human Services Belmont Report.
- Resource Availability should be renamed Data Availability and placed after the Methods. Please remove "All other source data are available as part of this manuscript." and "All other data in the manuscript are available upon request from the corresponding author."
- Declaration of interests should be renamed Disclosure and competing interests statement. Please review our policy at <https://www.embopress.org/competing-interests>.
- Acknowledgements: the funding information in the manuscript should match the information provided in the submission system. Please check whether Helmholtz Association - Initiative and Networking Fund needs to be added to the submission system.

3/ Figures:

- Please carefully check the composition of your figures EV2 and 7. Please note that figure re-use is allowed, but should be mentioned in the legend.
- The nomenclature Expanded View Figure 1 etc. and EV Figure 1, etc. should be updated to Figure EV1, etc.
- Please remove the "All Figures merged" PDF as we only need the figures uploaded separately.
- The 3 Appendix figures should be combined in a PDF together with their legends currently provided in the manuscript (need to be removed) - each legend should follow its figure and the PDF should have a title page with a short table of content listing each figure and its page. Alternatively, these figures could be made additional EV figures.
- The 3 Excel files titled Supplementary File 1-3 should be Datasets EV1-EV3 (the third one could be an EV table - Table EV1 - since it is a small table); their legends should be removed from the manuscript and each should be in its Excel file (for datasets it should be provided as a separate sheet/tab).
- The callouts of Supplementary Files 1-3 need to be updated to the correct callouts.
- Please address the queries from our data editors in the figure legends:
- Please define the annotated p values ****/***/**/* as well as provide the exact p-values in the legends.
- Please indicate the statistical test used for data analysis in the legends of figures 2C, D, F; 5A
- Please note that information related to n is missing in the legends of figures 2C, D, F
- Please note that the error bars are not defined in the legends of figures 1L, M; 5D E, F, G; EV1 O, EV3 E-J
- Please note that the measure of center for the error bars needs to be defined in the legends of figures EV4 B, C
- Please note that the arrow heads are not defined in the legend of figure 5I, 7J, K, L, M. This needs to be rectified.

4/ Source Data:

- Please check the provided file for Figure 5J
- Please provide source data for Figure 7.

5/ Checklist:

- Cell lines authentication and mycoplasma contamination: please check whether you need to fill this section and make sure the corresponding information is in the manuscript.
- Please check 'human clinical and genomic datasets' subsection and adjust if needed.

6/ Thank you for providing the Paper Explained; please include in the manuscript text file.

7/ I included minor edits to your synopsis text, please review and amend as you see fit:

"State-of-the-art single-cell RNA sequencing and proteomics approaches were employed to benchmark porcine and human pancreatic ductal organoid systems for use as a drug screening and testing platform.

- No phenotypic differences were observed in the developmental stage from which the porcine pancreatic ductal organoids were derived.
- Porcine pancreatic ductal organoids retained their identity in vitro, but switched to a progenitor fate during passaging, unlike the human system.
- The capacity for differentiation towards pancreatic lineages was limited.
- Toxicology screening using pancreatic ductal organoids revealed that certain FDA-approved drug classes can be involved in rare cases of drug-induced pancreatitis."

Please resize your visual abstract to a tiff/jpeg/PNG file 550 px wide x 300-600 px high and make sure that the text remains legible. A cropped portion of this image will serve as thumbnail for the table of content on our webpage.

8/ As part of the EMBO Publications transparent editorial process initiative (see our Editorial at <http://embomolmed.embopress.org/content/2/9/329>), EMBO Molecular Medicine will publish online a Review Process File (RPF) to accompany accepted manuscripts.

This file will be published in conjunction with your paper and will include the anonymous referee reports, your point-by-point response and all pertinent correspondence relating to the manuscript. Let us know whether you agree with the publication of the RPF and as here, if you want to remove or not any figures from it prior to publication.

I look forward to receiving your revised manuscript.

With kind regards,

Lise

***** Reviewer's comments *****

Referee #1 (Remarks for Author):

In the revised version of this manuscript, the authors have satisfactorily responded to all the points raised by this referee. The new set of experiments included has greatly improved the work.

Referee #2 (Comments on Novelty/Model System for Author):

In the revised version of the manuscript titled "Benchmarking porcine pancreatic ductal organoids for drug screening applications" Karampelias and colleagues aim at establishing and characterizing porcine-derived pancreatic adult stem cell organoids. They characterize the presence of ductal cells, compare them to human pancreatic organoids, and assess their potential to differentiate in culture. Finally, they explore the potential use in a drug screening setup. The revised version of the manuscript addresses a considerable number of points raised during the first round of revision. The authors have considered and incorporated data and analysis corresponding to the single cell dataset, as well as in the drug screening section of the manuscript, as well as rewritten considerable text sections accompanying these parts. Furthermore, the addition of the proteomic dataset could be a nice complement to support the claims of ductal cell identity and benchmark against the human pancreatic derived organoid system. Unfortunately, this is not always the case, and several key points remain unaddressed.

Especially, it is not clear yet to what extent these organoids are of ductal nature and how they compare to the human counterpart. Additionally, the drug screening avenue could be of great potential but unfortunately the experimental set up does not allow conclusions, since still only 1 replicate per donor is used in the secondary screening. I develop these points further in the following comments:

Ductal identity confirmation

The validation of the ductal identity of the PPDOs is confusing and the three main lines of evidence (stainings, proteomics and scRNAseq) presented by the authors are still contradictory.

From the scRNAseq analysis, it seems that there is a very small percentage of ductal cells, approximately 2% (Appendix figure 1A) with a gene signature representing a duct state found in vivo (Appendix figure 3D) as the authors affirm in the text (lines 249-254).

Stainings show the presence of some ductal makers. However, the lack of quantification in the stainings (Fig1 and EVFig1) precludes making a comparison with the scRNAseq results in terms of ductal cell percentage in the cultures. Additionally, the exact passage number for which the organoids were used in stainings is missing. This is important because organoids under passage 2 can carry over cells present in the originating tissue.

Authors should clearly define what is considered early and late passage as now the text states that dense phenotype appeared from passage 3 and was dominant from passage 6 (lines 127-130). It is not clear whether early passage is up to 2 passages or 6.

Proteomics data only shows z-scores for relative abundance of a selected panel of proteins in early and late passages. The main conclusion that can be drawn from this figure is that early passage organoids have a more ductal character that is lost over time in culture. However, the lack of absolute values and primary porcine pancreas samples impairs assessing how early passage organoids retain a relevant level of ductal markers in vitro.

To reconcile these discrepancies, the authors should quantify the stainings of early/late passage PPDOs for ductal cell markers, including primary porcine pancreas samples to be able to assess how representative the stainings are. Additionally, including primary porcine pancreas samples in the proteomics analysis will be good. Besides, showing protein rank plots per sample with the main ductal markers highlighted might help assess the ductal character of the samples. Finally, showing the protein levels (proteomics) of the porcine ductal markers identified in Appendix Fig1A at the protein level will highlight the ductal character of the cultures.

In section "Differentiation potential of PPDOs", the references added do not show that the pathways "one carbon compound transport" and "retinoic acid metabolism" are well established ductal markers but they pathways that play a role in the ability of certain ductal-acinar subpopulations to differentiated towards beta cells during development. Since the authors claim in this section that the differentiation medium promotes a mature ductal fate (line 329, 347), how are the canonical markers that have been mentioned by the authors in other sections induced in the differentiation medium (SLC4A4, KRT7, S100A2, CFTR, etc.)?

PPDO-HPDO comparison

The analysis comparing porcine and human organoid systems is still not conclusive.

The way the proteomic data is presented does not allow comparing the two organoid systems. Fig3E compares PPDOs from different stages and shows that EPN(LPS) have decreased ductal markers. Fig3G shows that HPDO has increased ductal markers and decreased acinar compared to the exocrine pancreas fraction. Thus, these comparisons of unrelated conditions do not allow to conclude on the similarity of the human and porcine samples. Protein rank plots showing how ductal/acinar/stem markers rank within all the detected proteins in both porcine and human organoid samples can help assessing their ductal character (relevant also for previous comment in "Ductal identity confirmation" section) and similarity at proteome level between porcine and human organoid systems.

Drug screening

Authors show in Fig6 B and C that the same compounds have different effects in PPDOs depending on if they are of adult or embryonic origin in the primary screening. In the secondary screening, the authors pool 2 late post-natal and 1 adult line and use 1 data point per line as replicate. This could influence the big spread between replicates observed in the secondary screening. Independently of the age effect, and as stated by the authors, great donor variability is observed in organoid cultures generally. Thus, at least 3 replicates per donor line should be used to assess whether the compounds affect proliferation in a donor-dependent manner. As it is right now, it is not possible to assess if the lack of response is real, is due to technical variability, or if potential donor-specific effects are masked by lack of replicates. Additionally, in Fig6O authors should have included a condition where organoids were grown in complete medium to assess how much the compound effects compare to the positive control media condition. It is stated in the text (line 378-379) but not shown in Figure6.

Minor points

I wonder if in line 82 "Stem cell-derived organoids" should read "Induced-Pluripotent stem cell organoids".

In line 305 there is a reference to figure panel missing

Conclusion on lines 328-329 is not substantiated by data in figure 5

Noggin is a BMP inhibitor. If you remove noggin from the medium, you are increasing BMP signaling in organoid cells, not removing BMP signals (lines 264-266)

Referee #3 (Remarks for Author):

The authors thoroughly addressed all the comments raised, significantly enhancing the strength of the manuscript.

***** Reviewer's comments *****

We would like to thank again all reviewers for taking the time to read our revised manuscript. We deeply appreciate the scholarly feedback that has improved our work. We are happy that reviewers 1 and 3 were satisfied with the revision overall and therefore we will address/clarify the remaining comments from reviewer 2 below.

Referee #1 (Remarks for Author):

In the revised version of this manuscript, the authors have satisfactorily responded to all the points raised by this referee. The new set of experiments included has greatly improved the work.

Referee #2 (Comments on Novelty/Model System for Author):

In the revised version of the manuscript titled "Benchmarking porcine pancreatic ductal organoids for drug screening applications" Karampelias and colleagues aim at establishing and characterizing porcine-derived pancreatic adult stem cell organoids. They characterize the presence of ductal cells, compare them to human pancreatic organoids, and assess their potential to differentiate in culture. Finally, they explore the potential use in a drug screening setup. The revised version of the manuscript addresses a considerable number of points raised during the first round of revision. The authors have considered and incorporated data and analysis corresponding to the single cell dataset, as well as in the drug screening section of the manuscript, as well as rewritten considerable text sections accompanying these parts. Furthermore, the addition of the proteomic dataset could be a nice complement to support the claims of ductal cell identity and benchmark against the human pancreatic derived organoid system. Unfortunately, this is not always the case, and several key points remain unaddressed. Especially, it is not clear yet to what extent these organoids are of ductal nature and how they compare to the human counterpart. Additionally, the drug screening avenue could be of great potential but unfortunately the experimental set up does not allow conclusions, since still only 1 replicate per donor is used in the secondary screening. I develop these points further in the following comments:

We are grateful for the feedback and are glad that the reviewer appreciated the amount of work performed to address the comments.

Ductal identity confirmation

The validation of the ductal identity of the PPDOs is confusing and the three main lines of evidence (stainings, proteomics and scRNAseq) presented by the authors are still contradictory.

From the scRNAseq analysis, it seems that there is a very small percentage of ductal cells, approximately 2% (Appendix figure 1A) with a gene signature representing a duct state found in vivo (Appendix figure 3D) as the authors affirm in the text (lines 249-254).

Stainings show the presence of some ductal makers. However, the lack of quantification in the stainings (Fig1 and EVFig1) precludes making a comparison with the scRNAseq results in terms of ductal cell percentage in the cultures. Additionally, the exact passage number for which the organoids were used in stainings is missing. This is important because organoids under passage 2 can carry over cells present in the originating tissue.

Authors should clearly define what is considered early and late passage as now the text states that dense phenotype appeared from passage 3 and was dominant from passage 6 (lines 127-130). It is not clear whether early passage is up to 2 passages or 6.

Proteomics data only shows z-scores for relative abundance of a selected panel of proteins in early and late passages. The main conclusion that can be drawn from this figure is that early passage organoids have a more ductal character that is lost over time in culture. However, the lack of absolute values and primary porcine pancreas samples impairs assessing how early passage organoids retain a relevant level of ductal markers in vitro.

To reconcile these discrepancies, the authors should quantify the stainings of early/late passage PPDOs for ductal cell markers, including primary porcine pancreas samples to be able to assess how representative the stainings are. Additionally, including primary porcine pancreas samples in the proteomics analysis will be good. Besides, showing protein rank plots per sample with the main ductal markers highlighted might help assess the ductal character of the samples. Finally, showing the protein levels (proteomics) of the porcine ductal markers identified in Appendix Fig1A at the protein level will highlight the ductal character of the cultures.

We thank the reviewer for the helpful comments and observations. To address the points raised in the section above, we have adopted our text as follows:

- 1. In the revised manuscript (lines 129-132), we clarify the exact passaging that the morphological transformation occurs in the PPDO system to the level that we are comfortable reporting in alignment with our data.*
- 2. We tried to further clarify (lines 189-190) that the passage the scRNA-Seq were collected reflect the timepoint that mature ductal cell identity is gradually lost and the progenitor signature appears. Taken together, all our three lines of evidence (stainings-scRNASeq-proteomics) are in agreement with our statements that the PPDO undergo a loss of mature ductal cell identity over passaging and as acknowledged by the reviewer above.*
- 3. Regarding the comment of quantification, all ductal markers on our PPDO and HPDO staining were present with the same intensity for 100% of the cells and therefore it is redundant to report such quantification (unless otherwise stated as in Fig4 – AGR2 expression). We have updated the text on line 146 to reflect this quantification information.*
- 4. All proteomics data for all samples including the raw normalized abundance levels that the reviewer requested are included in supplementary Dataset EV1. This Dataset file together with all the raw spectral proteomics data are available to the community. They contain all the information that we used to claim the ductal cell identity of the early passage PPDO and characterize their transformation in vitro. We believe that our data interpretation accurately reflects the observed differences and highlighted in the text as acknowledged by the reviewer. Further, it is not feasible at this stage to add more porcine pancreas samples within the timeline.*

Moreover, in our heatmap, we picked interesting genes from the list that highlight the differences of the transformation and relate to the available literature and especially include all the prototypical ductal marker proteins (KRT7, MUC1, SPP1) as requested from the reviewer (Fig 2). For further clarification, we add the normalized abundance of the typical porcine ductal markers shown in Appendix Fig 3D along with proteins detected in the staining experiments with their values in the table below.

Protein name	PPDO ePN ePS	PPDO ePN IPS	HPDO
SLC4A4	11.91548817	9.362214896	14.16249985
KRT7	15.99045129	12.31215448	17.7308565
CFTR	8.05327572	6.895952472	8.467885492
CLDN10	7.46672751	4.948366749	6.267655012
SOX9	7.529822508	6.34318461	12.31742
KRT8	14.77332486	7.541910551	22.68738429
CDH1	11.08840257	10.45940369	15.08841825

Table 1: Normalized and imputed values from the proteomics dataset for ductal cell identity genes.

- As part of the response to the previous revision comments, we have added representative ductal markers for the porcine pancreas in our Figure EV1 and we include several ductal markers of our stainings in the figures.

In section "Differentiation potential of PPDOs", the references added do not show that the pathways "one carbon compound transport" and "retinoic acid metabolism" are well established ductal markers but they pathways that play a role in the ability of certain ductal-acinar subpopulations to differentiated towards beta cells during development. Since the authors claim in this section that the differentiation medium promotes a mature ductal fate (line 329, 347), how are the canonical markers that have been mentioned by the authors in other sections induced in the differentiation medium (SLC4A4, KRT7, S100A2, CFTR, etc.)?

We appreciate the chance to clarify our point. The references indeed point to pathways involved in ductal cell fate that, when perturbed, lead to endocrine cell differentiation and loss of ductal cell identity. Further the one-carbon compound transport genes relate to carbonic anhydrases, prototypical ductal marker genes relate to carbon transfer, which are significantly upregulated upon differentiation media exposure. Since we acknowledge that cell fate mapping is a complicated issue, we have updated the results section (line 336) to include this sentence "but a more thorough comparison is needed to further substantiate this claim" and acknowledge the limitations of our interpretations.

PPDO-HPDO comparison

The analysis comparing porcine and human organoid systems is still not conclusive.

The way the proteomic data is presented does not allow comparing the two organoid systems. Fig3E compares PPDOs from different stages and shows that EPN(LPS) have decreased ductal markers. Fig3G shows that HPDO has increased ductal markers and decreased acinar compared to the exocrine pancreas fraction. Thus, these comparisons of unrelated conditions do not allow to conclude on the similarity of the human and porcine samples. Protein rank plots showing how ductal/acinar/stem markers rank within all the detected proteins in both porcine and human organoid samples can help assessing their ductal character (relevant also for previous comment in "Ductal identity confirmation" section) and similarity at proteome level between porcine and human organoid systems.

We agree with the reviewer that perhaps the comparison of the proteomics datasets between pig and human can be expanded. Given the limitations of the proteomics data analysis as in cross-species comparison and integration, we took the 100 highly expressed proteins of the PPDO (ePN and early/late passage) and HPDO and constructed a Venn diagram shown below. It is evident that around 50 proteins are shared between pig and human organoids in this basic analysis, which a big percentage considering species differences and technology/computational analysis limitations. This Venn diagram will be published as part of the RPF and will be available to the community as further proof of the PPDO-HPDO similarities. We also report this in the revised manuscript (line 181) and include a tab in the Dataset EV1 with this exact comparison.

Drug screening

Authors show in Fig6 B and C that the same compounds have different effects in PPDOs depending on if they are of adult or embryonic origin in the primary screening. In the secondary screening, the authors pool 2 late post-natal and 1 adult line and use 1 data point per line as replicate. This could influence the big spread between replicates observed in the

secondary screening. Independently of the age effect, and as stated by the authors, great donor variability is observed in organoid cultures generally. Thus, at least 3 replicates per donor line should be used to assess whether the compounds affect proliferation in a donor-dependent manner. As it is right now, it is not possible to assess if the lack of response is real, is due to technical variability, or if potential donor-specific effects are masked by lack of replicates. Additionally, in Fig6O authors should have included a condition where organoids were grown in complete medium to assess how much the compound effects compare to the positive control media condition. It is stated in the text (line 378-379) but not shown in Figure6.

We agree with the reviewer that, given setup feasibility, technical replicates are of importance to remove technical noise on large scale assays. We argue, however, that our total of 6-8 biological replicates across two species also provides a clear picture of the effect sizes that are biologically meaningful. Nevertheless, we acknowledge the importance of the technical replicates in our updated discussion in lines 519-521. The information on the complete media relating to Fig6O was part of the previous version of the manuscript but the comparison was removed during revisions following previous reviewers' comments for clarity.

Minor points

I wonder if in line 82 "Stem cell-derived organoids" should read "Induced-Pluripotent stem cell organoids".

The phrasing is correct as stem-cells can refer to both iPSC or embryonic derived stem cells

In line 305 there is a reference to figure panel missing

We added the callout from the next sentence below.

Conclusion on lines 328-329 is not substantiated by data in figure 5

See discussion point above.

Noggin is a BMP inhibitor. If you remove noggin from the medium, you are increasing BMP signaling in organoid cells, not removing BMP signals (lines 264-266)

We have adopted the text to incorporate this information more clearly.

Referee #3 (Remarks for Author):

The authors thoroughly addressed all the comments raised, significantly enhancing the strength of the manuscript.

17th Oct 2025

Dear Prof. Lickert, Dear Heiko,

Thank you for submitting your revised files. I am pleased to inform you that your manuscript is now accepted for publication in EMBO Molecular Medicine!

Please note that the nomenclature in the Appendix figure legends still needs to be changed to "Appendix Figure S1" etc. Once you have made the change, please send me the revised file via email, and I'll upload it in our system before sending your manuscript to our publisher.

With kind regards,
Lise
